# Non-asymptotic Analysis of Biased Adaptive Stochastic Approximation

Sobihan Surendran[1,2]*, Adeline Fermanian[2], Antoine Godichon-Baggioni[1], Sylvain Le Corff[1]
[1]Sorbonne Université, CNRS,
Laboratoire de Probabilités, Statistique et Modélisation, Paris, France
[2]LOPF, Califrais' Machine Learning Lab, Paris, France

## Abstract

Stochastic Gradient Descent (SGD) with adaptive steps is widely used to train deep neural networks and generative models. Most theoretical results assume that it is possible to obtain unbiased gradient estimators, which is not the case in several recent deep learning and reinforcement learning applications that use Monte Carlo methods. This paper provides a comprehensive non-asymptotic analysis of SGD with biased gradients and adaptive steps for non-convex smooth functions. Our study incorporates time-dependent bias and emphasizes the importance of controlling the bias of the gradient estimator. In particular, we establish that Adagrad, RMSProp, and AMSGRAD, an exponential moving average variant of Adam, with biased gradients, converge to critical points for smooth non-convex functions at a rate similar to existing results in the literature for the unbiased case. Finally, we provide experimental results using Variational Autoenconders (VAE) and applications to several learning frameworks that illustrate our convergence results and show how the effect of bias can be reduced by appropriate hyperparameter tuning.

## 1 Introduction

Stochastic Gradient Descent (SGD) algorithms are standard methods to train statistical models based on deep architectures. Consider a general optimization problem:

$$\theta^* \in \operatorname*{argmin}_{\theta \in \mathbb{R}^d} V(\theta) \,, \tag{1}$$

where $V$ is the objective function. Then, Gradient Descent methods produce a sequence of parameter estimates as follows: $\theta_0 \in \mathbb{R}^d$ and for all $n \in \mathbb{N}$,

$$\theta_{n+1} = \theta_n - \gamma_{n+1} \nabla V(\theta_n) \,,$$

where $\nabla V$ denotes the gradient of $V$ and for all $n \geq 1$, $\gamma_n > 0$ is the learning rate. In many cases, it is not possible to compute the exact gradient of the objective function, hence the introduction of vanilla Stochastic Gradient Descent, defined for all $n \in \mathbb{N}$ by:

$$\theta_{n+1} = \theta_n - \gamma_{n+1} \widehat{\nabla V}(\theta_n) \,,$$

where $\widehat{\nabla V}(\theta_n)$ is an estimator of $\nabla V(\theta_n)$. For example, in deep learning, stochasticity emerges with the use of mini-batches. While these algorithms have been extensively studied, both theoretically and practically, see, e.g., [10], many questions remain open. In particular, most results are based on the case where the estimator $\widehat{\nabla V}$ is unbiased. Although this assumption is valid in the case of vanilla SGD, it breaks down in many common applications. For example, zeroth-order methods used

---

*Corresponding author: sobihan.surendran@sorbonne-universite.fr

38th Conference on Neural Information Processing Systems (NeurIPS 2024).

to optimize black-box functions [61] in generative adversarial networks [58, 16] have access only to noisy biased realizations of the objective functions.

Furthermore, in reinforcement learning algorithms such as Q-learning [42], policy gradient [5], and temporal difference learning [8, 52, 18], gradient estimators are often obtained using a Markov chain with state-dependent transition probability. These estimators are then biased [69, 23]. Other examples of biased gradients can be found in the field of generative modeling with Markov Chain Monte Carlo (MCMC) and Sequential Monte Carlo (SMC) [34, 13]. In particular, the Importance Weighted Autoencoder (IWAE) proposed by [12], which is an extension of the standard Variational Autoencoder (VAE) [48], yields biased estimators. Finally, this is also the case in Bilevel Optimization [43, 36, 41] and Conditional Stochastic Optimization [40, 39].

Moreover, in practical applications, vanilla SGD shows difficulties in calibrating the step sequences. Therefore, modern variants of SGD employ adaptive steps that use past stochastic gradients or Hessians to avoid saddle points and deal with ill-conditioned problems. The idea of adaptive steps was first proposed in the online learning literature by [4] and later adopted in stochastic optimization, with the Adagrad algorithm of [27].

In this paper, we give non-asymptotic convergence guarantees for modern variants of SGD where both the estimators are biased and the steps are adaptive. To our knowledge, existing results consider either adaptive steps but unbiased estimators [27, 77, 67, 74, 19], or biased estimators with non-adaptive steps [70, 44, 2, 22, 21].

More precisely, our contributions are summarized as follows.

- We provide convergence guarantees for the Biased Adaptive Stochastic Approximation framework, under weak assumptions on the bias. To the best of our knowledge, these are the first convergence results to incorporate adaptive steps in biased Stochastic Approximation.

- In particular, we establish that Adagrad, RMSProp, and AMSGRAD, an exponential moving average variant of Adam, with a biased gradient, converge to a critical point for non-convex smooth functions with a convergence rate of $\mathcal{O}(\log n/\sqrt{n} + b_n)$, where $b_n$ is related to the bias at iteration $n$. However, we achieve an improved linear convergence rate with the Polyak-Łojasiewicz (PL) condition.

- Finally, we show how our theoretical results apply to several applications with biased gradients. In particular, we show that our hypotheses hold for Stochastic Bilevel Optimization and Conditional Stochastic Optimization, but also for Self-Normalized Importance Sampling estimators or Coordinate Sampling. We also propose a first non-asymptotic bound on the bias of IWAE, which allows us to illustrate through several experiments the effect of bias on the convergence of the optimization, and to show how this effect can be reduced by an appropriate choice of hyperparameters.

**Organization of the paper.** In Section 2, we introduce the setting of the paper and relevant related works. In Section 3, we present the Adaptive Stochastic Approximation framework and the main assumptions. In Section 4, we present our principal results, i.e., convergence rates for the risk when the PL condition is assumed, and on the gradient norm without this hypothesis. We illustrate our results in Section 5. All proofs are postponed to the appendix.

## 2 Setting and Related Works

**Stochastic Approximation.** Stochastic Approximation (SA) methods go far beyond SGD. They consist of sequential algorithms designed to find the zeros of a function when only noisy observations are available. Indeed, [68] introduced the Stochastic Approximation algorithm as an iterative recursive algorithm to solve the following integration equation:

$$h(\theta) = \mathbb{E}_\pi \left[ H_\theta(X) \right] = \int_{\mathsf{X}} H_\theta(x)\pi(x)\mathrm{d}x = 0 \,, \tag{2}$$

where $h$ is the mean field function, $X$ is a random variable taking values in a measurable space $(\mathsf{X}, \mathcal{X})$, and $\mathbb{E}_\pi$ is the expectation under the distribution $\pi$. In this context, $H_\theta$ can be any arbitrary function. If $H_\theta(X)$ is an unbiased estimator of the gradient of the objective function, then $h(\theta) = \nabla V(\theta)$. As

a result, the minimization problem (1) is then equivalent to solving problem (2), and we can note that SGD is a specific instance of SA. SA methods are then defined as follows:

$$\theta_{n+1} = \theta_n - \gamma_{n+1} H_{\theta_n}(X_{n+1}) \ ,$$

where the term $H_{\theta_n}(X_{n+1})$ is the $n$-th stochastic update, also known as the drift term, and is a potentially biased estimator of $\nabla V(\theta_n)$. It depends on a random variable $X_{n+1}$ which takes its values in $(\mathsf{X}, \mathcal{X})$. In machine learning, $V$ typically represents the theoretical risk, $\theta$ the model parameters, and $X_{n+1}$ the data.

**Adaptive Stochastic Gradient Descent.** SGD can be traced back to [68], and its averaged counterpart was proposed by [66]. The non-asymptotic analysis of SGD in both convex and strong convex cases can be found in [59]. [32] prove the convergence of a random iterate of SGD for nonconvex smooth functions, which was already suggested by the results of [9]. They show that SGD with constant or decreasing stepsize $\gamma_n = 1/\sqrt{n}$ converges to a stationary point of a non-convex smooth function $V$ at a rate of $\mathcal{O}(1/\sqrt{n})$ where $n$ is the number of iterations.

Most adaptive first-order methods, such as Adam [47], Adadelta [78], RMSProp [72], and NADA [25], are based on the blueprint provided by the Adagrad family of algorithms. The first known work on adaptive steps for non-convex stochastic optimization, in the asymptotic case, was presented by [50]. [74] proved that Adagrad converges to a critical point for non-convex objectives at a rate of $\mathcal{O}(\log n/\sqrt{n})$ when using a scalar adaptive step. In addition, [79] extended this proof to multidimensional settings. More recently, [19] focused on the convergence rates for Adagrad and Adam. Furthermore, several modified versions of Adam have been proposed, such as AMSGRAD [77] and YOGI [67].

**Biased Stochastic Approximation.** The asymptotic results of Biased SA have been studied by [70]. The non-asymptotic analysis can be found in the reinforcement learning literature, especially in the context of temporal difference (TD) learning, as explored by [8, 52, 18]. The case of non-convex smooth functions has been studied by [44]. The authors establish convergence results for the mean field function at a rate of $\mathcal{O}(\log n/\sqrt{n} + b)$, where $b$ corresponds to the bias and $n$ to the number of iterations. For strongly convex functions, the convergence of SGD with biased gradients can be found in [2], specifically addressing the case of Martingale noise with a constant step size.

[46, 21] introduce a novel assumption, known as "Expected Smoothness", which is the weakest assumption compared to the existing literature on biased SGD that we extend to cover the adaptive case. The authors provide convergence results in the case of non-convex smooth functions. Convergence results with assumptions on the control of bias and MSE can be found in [56, 22]. Applications of biased gradients can be found in Bilevel Optimization [43, 36, 41] and Conditional Stochastic Optimization [40, 39]. Moreover, biased gradients are also used in various other applications [38, 54, 6, 56]. Finally, [3] studied convergence results of biased gradients with Adagrad in the Markov chain case, focusing on the norm of the gradient of the Moreau envelope while assuming the boundedness of the objective function.

Our analysis provides non-asymptotic results in a more general setting, for a wide variety of objective functions and adaptive algorithms and treating both the Martingale and Markov chain cases.

## 3 Adaptive Stochastic Approximation

### 3.1 Framework

Consider the optimization problem (1) where the objective function $V$ is assumed to be differentiable. In this paper, we focus on the following SA algorithm with adaptive steps: $\theta_0 \in \mathbb{R}^d$ and for all $n \in \mathbb{N}$,

$$\theta_{n+1} = \theta_n - \gamma_{n+1} A_n H_{\theta_n}(X_{n+1}) \ , \tag{3}$$

where $\gamma_{n+1} > 0$ and $A_n$ is a sequence of symmetric and positive definite matrices. In a context of biased gradient estimates, choosing

$$A_n = \left[ \delta I_d + \left( \frac{1}{n+1} \sum_{k=0}^{n} H_{\theta_k}(X_{k+1}) H_{\theta_k}(X_{k+1})^\top \right) \right]^{-1/2}$$

can be assimilated to the full Adagrad algorithm [27]. However, computing the square root of the inverse becomes expensive in high dimensions, so in practice, Adagrad is often used with diagonal

matrices. This approach has been shown to be particularly effective in sparse optimization settings. Denoting by $\mathrm{Diag}(A)$ the matrix formed with the diagonal terms of $A$ and setting all other terms to 0, Adagrad with diagonal matrices is defined in our context as:

$$A_n = \left[\delta I_d + \mathrm{Diag}\big(\bar{H}_n(X_{1:n+1}, \theta_{0:n})\big)\right]^{-1/2}, \tag{4}$$

where

$$\bar{H}_n(X_{1:n+1}, \theta_{0:n}) = \frac{1}{n+1} \sum_{k=0}^{n} H_{\theta_k}(X_{k+1}) H_{\theta_k}(X_{k+1})^{\top}.$$

In RMSProp [72], $\bar{H}_n(X_{1:n+1}, \theta_{0:n})$ in (4) is an exponential moving average of the past squared gradients, defined by:

$$\bar{H}_n(X_{1:n+1}, \theta_{0:n}) = (1 - \rho) \sum_{k=0}^{n} \rho^{n-k} H_{\theta_k}(X_{k+1}) H_{\theta_k}(X_{k+1})^{\top},$$

where $\rho$ is the moving average parameter. Furthermore, when $A_n$ is a recursive estimate of the inverse Hessian, it corresponds to the Stochastic Newton algorithm [11].

## 3.2 Assumptions

Consider the following assumptions.

**H1** There exists a constant $\mu > 0$ such that for all $\theta \in \mathbb{R}^d$,

$$2\mu\big(V(\theta) - V(\theta^*)\big) \leq \|\nabla V(\theta)\|^2.$$

H1 corresponds to the Polyak-Łojasiewicz condition, which is weaker than strong convexity and remains satisfied even when the function is non-convex. It ensures uniqueness of the minimizer $\theta^*$. The PL condition has been extensively studied theoretically [45] and has been verified empirically in many applications, such as over-parameterized deep networks [26] and Linear Quadratic Regulator models [29].

**H2** The objective function $V$ is $L$-smooth. For all $(\theta, \theta') \in \mathbb{R}^d \times \mathbb{R}^d$,

$$\|\nabla V(\theta) - \nabla V(\theta')\| \leq L\|\theta - \theta'\|.$$

This assumption is crucial to obtain our convergence rate and is very common see, e.g., [59, 10]. Under this assumption, for all $(\theta, \theta') \in \mathbb{R}^d \times \mathbb{R}^d$,

$$V(\theta) \leq V(\theta') + \langle \nabla V(\theta'), \theta - \theta'\rangle + \frac{L}{2}\|\theta - \theta'\|^2. \tag{5}$$

**H3** ($i$) Biased Gradients: There exist two non-increasing positive sequences $(\lambda_n)_{n \geq 1}$ and $(r_n)_{n \geq 1}$ such that for all $n \in \mathbb{N}$,

$$\mathbb{E}\big[\langle \nabla V(\theta_n), A_n H_{\theta_n}(X_{n+1})\rangle\big] \geq \lambda_{n+1}\big(\mathbb{E}\big[\|\nabla V(\theta_n)\|^2\big] - r_{n+1}\big).$$

($ii$) Expected Smoothness: there exists a non-increasing non-negative sequence $(\sigma_n^2)_{n \geq 1}$, and positive constants $\tilde{\sigma}_1, \tilde{\sigma}_2$ such that for all $n \in \mathbb{N}$,

$$\mathbb{E}\big[\|H_{\theta_n}(X_{n+1})\|^2\big] \leq \sigma_n^2 + \tilde{\sigma}_1 \mathbb{E}\big[\|\nabla V(\theta_n)\|^2\big] + \tilde{\sigma}_2 \mathbb{E}\big[V(\theta_n) - V(\theta^*)\big].$$

In this assumption, $r_{n+1}$ represents an additive bias term, generally of the order of the square of the bias, and $\lambda_{n+1}$ may depend on the minimum eigenvalue of $A_n$. In [21, Theorem 2], it has been demonstrated that this assumption is weaker than the alternatives used in the literature on biased SGD. We have extended these assumptions to the adaptive case. It is important to note that the first point of H3 depends on the application (objective function $V$) and on the adaptive algorithm (matrix $A_n$) that we want to use. The purpose of this assumption is to provide a more general framework that covers all possible applications and adaptive algorithms. In the biased SGD setting, if the bias term $\|\mathbb{E}[H_{\theta_n}(X_{n+1}) \mid \mathcal{F}_n] - \nabla V(\theta_n)\|$ is bounded by $\tilde{b}_{n+1}$, we can easily verify the first point of

H3 by considering $\lambda_{n+1} = 1/2$ and $r_{n+1} = \tilde{b}_{n+1}^2$. We show in Section 4.3 that this assumption is also verified in algorithms such as Adagrad and RMSProp. The second point of H3 is a weaker assumption compared to bounding the variance of the noise term. Applications where we can verify these assumptions are discussed in Appendix D.

We finally consider an additional assumption on $A_n$. Let $\|A\|$ be the spectral norm of a matrix $A$.

**H4** There exists $(\beta_n)_{n \geq 1}$ such that for all $n \in \mathbb{N}$, $\|A_n\| := \lambda_{\max}(A_n) \leq \beta_{n+1}$ .

In our setting, since $A_n$ is assumed to be a symmetric matrix, the spectral norm is equal to the largest eigenvalue. H4 plays a crucial role, as the estimates may diverge when this assumption is not satisfied. Given a sequence $(\beta_n)_{n \geq 1}$, one way to ensure that H4 is satisfied is to replace the random matrices $A_n$ with

$$\tilde{A}_n = \frac{\min\{\|A_n\|, \beta_{n+1}\}}{\|A_n\|} A_n . \tag{6}$$

It is then clear that $\|\tilde{A}_n\| \leq \beta_{n+1}$. Furthermore, in most cases, especially for Adagrad, RMSProp and Stochastic Newton, control of $\lambda_{\max}(A_n)$ in H4 is satisfied. For example, in Adagrad and RMSProp, in (4), we have $\lambda_{\max}(A_n) \leq \delta^{-1/2}$.

## 4 Convergence Results

### 4.1 Convergence under the PL condition

In this section, we study the convergence rate of SGD with biased gradients and adaptive steps under the PL condition. We give below a simplified version of the bound we obtain on the risk and refer to Theorem A.2 in the appendix for a formal statement with explicit constants.

**Theorem 4.1.** *Assume that H1 - H4 hold. Let $\theta_n \in \mathbb{R}^d$ be the $n$-th iterate of the recursion* (3) *and $\gamma_n = C_\gamma n^{-\gamma}, \beta_n = C_\beta n^\beta, \lambda_n = C_\lambda n^{-\lambda}$ with $C_\gamma > 0, C_\beta > 0$, and $C_\lambda > 0$. Assume that $\gamma, \beta, \lambda \geq 0$ and $\gamma + \lambda < 1$. Then,*

$$\mathbb{E}[V(\theta_n) - V(\theta^*)] = \mathcal{O}\left(n^{-\gamma+2\beta+\lambda} + r_n\right). \tag{7}$$

The rate obtained is classical and shows the tradeoff between a term coming from the adaptive steps (with a dependence on $\gamma$, $\beta$, $\lambda$) and a term $r_n$ which depends on the control of the bias. To minimize the right hand-side of (7), we would like to have $\beta = \lambda = 0$. For example, it is verified in the case of Adagrad and RMSProp if the gradients are bounded, as will be discussed later.

We stress that Theorem 4.1 applies to any adaptive algorithm of the form (3), with the only assumption being H4. Without any information on these eigenvalues, the choice that $\beta_n \propto n^\beta$ and $\lambda_n \propto n^{-\lambda}$ allows us to remain very general, which can even be seen as a worst-case scenario. Finally, note that non-adaptive SGD is a particular case of Theorem 4.1. Thus, our theorem gives new results also in the non-adaptive case with generic step sizes and biased gradients with decreasing bias.

### 4.2 Convergence without the PL condition

In the non-convex smooth case, theoretical results are generally based on a randomized version of SA, as described in [60, 32, 44]. Instead of considering the final parameter $\theta_n$, we introduce a random variable $R$, which takes its values in $\{1, \ldots, n\}$, and the quantity of interest becomes $\theta_R$. Note that this procedure is a technical tool, in practical applications we use classical SA. The following theorem provides a bound in expectation on the gradient of the objective function $V$, which is the best we can have given that no assumption is made about the existence of a global minimum of $V$.

**Theorem 4.2.** *Assume that H2 - H4 hold. Assume also that for all $k \in \mathbb{N}$, we have $\gamma_{k+1} \leq \lambda_{k+1}/(\tilde{\sigma}_1 L \beta_{k+1}^2)$. For any $n \geq 1$, let $R \in \{0, \ldots, n\}$ be a discrete random variable such that:*

$$\mathbb{P}(R = k) := \frac{w_{k+1}\gamma_{k+1}\lambda_{k+1}}{\sum_{j=0}^n w_{j+1}\gamma_{j+1}\lambda_{j+1}} ,$$

*where $w_{k+1} = \prod_{j=1}^{k+1}(1 + \tilde{\sigma}_2 \delta_j)^{-1}$ with $\delta_j = L\gamma_j^2 \beta_j^2/2$. Then,*

$$\mathbb{E}\left[\|\nabla V(\theta_R)\|^2\right] \leq 2 \frac{V^* + \alpha_{1,n} + \alpha_{2,n}}{\sum_{j=0}^n w_{j+1}\gamma_{j+1}\lambda_{j+1}} ,$$

*where*

$$\alpha_{1,n} = \sum_{k=0}^{n} w_{k+1}\gamma_{k+1}\lambda_{k+1}r_{k+1} \ , \ \ \alpha_{2,n} = \sum_{k=0}^{n} w_{k+1}\delta_{k+1}\sigma_k^2, \ \ and \ \ V^* = \mathbb{E}[V(\theta_0) - V(\theta^*)] \ .$$

If $\tilde{\sigma}_2 = 0$, Theorem 4.2 recovers the asymptotic convergence rate obtained by [44] with respect to the hyperparameters $\gamma$, $\beta$, and $\lambda$, and to the bias. We can observe that if $\gamma \leq \lambda + 2\beta$, the condition on $(\gamma_k)_{k \geq 1}$ can be met simply by tuning $C_\gamma$. In particular, if $A_n = I_d$, the requirement on the step sizes can be expressed as $\gamma_{k+1} \leq 1/(\tilde{\sigma}_1 L)$.

We give below the convergence rates obtained from Theorem 4.2 under the same assumptions on $\gamma_n$, $\beta_n$, and $\lambda_n$ as in Theorem 4.1.

**Corollary 4.3.** *Assume that H2-H4 hold. Let $\gamma_n = C_\gamma n^{-\gamma}, \beta_n = C_\beta n^\beta, \lambda_n = C_\lambda n^{-\lambda}$ with $C_\gamma > 0, C_\beta > 0$, and $C_\lambda > 0$. Assume that $\gamma, \beta, \lambda \geq 0$ and $\gamma + \lambda < 1$. Then, if $\tilde{\sigma}_2 = 0$, we have:*

$$\mathbb{E}\Big[ \|\nabla V(\theta_R)\|^2 \Big] = \begin{cases} \mathcal{O}\left(n^{-\gamma+\lambda+2\beta} + b_n\right) & if \ \gamma - \beta < 1/2 \ , \\ \mathcal{O}\left(n^{\gamma+\lambda-1} + b_n\right) & if \ \gamma - \beta > 1/2 \ , \\ \mathcal{O}\left(n^{\gamma+\lambda-1}\log n + b_n\right) & if \ \gamma - \beta = 1/2 \ , \end{cases}$$

*where the bias term $b_n$ can be constant or decreasing. In the latter case, writing $r_n = C_r n^{-r}$, we have:*

$$b_n = \begin{cases} \mathcal{O}\left(n^{-r}\right) & if \ r + \lambda + \gamma < 1 \ , \\ \mathcal{O}\left(n^{\gamma+\lambda-1}\right) & if \ r + \lambda + \gamma > 1 \ , \\ \mathcal{O}\left(n^{\gamma+\lambda-1}\log n\right) & if \ r + \lambda + \gamma = 1 \ . \end{cases}$$

In practice, the value of $r$ is known in advance while the other parameters can be tuned to achieve the optimal rate of convergence. In any scenario, we can never achieve a bound of $\mathcal{O}(1/\sqrt{n} + b_n)$, and the best rate we can reach is $\mathcal{O}(\log n/\sqrt{n} + b_n)$ when $\gamma = 1/2, \beta = 0$, and $\lambda = 0$. In this case, all eigenvalues of $A_n$ must be bounded from both below and above. Note that we could also have obtained such a rate by taking $\lambda_n = n^{-1/2}$ and $\beta_n = n^{-1/2}$ while keeping $\gamma_n$ constant. However, the assumption that $\beta_n = n^{-1/2}$ is too strong (fast decay of the eigenvalues of $A_n$), hence our choice of $\beta_n = C_\beta n^\beta$. Finally, for a decreasing bias, if $r \geq 1/2$, the bias term contributes to the convergence rate of the algorithm. Otherwise, the other term is the leading term of the upper bound. In both cases, the best achievable bound is $\mathcal{O}(\log n/\sqrt{n})$ if $r \geq 1/2$.

**Bounded Gradient Case.** Now, we analyze the convergence of Randomized Adaptive Stochastic Approximation when the stochastic updates are bounded, as given by the following assumption.

**H5** There exists $M \geq 0$ such that for all $n \in \mathbb{N}$, $\|H_{\theta_n}(X_{n+1})\| \leq M$.

Boundedness of the stochastic gradient of the objective function is a classical assumption in adaptive stochastic optimization [67, 74, 19, 73].

**Corollary 4.4.** *Assume that H2-H5 hold. Let $\gamma_n = C_\gamma n^{-\gamma}, \beta_n = C_\beta n^\beta, \lambda_n = C_\lambda n^{-\lambda}$ with $C_\gamma > 0, C_\beta > 0$, and $C_\lambda > 0$. Assume that $\gamma, \beta, \lambda \geq 0$ and $\gamma + \lambda < 1$. For any $n \geq 1$, let $R \in \{0, \dots, n\}$ be a uniformly distributed random variable. Then,*

$$\mathbb{E}\left[\|\nabla V(\theta_R)\|^2\right] \leq \frac{V^* + \alpha'_{1,n} + LM^2\alpha'_{2,n}/2}{\sqrt{n}} \ ,$$

*where $\alpha'_{1,n} = \sum_{k=0}^{n} \gamma_{k+1}\lambda_{k+1}r_{k+1}$, $\alpha'_{2,n} = \sum_{k=0}^{n} \gamma_{k+1}^2\beta_{k+1}^2$, and $V^* = \mathbb{E}[V(\theta_0) - V(\theta^*)]$.*

Importantly, in Corollary 4.4, there are no assumptions on the step sizes, and we obtain a better bound than in Theorem 4.2.

## 4.3 Application to Adagrad and RMSProp

We give a convergence analysis of Adagrad and RMSProp with a biased gradient estimator. First, note that, under H5, for all eigenvalues $\lambda$ of $A_n$, the adaptive matrix in Adagrad or RMSProp, it holds that $(M^2 + \delta)^{-1/2} \leq \lambda \leq \delta^{-1/2}$, i.e., H4 is satisfied with $\lambda = 0$ and $\beta = 0$.

**Corollary 4.5.** *Assume that H2 and H5 hold. Let $\gamma_n = c_\gamma n^{-1/2}$ and $A_n$ denote the adaptive matrix in Adagrad or RMSProp. For any $n \geq 1$, let $R \in \{0, \ldots, n\}$ be a uniformly distributed random variable. Suppose that for any $n \geq 1$, there exist positive constants $\alpha$ and $C_\alpha$ such that:*

$$\left\| \mathbb{E}\left[ H_{\theta_n}\left( X_{n+1} \right) | \mathcal{F}_n \right] - \nabla V\left( \theta_n \right) \right\| \leq C_\alpha n^{-\alpha} . \tag{8}$$

*Then,*

$$\mathbb{E}\left[ \left\| \nabla V\left( \theta_R \right) \right\|^2 \right] = \mathcal{O}\left( \frac{\log n}{\sqrt{n}} + b_n \right) ,$$

*where the bias $b_n$ is explicitly given in Appendix A.5.*

In the case of an unbiased gradient, we obtain the same bound of $\mathcal{O}(\log n / \sqrt{n})$ as in [74, 79, 19] under the same assumptions. If the bias is of the order $\mathcal{O}(n^{-1/4})$, the algorithm achieves the same convergence rate as in the case of an unbiased gradient.

## 4.4 AMSGRAD with Biased Gradients

Finally, we show the convergence of AMS-GRAD [67] with a biased gradient estimator. At each iteration, AMSGRAD uses an exponential moving average of past gradients instead of the current gradient as in Equation (3), which is detailed in Algorithm 1. The key difference between Adam and AMSGRAD lies in their handling of the second moment estimate. Specifically, AMSGRAD uses the updated term $\hat{V}_k = \max(\hat{V}_{k-1}, \text{Diag}(V_k))$ instead of directly using $V_k$, with the maximum taken coordinate-wise. This approach is crucial, as it ensures that the eigenvalues of $A_n$ decrease at each iteration. The following theorem provides a bound in expectation on the gradient of the objective function $V$ using randomized iterations with AMSGRAD.

---
**Algorithm 1 AMSGRAD with Biased Gradients**

---
**Input:** Initial point $\theta_0$, maximum number of iterations $n$, step sizes $\{\gamma_k\}_{k \geq 1}$, momentum parameters $\rho_1, \rho_2 \in [0, 1)$ and regularization parameter $\delta \geq 0$.
Set $m_0 = 0$, $V_0 = 0$ and $\hat{V}_0 = 0$
**for** $k = 0$ to $n - 1$ **do**
  Compute the stochastic update $H_{\theta_k}\left( X_{k+1} \right)$
  $m_k = \rho_1 m_{k-1} + (1 - \rho_1) H_{\theta_k}(X_{k+1})$
  $V_k = \rho_2 V_{k-1} + (1 - \rho_2) H_{\theta_k}(X_{k+1}) H_{\theta_k}(X_{k+1})^\top$
  $\hat{V}_k = \max\left( \hat{V}_{k-1}, \text{Diag}(V_k) \right)$
  $A_k = \left[ \delta I_d + \hat{V}_k \right]^{-1/2}$
  $\theta_{k+1} = \theta_k - \gamma_{k+1} A_k m_k$
**end for**
**Output:** $(\theta_k)_{0 \leq k \leq n}$

---

**Theorem 4.6.** *Assume that H2, H3 (i), and H5 hold. Let $\gamma_n = c_\gamma n^{-1/2}$, $A_n$ denote the adaptive matrix of AMSGRAD in Algorithm 1, and $\rho_1, \rho_2 \in [0, 1)$. For any $n \geq 1$, let $R \in \{0, \ldots, n\}$ be a uniformly distributed random variable. Then,*

$$\mathbb{E}\left[ \left\| \nabla V\left( \theta_R \right) \right\|^2 \right] = \mathcal{O}\left( \frac{\log n}{\sqrt{n}} + b_n \right) ,$$

*where $b_n$ corresponds to the bias which comes from $r_n$ in H3(i). Choosing $r_n = C_r n^{-r}$, we get:*

$$b_n = \begin{cases} \mathcal{O}\left( n^{-r} \right) & \text{if } r < 1/2 , \\ \mathcal{O}\left( n^{-1/2} \right) & \text{if } r > 1/2 , \\ \mathcal{O}\left( n^{-1/2} \log n \right) & \text{if } r = 1/2 . \end{cases}$$

If the bias is of the order $\mathcal{O}(n^{-1/4})$, we achieve a convergence rate of $\mathcal{O}(\log n / \sqrt{n})$, which is the same as that of an unbiased gradient [19] and similar to that of Adagrad and RMSProp. It is worth noting that our results are also applicable to SGD momentum by taking $A_n = I_d$ in Algorithm 1.

## 4.5 Convergence Results in i.i.d. and Markov Chain cases

For illustrative purposes, in this subsection we give the form of the bias of the gradient estimator, denoted by $\tilde{b}_n$, in two simple scenarios, i.e., when $\{X_n, n \in \mathbb{N}\}$ is either an i.i.d. sequence or a Markov chain. For Adagrad, RMSProp, and AMSGRAD, bounding the bias of the gradient estimator is a sufficient condition for verifying H3($i$), which in turn enables us to derive convergence results in each scenario.

**I.i.d. case.** Assume that $\{X_n, n \in \mathbb{N}\}$ are i.i.d. random variables. If the mean field function $h(\theta_n) = \mathbb{E}[H_{\theta_n}(X_{n+1}) \mid \mathcal{F}_n]$ aligns with the true gradient, then the estimator is unbiased. Otherwise, the bias of the gradient estimator is

$$\tilde{b}_{n+1} = \|h(\theta_n) - \nabla V(\theta_n)\| .$$

**Markov Chain case.** Assume now that $\{X_n, n \in \mathbb{N}\}$ is a Markov Chain. The bias consists of two parts: the difference between the mean field function and the true gradient, and a term due to the Markov chain dynamics. For all $T \geq 0$, we define the stochastic update as follows:

$$H_{\theta_k}(X_{k+1}) = \frac{1}{T} \sum_{i=1}^{T} H_{\theta_k}\left(X_{k+1}^{(i)}\right),$$

where $X_{k+1}^{(i)}$ represents the i-th sample generated at iteration $k+1$. This multi-sample estimator is commonly used in applications such as Reinforcement Learning, Markov Chain Monte Carlo, and Sequential Monte Carlo methods, effectively reducing the variance of the gradient estimator. The mixing time $\tau_{\text{mix}}$ of a Markov chain with stationary distribution $\pi$ and transition kernel $P$ is characterized as:

$$\tau_{\text{mix}} := \inf\left\{t \; ; \; \sup_x D_{\text{TV}}(P^t(x, \cdot), \pi) \leq \frac{1}{4}\right\},$$

where $D_{\text{TV}}$ denotes the total variation distance. For an ergodic Markov chain with stationary distribution $\pi$, the bias of this gradient estimator when using $T$ samples per step is

$$\tilde{b}_{n+1} = \|h(\theta_n) - \nabla V(\theta_n)\| + M\sqrt{\tau_{\text{mix}}/T} ,$$

where $h(\theta) = \int H_\theta(x)\pi(dx)$. If the general optimization problem reduces to the following stochastic optimization problem with Markov noise, as considered in most of the literature [28, 24, 7]:

$$\min_{\theta \in \mathbb{R}^d} V(\theta) := \mathbb{E}_{x \sim \pi}[f(\theta; x)],$$

where $\theta \mapsto f(\theta; x)$ is a loss function, and $\pi$ is some stationary data distribution of the Markov Chain and $H_{\theta_k}(X_{k+1}^{(i)}) = \nabla f(\theta_k; X_{k+1}^{(i)})$, then $\tilde{b}_{n+1} = M\sqrt{\tau_{\text{mix}}/T}$, similar to SGD with Markov Noise [24].

## 5 Applications and Experiments

### 5.1 Bilevel and Conditional Stochastic Optimization

We can now apply our theoretical results in various settings where biased gradients are involved. In particular, they apply to the fields of Stochastic Bilevel Optimization and Conditional Stochastic Optimization. Stochastic Bilevel Optimization consists of minimizing an objective function $V$ with respect to $\theta$, where $V$ is itself a function of $\phi^*(\theta)$ and $\phi^*(\theta)$ is obtained by solving another minimization problem. Conditional Stochastic Optimization focuses on optimizing the expected value of a function that contains a nested conditional expectation on a random variable $\eta$. We provide in Table 1 a summary of the assumptions satisfied in these settings, which allow to apply the results of Section 4 and to obtain a $\mathcal{O}(\log n/\sqrt{n} + b_n)$ convergence rate in both cases, and explicit forms for $b_n$. To our knowledge, these are the first convergence rates obtained in these settings.

We refer to Appendix D for other examples in which the bias of the estimator can be controlled, in particular Self-Normalized Importance Sampling (Appendix D.1), Sequential Monte Carlo Methods (Appendix D.2), Policy Gradient (Appendix D.3), Zeroth-Order Gradient (Appendix D.4), and Coordinate Sampling (Appendix D.5).

### 5.2 Experiments with IWAE and BR-IWAE

In this section, we illustrate our theoretical results in the context of deep VAE. The experiments were conducted using PyTorch [65], and the source code can be found here[2]. In generative models,

---

[2]https://github.com/SobihanSurendran/Adaptive-SA

Table 1: Bilevel and Conditional Stochastic Optimization with our Biased Adaptive SA framework.

| Applications | Stochastic Bilevel Optimization | Conditional Stochastic Optimization |
|---|---|---|
| Problem | $\min_{\theta \in \mathbb{R}^d} \mathbb{E}_\xi \left[ f(\theta, \phi^*(\theta); \xi) \right]$ 
 s.t. $\phi^*(\theta) \in \underset{\phi \in \mathbb{R}^q}{\operatorname{argmin}} \mathbb{E}_\zeta \left[ g(\theta, \phi; \zeta) \right]$ | $\min_{\theta \in \mathbb{R}^d} \mathbb{E}_\xi \left[ f_\xi \left( \mathbb{E}_{\eta|\xi} \left[ g_\eta(\theta, \xi) \right] \right) \right]$ |
| Lipchitz Constant H2 | Lemma C.2 | Lemma C.6 |
| Bias Control H3 | Lemma C.3 | Lemma C.5 |
| Gradient Bound H5 | Lemma C.3 | Lemma C.6 |
| Convergence | Theorem C.4 | Theorem C.7 |

the objective is to maximize the marginal likelihood $\log p_\theta(x)$, which is the marginalization of $(x, z) \mapsto p_\theta(x, z)$, where $x$ represents the observations and $z$ is the latent variable. Under some simple technical assumptions, by Fisher's identity, we have:

$$\nabla_\theta \log p_\theta(x) = \int \nabla_\theta \log p_\theta(x, z) p_\theta(z \mid x) \mathrm{d}z \ . \tag{9}$$

However, in most cases, the conditional density $z \mapsto p_\theta(z \mid x)$ is intractable and can only be sampled. Variational Autoencoders introduce an additional parameter $\phi$ and a family of variational distributions $z \mapsto q_\phi(z \mid x)$ to approximate the true posterior distribution. Parameters are estimated by maximizing the Evidence Lower Bound (ELBO):

$$\log p_\theta(x) \geq \mathbb{E}_{q_\phi(\cdot|x)} \left[ \log \frac{p_\theta(x, Z)}{q_\phi(Z \mid x)} \right] =: \mathcal{L}_{\text{ELBO}}(\theta, \phi; x) \ .$$

The Importance Weighted Autoencoder (IWAE) [12] is a variant of the VAE that incorporates importance weighting to obtain a tighter ELBO. The IWAE objective can be written as follows:

$$\mathcal{L}_k^{\text{IWAE}}(\theta, \phi; x) = \mathbb{E}_{q_\phi^{\otimes k}(\cdot|x)} \left[ \log \frac{1}{k} \sum_{\ell=1}^{k} \frac{p_\theta(x, Z^{(\ell)})}{q_\phi(Z^{(\ell)} \mid x)} \right],$$

where $k$ corresponds to the number of samples drawn from the variational posterior distribution. The estimator of the gradient of ELBO in IWAE is a biased estimator of $\nabla_\theta \log p_\theta(x)$. In Theorem B.1, we establish that the bias of this estimator is of order $\mathcal{O}(1/k)$, thereby allowing us to derive a convergence rate for IWAE. Since bias has an impact on convergence rates, we propose to use one of the bias reduction techniques, the Biased Reduced Importance Weighted Autoencoder (BR-IWAE) [14], which is detailed in Appendix B.

**Dataset and Model.** We conduct our experiments on the CIFAR-10 dataset [51] and use a Convolutional Neural Network (CNN) architecture with the Rectified Linear Unit (ReLU) activation function for both the encoder and the decoder. The latent space dimension is set to 100. We estimate the log-likelihood using VAE, IWAE, and BR-IWAE models, all of which are trained for 100 epochs.

Training is conducted using Adagrad, RMSProp, and Adam with a decaying learning rate. Although AMSGRAD is analyzed in our theoretical results, we use Adam for the experiments due to its widespread use in practice. Additional details are provided in Appendix E.

First, we set $k = 5$ samples in both IWAE and BR-IWAE. The test losses are presented in Figure 1. We show the negative log-likelihood on the test dataset for VAE, IWAE, and BR-IWAE with Adagrad, RMSProp, and Adam. As expected, we observe that IWAE outperforms VAE, while BR-IWAE outperforms IWAE by reducing bias in all cases.

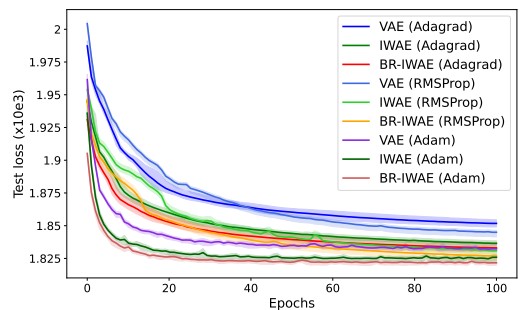

Figure 1: Negative Log-Likelihood on the test set for Different Generative Models with Adagrad, RMSProp, and Adam on CIFAR-10. Bold lines represent the mean over 5 independent runs.

Then, we illustrate empirically the convergence rates obtained in Corollary 4.5 and Theorem 4.6 for IWAE. Since the bias of the estimator of the gradient in IWAE is of the order $\mathcal{O}(1/k)$, choosing a bias of order $\mathcal{O}(n^{-\alpha})$ is equivalent to using $n^{\alpha}$ samples at iteration $n$ to estimate the gradient. We plot in Figure 2 the gradient squared norm $\|\nabla V(\theta_n)\|^2$ and the Negative Log-Likelihood is given in Appendix E.2. Note that all figures are with respect to epochs, whereas here, $n$ represents the number of updates of the gradient. The dashed curves correspond to the expected convergence rate $\mathcal{O}(n^{-1/4})$ for $\alpha = 1/8$ and $\mathcal{O}(\log n/\sqrt{n})$ for $\alpha = 1/4$ and $\alpha = 1/2$.

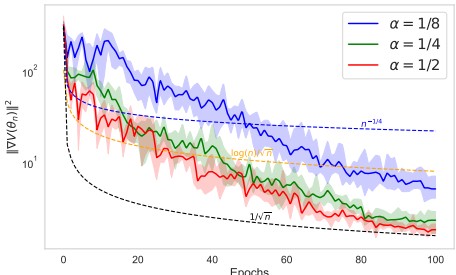 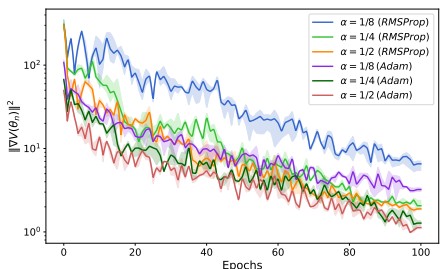

Figure 2: Value of $\|\nabla V(\theta_n)\|^2$ in IWAE with Adagrad (on the left), RMSProp, and Adam (on the right). Bold lines represent the mean over 5 independent runs. Figures are plotted on a logarithmic scale for better visualization. Both figures have the same scale, so we have not shown the dashed theoretical curves on the right for better clarity.

We observe that the algorithms converge at the expected theoretical rates, and even faster. In Appendix E.2, we have included an additional experiment on the FashionMNIST dataset [76], which shows similar behavior, but the convergence is closer to the expected rates, suggesting that our upper bounds may be tight. We see similar convergence rates for Adagrad, RMSProp, and Adam, although, as expected, Adam performs slightly better. Moreover, it is clear that convergence is faster with a larger $\alpha$ but beyond a certain threshold for $\alpha$ the rate of convergence does not change significantly. Since choosing a larger $\alpha$ induces an additional computational cost, it is crucial to choose an appropriate value that achieves fast convergence without being too computationally expensive. Choosing an optimal number of samples at each iteration remains an open problem depending on the chosen generative model.

## 6 Discussion

This paper provides a non-asymptotic analysis of Biased Adaptive Stochastic Approximation with and without the PL condition in the non-convex smooth setting. We derive a convergence rate of $\mathcal{O}(\log n/\sqrt{n} + b_n)$ for non-convex smooth functions, where $b_n$ corresponds to the time-dependent decreasing bias, and an improved linear convergence rate with the Polyak-Łojasiewicz (PL) condition. We also establish that Adagrad, RMSProp, and AMSGRAD with biased gradients converge to critical points for non-convex smooth functions. Our results provide insights on hyper-parameters tuning to achieve fast convergence and reduce computational time. A natural extension of this work is the analysis of the assumptions, the bias and convergence rates for specific deep learning architectures. A theoretical analysis of the Monte Carlo effort required at each iteration to obtain an optimal convergence rate is another interesting perspective.

## Acknowledgements

The Ph.D. of Sobihan Surendran was funded by the Paris Region PhD Fellowship Program of Région Ile-de-France. We would like to thank SCAI (Sorbonne Center for Artificial Intelligence) for providing the computing clusters. We also express our gratitude to the reviewers for their insightful comments and suggestions, which have helped improve this paper.

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

# Supplementary Material for "Non-asymptotic Analysis of Biased Adaptive Stochastic Approximation"

## Table of Contents

# A Convergence Proofs

## A.1 Proof of Theorem 4.1

We first establish a technical lemma which is essential for the proof.

**Lemma A.1.** *Let $(\delta_n)_{n \geq 0}, (\gamma_n)_{n \geq 1}, (\eta_n)_{n \geq 1}$, and $(v_n)_{n \geq 1}$ be some positive sequences satisfying the following assumptions.*

- *The sequence $\delta_n$ follows the recursive relation:*

$$\delta_n \leq (1 - 2\omega\gamma_n + \eta_n\gamma_n)\,\delta_{n-1} + v_n\gamma_n \ ,$$

  *with $\delta_0 \geq 0$ and $\omega > 0$.*

- *Let $n_0 = \inf\{n \geq 1 : \eta_n \leq \omega\}$, then for all $n \geq n_0 + 1$, we assume that $\omega\gamma_n \leq 1$.*

*Then, for all $n \in \mathbb{N}$,*

$$\delta_n \leq \exp\left(-\omega \sum_{k=n/2}^{n} \gamma_k\right) \exp\left(2\sum_{k=1}^{n} \eta_k\gamma_k\right)\left(\delta_0 + 2 \max_{1 \leq k \leq n} \frac{v_k}{\eta_k}\right) + \frac{1}{\omega} \max_{n/2 \leq k \leq n} v_k.$$

The proof is given in [35, Proposition 6.1]

**Theorem A.2.** *Assume that H1 - H4 hold. Let $\theta_n \in \mathbb{R}^d$ be the n-th iterate of the recursion* (3). *Then,*

$$\mathbb{E}\left[V\left(\theta_n\right) - V(\theta^*)\right] \leq \left(\mathbb{E}\left[V\left(\theta_0\right) - V(\theta^*)\right] + \frac{2}{\tilde{\sigma}} \max_{1 \leq k \leq n} \frac{\lambda_{k+1}v_k}{\beta_{k+1}^2\gamma_{k+1}}\right) \exp\left(-\frac{\mu}{2} \sum_{k=n/2}^{n} \lambda_{k+1}\gamma_{k+1}\right)$$

$$\times \exp\left(2\sum_{k=1}^{n} C_k\beta_{k+1}^2\gamma_{k+1}^2\right) + \frac{2}{\mu} \max_{n/2 \leq k \leq n} v_k \ ,$$

*where*

$$\tilde{\sigma} = \frac{\tilde{\sigma}_2 L}{2} + \tilde{\sigma}_1 L^2, \quad C_k = \max\left\{\tilde{\sigma}, \frac{\mu^2\lambda_{k+1}^2}{4\beta_{k+1}^2}\right\} \quad \text{and} \quad v_k = r_{k+1} + \frac{L\sigma_k^2}{2}\frac{\beta_{k+1}^2}{\lambda_{k+1}}\gamma_{k+1} \ .$$

*with the convention $C_k = 1$ if $\tilde{\sigma}_1 = \tilde{\sigma}_2 = 0$.*

*Proof.* Since $V$ is $L$-smooth (Assumption H2) and using the recursion (3) of Adaptive SA, we obtain:

$$V\left(\theta_{n+1}\right) \leq V\left(\theta_n\right) + \langle \nabla V\left(\theta_n\right), \theta_{n+1} - \theta_n \rangle + \frac{L}{2} \|\theta_{n+1} - \theta_n\|^2$$

$$\leq V\left(\theta_n\right) - \gamma_{n+1} \langle \nabla V\left(\theta_n\right), A_n H_{\theta_n}\left(X_{n+1}\right)\rangle + \frac{L\gamma_{n+1}^2}{2} \|A_n\|^2 \|H_{\theta_n}\left(X_{n+1}\right)\|^2 \ .$$

Writing $V_n = V\left(\theta_n\right) - V(\theta^*)$, we get

$$V_{n+1} \leq V_n - \gamma_{n+1} \langle \nabla V\left(\theta_n\right), A_n H_{\theta_n}\left(X_{n+1}\right)\rangle + \frac{L}{2}\gamma_{n+1}^2\beta_{n+1}^2 \|H_{\theta_n}\left(X_{n+1}\right)\|^2 \ .$$

Then, using H3,

$$\mathbb{E}\left[V_{n+1}\right] \leq \mathbb{E}\left[V_n\right] - \gamma_{n+1}\mathbb{E}\left[\langle \nabla V\left(\theta_n\right), A_n H_{\theta_n}\left(X_{n+1}\right)\rangle\right] + \frac{L}{2}\gamma_{n+1}^2\beta_{n+1}^2\sigma_n^2$$

$$+ \frac{L}{2}\gamma_{n+1}^2\beta_{n+1}^2\left(\tilde{\sigma}_1\mathbb{E}[\|\nabla V\left(\theta_n\right)\|^2] + \tilde{\sigma}_2\mathbb{E}[V_n]\right)$$

$$\leq \left(1 + \frac{\tilde{\sigma}_2 L}{2}\beta_{n+1}^2\gamma_{n+1}^2\right)\mathbb{E}[V_n] - \gamma_{n+1}\left(\lambda_{n+1} - \frac{\tilde{\sigma}_1 L}{2}\gamma_{n+1}\beta_{n+1}^2\right)\mathbb{E}[\|\nabla V\left(\theta_n\right)\|^2]$$

$$+ \gamma_{n+1}\lambda_{n+1}r_{n+1} + \frac{L\sigma_n^2}{2}\gamma_{n+1}^2\beta_{n+1}^2 \ .$$

Given the smoothness condition (5), with $\theta' = \theta_n$ and $\theta = \theta_n - \frac{1}{L}\nabla V(\theta_n)$, we derive:

$$V(\theta^*) \leq V(\theta_n) - \frac{1}{L}\|\nabla V(\theta_n)\|^2 + \frac{1}{2L}\|\nabla V(\theta_n)\|^2 ,$$

$$\|\nabla V(\theta_n)\|^2 \leq 2LV_n .$$

Using the above inequality and the Polyak-Łojasiewicz condition (H1), we obtain:

$$\mathbb{E}[V_{n+1}] \leq \left(1 - \mu\lambda_{n+1}\gamma_{n+1} + \left(\frac{\tilde{\sigma}_2 L}{2} + \tilde{\sigma}_1 L^2\right)\beta_{n+1}^2\gamma_{n+1}^2\right)\mathbb{E}[V_n]$$
$$+ \gamma_{n+1}\lambda_{n+1}r_{n+1} + \frac{L\sigma_n^2}{2}\gamma_{n+1}^2\beta_{n+1}^2 .$$

By choosing $\bar{\gamma}_{n+1} = \lambda_{n+1}\gamma_{n+1}$, we get:

$$\mathbb{E}[V_{n+1}] \leq \left(1 - \mu\bar{\gamma}_{n+1} + \left(\frac{\tilde{\sigma}_2 L}{2} + \tilde{\sigma}_1 L^2\right)\frac{\beta_{n+1}^2}{\lambda_{n+1}^2}\bar{\gamma}_{n+1}^2\right)\mathbb{E}[V_n]$$
$$+ \bar{\gamma}_{n+1}r_{n+1} + \frac{L\sigma_n^2}{2}\frac{\beta_{n+1}^2}{\lambda_{n+1}}\bar{\gamma}_{n+1}\gamma_{n+1} .$$

In order to satisfy the assumptions of Lemma A.1, consider

$$C_n = \max\left\{\tilde{\sigma}, \frac{\mu^2\lambda_{n+1}^2}{4\beta_{n+1}^2}\right\} \quad \text{with} \quad \tilde{\sigma} = \frac{\tilde{\sigma}_2 L}{2} + \tilde{\sigma}_1 L^2 .$$

Then, since $C_n \geq \tilde{\sigma}$, we have:

$$\mathbb{E}[V_{n+1}] \leq \left(1 - \mu\bar{\gamma}_{n+1} + \frac{C_n\beta_{n+1}^2}{\lambda_{n+1}^2}\bar{\gamma}_{n+1}^2\right)\mathbb{E}[V_n] + \bar{\gamma}_{n+1}r_{n+1} + \frac{L\sigma_n^2}{2}\frac{\beta_{n+1}^2}{\lambda_{n+1}}\bar{\gamma}_{n+1}\gamma_{n+1} .$$

Now, using lemma A.1 by choosing:

$$\delta_n = \mathbb{E}[V_n], \quad \eta_n = \frac{C_n\beta_{n+1}^2}{\lambda_{n+1}^2}\bar{\gamma}_{n+1}, \quad \omega = \frac{\mu}{2}, \quad v_n = r_{n+1} + \frac{L\sigma_n^2}{2}\frac{\beta_{n+1}^2}{\lambda_{n+1}}\gamma_{n+1} ,$$

we have:

$$\mathbb{E}[V(\theta_n) - V(\theta^*)] \leq \left(\mathbb{E}[V(\theta_0) - V(\theta^*)] + \frac{2}{\tilde{\sigma}}\max_{1 \leq k \leq n}\frac{v_k\lambda_{k+1}^2}{\beta_{k+1}^2\bar{\gamma}_{k+1}}\right)\exp\left(-\frac{\mu}{2}\sum_{k=n/2}^n\bar{\gamma}_{k+1}\right)$$

$$\times \exp\left(2\sum_{k=1}^n C_k\beta_{k+1}^2\bar{\gamma}_{k+1}^2/\lambda_{k+1}^2\right) + \frac{2}{\mu}\max_{n/2 \leq k \leq n}\{v_k\} ,$$

which concludes the proof by taking $\bar{\gamma}_{n+1} = \lambda_{n+1}\gamma_{n+1}$.  □

## A.2  Proof of Theorem 4.2

By H2, using (5), we obtain:

$$V(\theta_{k+1}) \leq V(\theta_k) + \langle\nabla V(\theta_k), \theta_{k+1} - \theta_k\rangle + \frac{L}{2}\|\theta_{k+1} - \theta_k\|^2 ,$$

which, using the recursion (3) of Adaptive SA and H4, yields:

$$V(\theta_{k+1}) \leq V(\theta_k) - \gamma_{k+1}\langle\nabla V(\theta_k), A_k H_{\theta_k}(X_{k+1})\rangle + \delta_{k+1}\|H_{\theta_k}(X_{k+1})\|^2 ,$$

with $\delta_{k+1} = L\gamma_{k+1}^2\beta_{k+1}^2/2$. Using Assumption H3, we have:

$$\mathbb{E}[V(\theta_{k+1})] \leq \mathbb{E}[V(\theta_k)] - \gamma_{k+1}\lambda_{k+1}\mathbb{E}[\|\nabla V(\theta_k)\|^2] + \gamma_{k+1}\lambda_{k+1}r_{k+1} + \delta_{k+1}\sigma_k^2$$
$$+ \delta_{k+1}\left(\tilde{\sigma}_1\mathbb{E}[\|\nabla V(\theta_k)\|^2] + \tilde{\sigma}_2\mathbb{E}[V(\theta_k) - V(\theta^*)]\right) .$$

Therefore,

$$\gamma_{k+1}\left(\lambda_{k+1} - \frac{L\tilde{\sigma}_1}{2}\gamma_{k+1}\beta_{k+1}^2\right)\mathbb{E}\left[\|\nabla V(\theta_k)\|^2\right]$$
$$\leq (1+\tilde{\sigma}_2\delta_{k+1})\left(\mathbb{E}[V(\theta_k)] - V(\theta^*)\right) - \left(\mathbb{E}[V(\theta_{k+1})] - V(\theta^*)\right)$$
$$+ \gamma_{k+1}\lambda_{k+1}r_{k+1} + \delta_{k+1}\sigma_k^2 .$$

Let us now consider the sequence of weights $w_k$ defined by $w_0 = 1$ and $w_k = \prod_{j=1}^k (1+\tilde{\sigma}_2\delta_j)^{-1}$. Then,

$$w_{k+1}\gamma_{k+1}\left(\lambda_{k+1} - \frac{L\tilde{\sigma}_1}{2}\gamma_{k+1}\beta_{k+1}^2\right)\mathbb{E}\left[\|\nabla V(\theta_k)\|^2\right]$$
$$\leq w_k\left(\mathbb{E}[V(\theta_k)] - V(\theta^*)\right) - w_{k+1}\left(\mathbb{E}[V(\theta_{k+1})] - V(\theta^*)\right)$$
$$+ w_{k+1}\gamma_{k+1}\lambda_{k+1}r_{k+1} + w_{k+1}\delta_{k+1}\sigma_k^2.$$

In the sequel, let us denote $V_n = V(\theta_n) - V(\theta^*)$, so that

$$\sum_{k=0}^n w_{k+1}\gamma_{k+1}\lambda_{k+1}\left(1 - \frac{L\tilde{\sigma}_1}{2\lambda_{k+1}}\gamma_{k+1}\beta_{k+1}^2\right)\mathbb{E}\left[\|\nabla V(\theta_k)\|^2\right]$$
$$\leq w_0\mathbb{E}[V_0] - w_{n+1}\mathbb{E}[V_{n+1}] + \frac{1}{2}\sum_{k=0}^n w_{k+1}\gamma_{k+1}\lambda_{k+1}r_{k+1} + \sum_{k=0}^n w_{k+1}\delta_{k+1}\sigma_k^2.$$

Then, given that $\gamma_{k+1} \leq \lambda_{k+1}/(L\tilde{\sigma}_1\beta_{k+1}^2)$, we have

$$\frac{1}{2}\mathbb{E}\left[\sum_{k=0}^n w_{k+1}\gamma_{k+1}\lambda_{k+1}\|\nabla V(\theta_k)\|^2\right] \leq w_0\mathbb{E}[V_0] - w_{n+1}\mathbb{E}[V_{n+1}]$$
$$+ \frac{1}{2}\sum_{k=0}^n w_{k+1}\gamma_{k+1}\lambda_{k+1}r_{k+1} + \sum_{k=0}^n w_{k+1}\delta_{k+1}\sigma_k^2 .$$

Consequently, by definition of the discrete random variable $R$,

$$\mathbb{E}\left[\|\nabla V(\theta_R)\|^2\right] = \sum_{k=0}^n \frac{w_{k+1}\gamma_{k+1}\lambda_{k+1}}{\sum_{j=0}^n w_{j+1}\gamma_{j+1}\lambda_{j+1}}\mathbb{E}\left[\|\nabla V(\theta_k)\|^2\right]$$
$$\leq 2\frac{\mathbb{E}[V_0] - w_{n+1}\mathbb{E}[V_{n+1}] + \sum_{k=0}^n w_{k+1}\gamma_{k+1}r_{k+1} + \sum_{k=0}^n w_{k+1}\delta_{k+1}\sigma_k^2}{\sum_{j=0}^n w_{j+1}\gamma_{j+1}\lambda_{j+1}},$$

which concludes the proof by noting that $V(\theta_{n+1}) \geq V(\theta^*)$.

### A.3 Proof of Corollary 4.3

The proof is a direct consequence of the fact that for a sufficiently large $n$:

$$\sum_{k=1}^n \frac{1}{k^s} = \begin{cases} \mathcal{O}\left(n^{-s+1}\right) & \text{if } 0 \leq s < 1 , \\ \mathcal{O}(1) & \text{if } s > 1 , \\ \mathcal{O}(\log n) & \text{if } s = 1 . \end{cases}$$

### A.4 Proof of Corollary 4.4

By H2, using (5), we obtain:

$$V(\theta_{k+1}) \leq V(\theta_k) + \langle \nabla V(\theta_k), \theta_{k+1} - \theta_k \rangle + \frac{L}{2}\|\theta_{k+1} - \theta_k\|^2$$
$$\leq V(\theta_k) - \gamma_{k+1}\langle \nabla V(\theta_k), A_k H_{\theta_k}(X_{k+1})\rangle + \frac{L\gamma_{k+1}^2}{2}\|A_k\|^2\|H_{\theta_k}(X_{k+1})\|^2 ,$$

which, using H4 and H5 yields:

$$V\left(\theta_{k+1}\right) \leq V\left(\theta_{k}\right) - \gamma_{k+1}\left\langle\nabla V\left(\theta_{k}\right), A_{k}H_{\theta_{k}}\left(X_{k+1}\right)\right\rangle + \frac{L}{2}\gamma_{k+1}^{2}\beta_{k+1}^{2}M^{2}.$$

Using H3,

$$\mathbb{E}[V(\theta_{k+1})|\mathcal{F}_{k}] \leq V(\theta_{k}) - \gamma_{k+1}\lambda_{n+1}\|\nabla V(\theta_{k})\|^{2} + \gamma_{k+1}\lambda_{k+1}r_{k+1} + \frac{LM^{2}}{2}\gamma_{k+1}^{2}\beta_{k+1}^{2}.$$

Therefore,

$$\gamma_{k+1}\lambda_{k+1}\|\nabla V\left(\theta_{k}\right)\|^{2} \leq V\left(\theta_{k}\right) - \mathbb{E}\left[V\left(\theta_{k+1}\right)|\mathcal{F}_{k}\right] + \gamma_{k+1}\lambda_{k+1}r_{k+1} + \frac{LM^{2}}{2}\gamma_{k+1}^{2}\beta_{k+1}^{2},$$

and

$$\sum_{k=0}^{n}\gamma_{k+1}\lambda_{k+1}\mathbb{E}\left[\|\nabla V\left(\theta_{k}\right)\|^{2}\right] \leq \mathbb{E}\left[V\left(\theta_{0}\right) - V\left(\theta_{n+1}\right)\right] + \sum_{k=0}^{n}\gamma_{k+1}\lambda_{k+1}r_{k+1}$$

$$+ \frac{LM^{2}}{2}\sum_{k=0}^{n}\gamma_{k+1}^{2}\beta_{k+1}^{2}.$$

Consequently, by definition of the discrete random variable $R$,

$$\mathbb{E}\left[\|\nabla V\left(\theta_{R}\right)\|^{2}\right] = \frac{1}{n}\sum_{k=0}^{n}\mathbb{E}\left[\|\nabla V\left(\theta_{k}\right)\|^{2}\right]$$

$$\leq \sum_{k=0}^{n}\frac{\gamma_{k+1}\lambda_{k+1}}{\sqrt{n}}\mathbb{E}\left[\|\nabla V\left(\theta_{k}\right)\|^{2}\right]$$

$$\leq \frac{V_{0,n} + \sum_{k=0}^{n}\gamma_{k+1}\lambda_{k+1}r_{k+1} + LM^{2}\sum_{k=0}^{n}\gamma_{k+1}^{2}\beta_{k+1}^{2}/2}{\sqrt{n}},$$

where $V_{0,n} = \mathbb{E}[V(\theta_{0}) - V(\theta_{n+1})]$, which conclude the proof by noting that $V(\theta_{n+1}) \geq V(\theta^{*})$.

## A.5 Proof of Corollary 4.5

Here, we consider the case where the regularization is non-increasing, i.e., where $\delta = \beta_{n+1}^{-2}$. The constant case is strictly analogous. To verify H4, we demonstrate that the control of the maximum and minimum eigenvalues is satisfied for Adagrad and RMSProp.

**Adagrad**

- **Lower bound for the smallest eigenvalue of $\mathbf{A_n}$.** By assumption H5, we have:

$$\left\|\frac{1}{n+1}\sum_{k=0}^{n}H_{\theta_{k}}(X_{k+1})H_{\theta_{k}}(X_{k+1})^{\top}\right\| \leq M^{2}.$$

  This implies that:

$$\lambda_{\min}(A_{n}) = \lambda_{\max}\left(\beta_{n+1}^{-2}I_{d} + \text{Diag}\left(\frac{1}{n+1}\sum_{k=0}^{n}H_{\theta_{k}}(X_{k+1})H_{\theta_{k}}(X_{k+1})^{\top}\right)\right)^{-1/2}$$

$$\geq (\beta_{1}^{-2} + M^{2})^{-1/2}.$$

- **Upper bound for the largest eigenvalue of $\mathbf{A_n}$.**

$$\lambda_{\max}(A_{n}) = \lambda_{\min}\left(\beta_{n+1}^{-2}I_{d} + \text{Diag}\left(\frac{1}{n+1}\sum_{k=0}^{n}H_{\theta_{k}}(X_{k+1})H_{\theta_{k}}(X_{k+1})^{\top}\right)\right)^{-1/2} \leq \beta_{n+1}.$$

Therefore, by setting $\lambda_{n+1} = (\beta_1^{-2} + M^2)^{-1/2}$ and $\beta_n = C_\beta n^\beta$, we have $\lambda = 0$ and one can arbitrarily choose $\beta$ (one can take $\beta = 0$ for the constant regularization case).

**RMSProp**

- **Lower bound for the smallest eigenvalue of $\mathbf{A_n}$.** By assumption H5, we have:

$$\|V_n\| \leq (1-\rho) \sum_{k=1}^n \rho^{n-k} \|H_{\theta_k}(X_{k+1})\|^2 \leq M^2(1-\rho) \sum_{k=1}^n \rho^{n-k} \leq M^2 \ ,$$

where we used the fact that $\sum_{k=1}^n \rho^{n-k} \leq (1-\rho)^{-1}$. This implies that:

$$\lambda_{\min}(A_n) = \lambda_{\max}\left(\beta_{n+1}^{-2} I_d + \mathrm{Diag}\left(V_n\right)\right)^{-1/2} \geq (\beta_1^{-2} + M^2)^{-1/2} \ .$$

- **Upper bound for the largest eigenvalue of $\mathbf{A_n}$.** Note that

$$\lambda_{\max}(A_n) = \lambda_{\min}\left(\beta_{n+1}^{-2} I_d + \mathrm{Diag}\left(V_n\right)\right)^{-1/2} \leq \beta_{n+1} \ .$$

Therefore, under H2, H3(i), and H5, we can conclude that

$$\mathbb{E}\left[\|\nabla V\left(\theta_R\right)\|^2\right] = \mathcal{O}\left(\frac{\log n}{\sqrt{n}} + b_n\right) \ ,$$

where $b_n$ corresponds to the bias which comes from $r_n$ in H3(i). Choosing $r_n = C_r n^{-r}$, we get:

$$b_n = \begin{cases} \mathcal{O}\left(n^{-r}\right) & \text{if } r < 1/2 \ , \\ \mathcal{O}\left(n^{-1/2}\right) & \text{if } r > 1/2 \ , \\ \mathcal{O}\left(n^{-1/2} \log n\right) & \text{if } r = 1/2 \ . \end{cases}$$

Now, we show that under the control of bias, i.e.,

$$\|\mathbb{E}\left[H_{\theta_n}\left(X_{n+1}\right) | \mathcal{F}_n\right] - \nabla V\left(\theta_n\right)\| \leq C_\alpha n^{-\alpha} \ ,$$

we can verify H3(i) with a similar bound on the bias, where $r = 2\alpha$. This yields the bias term $b_n$ as follows:

$$b_n = \begin{cases} \mathcal{O}\left(n^{-2\alpha}\right) & \text{if } \alpha < 1/4 \ , \\ \mathcal{O}\left(n^{-1/2}\right) & \text{if } \alpha > 1/4 \ , \\ \mathcal{O}\left(n^{-1/2} \log n\right) & \text{if } \alpha = 1/4 \ . \end{cases}$$

**Verifying Assumption H3 (i) for Adagrad.** Using the tower property, we have:

$$\mathbb{E}\left[\langle \nabla V\left(\theta_n\right), A_n H_{\theta_n}\left(X_{n+1}\right)\rangle\right] = \mathbb{E}\left[\mathbb{E}\left[\langle \nabla V\left(\theta_n\right), A_n H_{\theta_n}\left(X_{n+1}\right)\rangle | \mathcal{F}_n\right]\right] ,$$

where $(\mathcal{F}_n)_{n\geq 0}$ represents the filtration generated by the random variables $(\theta_0, \{X_k\}_{k\leq n})$. Let $\tilde{A}_n$ be an adaptive $\mathcal{F}_n$-measurable matrix. Then,

$$\mathbb{E}\left[\langle \nabla V\left(\theta_n\right), A_n H_{\theta_n}\left(X_{n+1}\right)\rangle | \mathcal{F}_n\right] = \underbrace{\left\langle \nabla V\left(\theta_n\right), \tilde{A}_n \mathbb{E}\left[H_{\theta_n}\left(X_{n+1}\right) | \mathcal{F}_n\right]\right\rangle}_{\text{Treated as in SGD but with } \lambda_{\min}(\tilde{A}_n)}$$

$$+ \underbrace{\mathbb{E}\left[\left\langle \nabla V\left(\theta_n\right), (A_n - \tilde{A}_n) H_{\theta_n}\left(X_{n+1}\right)\right\rangle | \mathcal{F}_n\right]}_{\text{Control error between } A_n \text{ and } \tilde{A}_n} .$$

We only verify Assumption H3(i) for Adagrad algorithm since it is analogous to RMSProp. Consider $A_n$ given by:

$$A_n = \left(\mathrm{diag}\left(\beta_{n+1}^{-2} I_d + \frac{1}{n+1} \sum_{k=0}^n H_{\theta_k}\left(X_{k+1}\right) H_{\theta_k}\left(X_{k+1}\right)^\top\right)\right)^{-1/2} .$$

First, writing

$$\tilde{A}_n = \left(\mathrm{diag}\left(\beta_{n+1}^{-2} I_d + \frac{1}{n+1} \sum_{k=0}^{n-1} H_{\theta_k}\left(X_{k+1}\right) H_{\theta_k}\left(X_{k+1}\right)^\top\right)\right)^{-1/2}$$

and denoting by $A[i]$ the i-th element of the diagonal of a matrix $A$, we have

$$A_n[i] - \tilde{A}_n[i] = u_n^{-1/2} \left( v_n^{1/2} - u_n^{1/2} \right) v_n^{-1/2} \leq 0 \ ,$$

where

$$u_n = \beta_{n+1}^{-2} + \frac{1}{n+1} \sum_{k=0}^{n} \left( H_{\theta_k} \left( X_{k+1} \right) [i] \right)^2 \quad \text{and} \quad v_n = \beta_{n+1}^{-2} + \frac{1}{n+1} \sum_{k=0}^{n-1} \left( H_{\theta_k} \left( X_{k+1} \right) [i] \right)^2 \ .$$

Then, since $u_n \geq v_n$,

$$A_n[i] - \tilde{A}_n[i] = \frac{v_n - u_n}{\sqrt{u_n v_n} \left( \sqrt{u_n} + \sqrt{v_n} \right)} \geq -\frac{1}{n+1} \left( H_{\theta_n} \left( X_{n+1} \right) [i] \right)^2 \frac{1}{2 v_n^{3/2}}$$

$$\geq -\frac{\beta_{n+1}^3}{n+1} \left( H_{\theta_n} \left( X_{n+1} \right) [i] \right)^2 \ .$$

Since the bias of $H_{\theta_n} \left( X_{n+1} \right)$ is bounded by $\tilde{b}_n := C_\alpha n^{-\alpha}$,

$$\mathbb{E} \left[ \langle \nabla V \left( \theta_n \right), A_n H_{\theta_n} \left( X_{n+1} \right) \rangle | \mathcal{F}_n \right]$$

$$= \left\langle \nabla V \left( \theta_n \right), \tilde{A}_n \mathbb{E} \left[ H_{\theta_n} \left( X_{n+1} \right) | \mathcal{F}_n \right] \right\rangle + \mathbb{E} \left[ \left\langle \nabla V \left( \theta_n \right), (A_n - \tilde{A}_n) H_{\theta_n} \left( X_{n+1} \right) \right\rangle | \mathcal{F}_n \right]$$

$$\geq \lambda_{\min} \left( \tilde{A}_n \right) \| \nabla V \left( \theta_n \right) \|^2 - \lambda_{\max} \left( \tilde{A}_n \right) \| \nabla V \left( \theta_n \right) \| \tilde{b}_n$$

$$- \| \nabla V \left( \theta_n \right) \| \frac{\beta_{n+1}^3}{n+1} \mathbb{E} \left[ \| H_{\theta_n} \left( X_{n+1} \right) \|^3 | \mathcal{F}_n \right] \ .$$

As $H_{\theta_n} \left( X_{n+1} \right)$ and the gradient of $V$ are uniformly bounded by $M$, $\lambda_{\min}(\tilde{A}_n) \geq (\beta_1^{-2} + M^2)^{-1/2}$, so that

$$\mathbb{E} \left[ \langle \nabla V \left( \theta_n \right), A_n H_{\theta_n} \left( X_{n+1} \right) \rangle | \mathcal{F}_n \right] \geq \frac{1}{\sqrt{\beta_1^{-2} + M^2}} \| \nabla V \left( \theta_n \right) \|^2 - \beta_{n+1} M \tilde{b}_n - M^4 \frac{\beta_{n+1}^3}{n+1} \ ,$$

and Assumption H3(i) is satisfied with $\lambda_{n+1} = (\beta_1^{-2} + M^2)^{-1/2}$ and $r_{n+1} = M \beta_{n+1}^2 \tilde{b}_n^2 / \lambda_{n+1} + M^4 \beta_{n+1}^3 / (n+1)$.

## A.6 Proof of Theorem 4.6

The proof of this theorem is inspired by [67] and [73], considering biased gradient estimators and decreasing step sizes. We define the operation $\max(D_1, D_2)$ for diagonal matrices $D_1$ and $D_2$ as the matrix formed by taking the maximum between the diagonal elements of $D_1$ and $D_2$. We say that the sequence $(A_n)_{n \geq 1}$ of diagonal matrices is decreasing if all diagonal terms are decreasing, in other words, if all eigenvalues are decreasing.

Let $\tilde{\theta}_{k+1} = \theta_{k+1} + \kappa (\theta_{k+1} - \theta_k)$, for $k \geq 1, \kappa \in [0, 1)$ and $m_k = \rho_1 m_{k-1} + (1 - \rho_1) g_k$ with $g_k = H_{\theta_k}(X_{k+1})$. Using the recursion of AMSGRAD, we have:

$$\tilde{\theta}_{k+1} - \tilde{\theta}_k = (1 + \kappa)\theta_{k+1} - (1 + 2\kappa)\theta_k + \kappa\theta_{k-1} = (1 + \kappa) \left( \theta_{k+1} - \theta_k \right) - \kappa \left( \theta_k - \theta_{k-1} \right)$$

$$= -(1 + \kappa)\gamma_{k+1} A_k m_k + \kappa\gamma_k A_{k-1} m_{k-1} \ .$$

Choosing $\kappa = \rho_1 / (1 - \rho_1)$, we can rewrite it as:

$$\tilde{\theta}_{k+1} - \tilde{\theta}_k = \kappa \left( \gamma_k A_{k-1} - \gamma_{k+1} A_k \right) m_{k-1} - \gamma_{k+1} A_k g_k \ .$$

By Assumption H2, $V$ is $L$-smooth, using the recursion of AMSGRAD together with a Taylor expansion with $\tilde{\theta}_k$, we obtain:

$$V(\tilde{\theta}_{k+1}) \leq V \left( \tilde{\theta}_k \right) + \left\langle \nabla V \left( \tilde{\theta}_k \right), \tilde{\theta}_{k+1} - \tilde{\theta}_k \right\rangle + \frac{L}{2} \left\| \tilde{\theta}_{k+1} - \tilde{\theta}_k \right\|^2$$

$$\leq V \left( \tilde{\theta}_k \right) - \gamma_{k+1} \left\langle \nabla V \left( \tilde{\theta}_k \right), A_k g_k \right\rangle + \kappa \left\langle \nabla V \left( \tilde{\theta}_k \right), (\gamma_k A_{k-1} - \gamma_{k+1} A_k) m_{k-1} \right\rangle$$

$$+ L\gamma_{k+1}^2 \| A_k g_k \|^2 + L\kappa^2 \| (\gamma_k A_{k-1} - \gamma_{k+1} A_k) m_{k-1} \|^2$$

$$\leq V \left( \tilde{\theta}_k \right) + T_{1,k} + T_{2,k} + T_{3,k} + T_{4,k} \ ,$$

where

$$T_{1,k} = -\gamma_{k+1} \langle \nabla V(\theta_k), A_k g_k \rangle + L \gamma_{k+1}^2 \|A_k g_k\|^2,$$

$$T_{2,k} = -\gamma_{k+1} \left\langle \nabla V\left(\tilde{\theta}_k\right) - \nabla V(\theta_k), A_k g_k \right\rangle,$$

$$T_{3,k} = \kappa \left\langle \nabla V\left(\tilde{\theta}_k\right), (\gamma_k A_{k-1} - \gamma_{k+1} A_k) m_{k-1} \right\rangle,$$

$$T_{4,k} = L\kappa^2 \|(\gamma_k A_{k-1} - \gamma_{k+1} A_k) m_{k-1}\|^2.$$

Note first that

$$\sum_{k=1}^n \mathbb{E}[T_{1,k}] = -\sum_{k=1}^n \gamma_{k+1} \mathbb{E}[\langle \nabla V(\theta_k), A_k g_k \rangle] + L \sum_{k=1}^n \gamma_{k+1}^2 \mathbb{E}\left[\|A_k g_k\|^2\right]$$

$$\leq -C_\lambda \sum_{k=1}^n \gamma_{k+1} \mathbb{E}\left[\|\nabla V(\theta_k)\|^2\right] + C_\lambda \sum_{k=1}^n \gamma_{k+1} r_{k+1} + L \sum_{k=1}^n \gamma_{k+1}^2 \mathbb{E}\left[\|A_k g_k\|^2\right],$$

where $C_\lambda = (\delta + M^2)^{-1/2}$.

For the second term, using the inequality $xy \leq x^2/2 + y^2/2$ for all $x, y$, and the smoothness of $V$, we get:

$$\sum_{k=1}^n \mathbb{E}[T_{2,k}] = -\sum_{k=1}^n \mathbb{E}\left[\left\langle \nabla V\left(\tilde{\theta}_k\right) - \nabla V(\theta_k), \gamma_{k+1} A_k g_k \right\rangle\right]$$

$$\leq \frac{1}{2} \sum_{k=1}^n \mathbb{E}\left[\left\|\nabla V\left(\tilde{\theta}_k\right) - \nabla V(\theta_k)\right\|^2\right] + \frac{1}{2} \sum_{k=1}^n \mathbb{E}\left[\|\gamma_{k+1} A_k g_k\|^2\right]$$

$$\leq \frac{L^2}{2} \sum_{k=1}^n \mathbb{E}\left[\left\|\tilde{\theta}_k - \theta_k\right\|^2\right] + \sum_{k=1}^n \frac{\gamma_{k+1}^2}{2} \mathbb{E}\left[\|A_k g_k\|^2\right]$$

$$\leq \frac{\kappa^2 L^2}{2} \sum_{k=1}^n \mathbb{E}\left[\|\theta_k - \theta_{k-1}\|^2\right] + \sum_{k=1}^n \frac{\gamma_{k+1}^2}{2} \mathbb{E}\left[\|A_k g_k\|^2\right]$$

$$\leq \frac{\kappa^2 L^2}{2} \sum_{k=1}^n \gamma_k^2 \mathbb{E}\left[\|A_{k-1} m_{k-1}\|^2\right] + \sum_{k=1}^n \frac{\gamma_{k+1}^2}{2} \mathbb{E}\left[\|A_k g_k\|^2\right].$$

For the third term, using the boundedness of the gradient of $V$ and the fact that $\|m_k\| \leq M$ by Lemma A.3, we have:

$$\sum_{k=1}^n \mathbb{E}[T_{3,k}] = \kappa \sum_{k=1}^n \mathbb{E}\left[\left\langle \nabla V\left(\tilde{\theta}_k\right), (\gamma_k A_{k-1} - \gamma_{k+1} A_k) m_{k-1} \right\rangle\right]$$

$$\leq \kappa M^2 \sum_{i=1}^d \sum_{k=1}^n \mathbb{E}[\gamma_k A_{k-1}[i] - \gamma_{k+1} A_k[i]]$$

$$\leq \kappa M^2 \sum_{i=1}^d \mathbb{E}[\gamma_1 A_0[i] - \gamma_{n+1} A_n[i]] \leq \kappa M^2 d C_\gamma,$$

where in the second inequality, we used the fact that $\gamma_k$ and $A_k$ are decreasing since we use $\hat{V}_k = \max(\hat{V}_{k-1}, V_k)$. For the last term, using the boundedness of the gradient of $V$ yields:

$$\sum_{k=1}^n \mathbb{E}[T_{4,k}] = L\kappa^2 \sum_{k=1}^n \mathbb{E}\left[\|(\gamma_k A_{k-1} - \gamma_{k+1} A_k) m_{k-1}\|^2\right]$$

$$\leq L\kappa^2 M^2 \sum_{i=1}^d \sum_{k=1}^n \mathbb{E}\left[(\gamma_k A_{k-1}[i] - \gamma_{k+1} A_k[i])^2\right]$$

$$\leq L\kappa^2 M^2 \sum_{i=1}^d \sum_{k=1}^n \mathbb{E}\left[(\gamma_k A_{k-1}[i])^2 - (\gamma_{k+1} A_k[i])^2\right]$$

$$\leq L\kappa^2 M^2 d C_\gamma^2,$$

where we used the inequality $(x - y)^2 \leq x^2 - y^2$ when $x \geq y$ in the second last inequality. Combining all these terms, we finally obtain:

$$C_\lambda \sum_{k=1}^{n} \gamma_{k+1} \mathbb{E}\left[\|\nabla V(\theta_k)\|^2\right]$$

$$\leq V^* + C_\lambda \sum_{k=1}^{n} \gamma_{k+1} r_{k+1} + L \sum_{k=1}^{n} \gamma_{k+1}^2 \mathbb{E}\left[\|A_k g_k\|^2\right] + \sum_{k=1}^{n} \frac{\gamma_{k+1}^2}{2} \mathbb{E}\left[\|A_k g_k\|^2\right]$$

$$+ \frac{\kappa^2 L^2}{2} \sum_{k=1}^{n} \gamma_k^2 \mathbb{E}\left[\|A_{k-1} m_{k-1}\|^2\right] + \kappa M^2 dC_\gamma + L\kappa^2 M^2 dC_\gamma^2 ,$$

where $V^* = \mathbb{E}[V(\theta_0) - V(\theta^*)] \geq \mathbb{E}[V(\theta_0) - V(\tilde{\theta}_{n+1})]$. Choosing $\gamma_n = n^{-1/2}$ and using Lemma A.3 and [15, Lemma 24] yields

$$\sum_{k=1}^{n} \gamma_{k+1}^2 \mathbb{E}\left[\|A_k m_k\|^2\right] \leq (1 - \rho_1) \sum_{k=1}^{n} \gamma_{k+1}^2 \mathbb{E}\left[\|A_k g_k\|^2\right] \leq (1 - \rho_1) dC_\gamma^2 \log\left(1 + \frac{nM^2}{\delta}\right)$$

$$= \mathcal{O}\left(d \log n\right).$$

Therefore, by dividing both sides by $C_\lambda n^{-1/2}$, we obtain

$$\frac{1}{n} \sum_{k=1}^{n} \mathbb{E}\left[\|\nabla V(\theta_k)\|^2\right] = \mathcal{O}\left(\frac{1}{\sqrt{n}} + \frac{d \log n}{\sqrt{n}} + \frac{d}{\sqrt{n}} + b_n\right),$$

which concludes the proof.

**Lemma A.3.** *Let $\gamma_{k+1} \leq \gamma_k$ for all $k \in \mathbb{N}$, and let $A_k$ be the adaptive matrix defined in Algorithm 1. Assume that $\rho_1 \in [0, 1)$. Then, for all $k \in \mathbb{N}$:*

$$\|m_k\| \leq M \quad \text{and} \quad \sum_{k=1}^{n} \gamma_{k+1}^2 \mathbb{E}\left[\|A_k m_k\|^2\right] \leq (1 - \rho_1) \sum_{k=1}^{n} \gamma_{k+1}^2 \|A_k g_k\|^2 .$$

*Proof.* For the first inequality, we have:

$$\|m_k\| = \left\|(1 - \rho_1) \sum_{\ell=1}^{k} \rho_1^{k-\ell} g_\ell\right\| \leq (1 - \rho_1) \sum_{\ell=1}^{k} \rho_1^{k-\ell} \|g_\ell\| \leq M(1 - \rho_1) \sum_{\ell \geq 0} \rho_1^\ell \leq M ,$$

where we used the fact that $\sum_{\ell \geq 0} \rho_1^\ell = 1/(1 - \rho_1)$. For the second inequality, using the fact that $\gamma_k$ and $A_k$ are decreasing (in the sense that all eigenvalues of $A_k$ are decreasing), since we use $\hat{V}_k = \max(\hat{V}_{k-1}, V_k)$, we can write:

$$\sum_{k=1}^{n} \gamma_{k+1}^2 \|A_k m_k\|^2 = \sum_{k=1}^{n} \gamma_{k+1}^2 \left\|A_k(1 - \rho_1) \sum_{\ell=1}^{k} \rho_1^{k-\ell} g_\ell\right\|^2$$

$$\leq (1 - \rho_1)^2 \sum_{k=1}^{n} \gamma_{k+1}^2 \sum_{\ell=1}^{k} \rho_1^{k-\ell} \|A_\ell g_\ell\|^2$$

$$\leq (1 - \rho_1)^2 \sum_{k=1}^{n} \sum_{\ell=1}^{k} \rho_1^{k-\ell} \gamma_{\ell+1}^2 \|A_\ell g_\ell\|^2$$

$$\leq (1 - \rho_1)^2 \sum_{\ell=1}^{n} \sum_{k=\ell}^{n} \rho_1^{k-\ell} \gamma_{\ell+1}^2 \|A_\ell g_\ell\|^2 ,$$

which concludes the proof. $\qquad\square$

## A.7 The Impact of regularization parameter $\delta$ in Adam

In our case, we have a dependence on $\delta$ in the logarithm, which is common for adaptive algorithms. The regularization parameter $\delta$, originally introduced to avoid the zero denominator issue when $V_k$ approaches $0$, is often overlooked. However, it has been empirically observed that the performance of adaptive methods can be sensitive to the choice of this parameter, especially when a very small $\delta$ is used, which has resulted in performance issues in some applications.

In practice, $\delta$ is typically chosen as $10^{-8}$. In our convergence rate analysis, even though the logarithm of $\delta^{-1}$ is small, it still impacts the convergence rate. A larger $\delta$ will lead to a better convergence rate, while a smaller $\delta$ will preserve stronger adaptivity. We need to find a better compromise between the convergence rate and the adaptivity to choose $\delta$. In [77, 67, 73], it was shown that by choosing $\delta$ between $10^{-3}$ and $10^{-1}$, better results were obtained in some applications of deep learning.

Furthermore, several modified versions of Adam have been proposed, such as AMSGRAD [77] and YOGI [67] with the discussion of the regularization parameter $\delta$. The authors of [73] proposed a new modified version of Adam called SADAM to represent the calibrated ADAM using the softplus function. In this algorithm, they define $\hat{V}_k = \text{softplus}\left(\sqrt{V_k}\right)$ while other terms remain unchanged. Since we have:

$$\hat{V}_k = \text{softplus}\left(\sqrt{V_k}\right) = \frac{1}{b}\log\left(1 + e^{b\sqrt{V_k}}\right) \approx \frac{1}{b}\log\left(e^{b\sqrt{V_k}}\right) = \sqrt{V_k}\,,$$

where $b$ is the parameter to control for achieving a better convergence rate. In this case, we have $\lambda_{\max}(A_k) \leq b/\log 2$, which is similar to $\delta^{-1/2}$ in Adagrad and Adam. Additionally, they demonstrate that $b \approx 50$ appears to be a good choice based on the empirical observations.

## B IWAE / BR-IWAE

### B.1 Importance Weighted Autoencoder (IWAE)

In this section, we elaborate on the IWAE procedure within our framework to illustrate its convergence rate. The IWAE objective function is defined as:

$$\mathcal{L}_k^{\text{IWAE}}(\theta, \phi; x) = \mathbb{E}_{q_\phi^{\otimes k}(\cdot|x)} \left[ \log \frac{1}{k} \sum_{\ell=1}^{k} \frac{p_\theta(x, Z^{(\ell)})}{q_\phi(Z^{(\ell)} \mid x)} \right],$$

where $k$ corresponds to the number of samples drawn from the encoder's approximate posterior distribution. Denoting $V$ as the objective function, i.e., $V(\theta) = \log p_\theta(x)$, the gradient of $V$ and the estimator of the gradient of the ELBO of the IWAE objective are given by:

$$\nabla_\theta V(\theta) = \nabla_\theta \log p_\theta(x) = \mathbb{E}_{p_\theta(\cdot|x)} \left[ \nabla_\theta \log p_\theta(x, z) \right],$$

$$\widehat{\nabla}_\theta \mathcal{L}_k^{\text{IWAE}}(\theta, \phi; x) = \sum_{\ell=1}^{k} \frac{w^{(\ell)}}{\sum_{\ell=1}^{k} w^{(\ell)}} \nabla_\theta \log p_\theta(x, z^{(\ell)}), \tag{10}$$

where $w^{(\ell)} = p_\theta(x, z^{(\ell)})/q_\phi(z^{(\ell)}|x)$ the unnormalized importance weights. Theorem B.1 provides an upper bound for the bias of this estimator.

**Theorem B.1.** *Let* $\mathsf{X} \subseteq \mathbb{R}^{d_x}$ *and* $\mathsf{Z} \subseteq \mathbb{R}^{d_z}$ *denote the data space and the latent space, respectively. Assume that there exists $M$ such that for all $\theta \in \Theta \subset \mathbb{R}^d$, $x \in \mathsf{X}$ and $z \in \mathsf{Z}$, $\|\nabla_\theta \log p_\theta(x, z)\| \leq M(x)$. Then, there exists a constant $C > 0$ such that for all $\theta \in \Theta$, $\phi \in \Phi$ and $x \in \mathsf{X}$,*

$$\left\| \mathbb{E}_{q_\phi^{\otimes k}(\cdot|x)} \left[ \widehat{\nabla}_\theta \mathcal{L}_k^{IWAE}(\theta, \phi; x) - \nabla_\theta V(\theta) \right] \right\| \leq \frac{C}{k},$$

*where $\nabla_\theta V(\theta)$ and $\widehat{\nabla}_\theta \mathcal{L}_k^{IWAE}(\theta, \phi; x)$ are defined in* (10).

*Proof.* The proof is adapted from [1, Theorem 2.1]. By definition,

$$\widehat{\nabla}_\theta \mathcal{L}_k^{\text{IWAE}}(\theta, \phi; x) - \nabla_\theta V(\theta) = \frac{\sum_{\ell=1}^{k} w^{(\ell)} \left( \nabla_\theta \log p_\theta(x, z^{(\ell)}) - \mathbb{E}_{p_\theta(\cdot|x)} \left[ \nabla_\theta \log p_\theta(x, z) \right] \right)}{\sum_{\ell=1}^{k} w^{(\ell)}}.$$

Writing $\tilde{H}(x, z^{(\ell)}) = \nabla_\theta \log p_\theta(x, z^{(\ell)}) - \mathbb{E}_{p_\theta(\cdot|x)} \left[ \nabla_\theta \log p_\theta(x, z) \right]$, yields

$$\hat{\nabla}_\theta \mathcal{L}_k^{\text{IWAE}}(\theta, \phi; x) - \nabla_\theta V(\theta) = \frac{\sum_{\ell=1}^{k} w^{(\ell)} \tilde{H}(x, z^{(\ell)})}{\sum_{\ell=1}^{k} w^{(\ell)}}.$$

Since $\mathbb{E}_{q_\phi}[w\tilde{H}(x, z)] = 0$, we have:

$$\widehat{\nabla}_\theta \mathcal{L}_k^{\text{IWAE}}(\theta, \phi; x) - \nabla_\theta V(\theta) = \frac{\frac{1}{k} \sum_{\ell=1}^{k} w^{(\ell)} \tilde{H}(x, z^{(\ell)}) - \mathbb{E}_{q_\phi} \left[ w\tilde{H}(x, z) \right]}{\frac{1}{k} \sum_{\ell=1}^{k} w^{(\ell)}}.$$

As $\sum_{\ell=1}^{k} w^{(\ell)} \tilde{H}(x, z^{(\ell)})/k$ is an unbiased estimator of $\mathbb{E}_{q_\phi} \left[ w\tilde{H}(x, z) \right]$,

$$\mathbb{E}_{q_\phi} \left[ \hat{\nabla}_\theta \mathcal{L}_k^{\text{IWAE}}(\theta, \phi; x) - \nabla_\theta V(\theta) \right]$$

$$= \mathbb{E}_{q_\phi} \left[ \left( \frac{1}{\frac{1}{k} \sum_{\ell=1}^{k} w^{(\ell)}} - \frac{1}{\mathbb{E}_{q_\phi}[w]} \right) \left( \frac{1}{k} \sum_{\ell=1}^{k} w^{(\ell)} \tilde{H}(x, z^{(\ell)}) - \mathbb{E}_{q_\phi} \left[ w\tilde{H}(x, z) \right] \right) \right],$$

so that

$$\mathbb{E}_{q_\phi} \left[ \hat{\nabla}_\theta \mathcal{L}_k^{\text{IWAE}}(\theta, \phi; x) - \nabla_\theta V(\theta) \right]$$

$$= \mathbb{E}_{q_\phi} \left[ \frac{\left( \frac{1}{k} \sum_{\ell=1}^{k} w^{(\ell)} \tilde{H}(x, z^{(\ell)}) - \mathbb{E}_{q_\phi} \left[ w\tilde{H}(x, z) \right] \right) \left( \mathbb{E}_{q_\phi}[w] - \frac{1}{k} \sum_{\ell=1}^{k} w^{(\ell)} \right)}{\mathbb{E}_{q_\phi}[w] \frac{1}{k} \sum_{\ell=1}^{k} w^{(\ell)}} \right].$$

Therefore,

$$\left\| \mathbb{E}_{q_\phi} \left[ \hat{\nabla}_\theta \mathcal{L}_k^{\text{IWAE}}(\theta, \phi; x) - \nabla_\theta V(\theta) \right] \right\| \leq A_1 + A_2,$$

where

$$A_1 = \left\| \mathbb{E}_{q_\phi} \left[ \left( \hat{\nabla}_\theta \mathcal{L}_k^{\text{IWAE}}(\theta, \phi; x) - \nabla_\theta V(\theta) \right) \mathbb{1}_{\left\{ \frac{2}{k} \sum_{\ell=1}^k w^{(\ell)} > \mathbb{E}_{q_\phi}[w] \right\}} \right] \right\|,$$

$$A_2 = \left\| \mathbb{E}_{q_\phi} \left[ \left( \hat{\nabla}_\theta \mathcal{L}_k^{\text{IWAE}}(\theta, \phi; x) - \nabla_\theta V(\theta) \right) \mathbb{1}_{\left\{ \frac{2}{k} \sum_{\ell=1}^k w^{(\ell)} \leq \mathbb{E}_{q_\phi}[w] \right\}} \right] \right\|.$$

Note that

$$A_1 \leq \left\| \mathbb{E}_{q_\phi} \left[ \frac{2}{\mathbb{E}_{q_\phi}[w]^2} \left( \frac{1}{k} \sum_{\ell=1}^k w^{(\ell)} \tilde{H}(x, z^{(\ell)}) - \mathbb{E}_{q_\phi} \left[ w \tilde{H}(x, z) \right] \right) \left( \mathbb{E}_{q_\phi}[w] - \frac{1}{k} \sum_{\ell=1}^k w^{(\ell)} \right) \right] \right\|$$

$$\leq \frac{2}{\mathbb{E}_{q_\phi}[w]^2} \mathbb{E}_{q_\phi} \left[ \left\| \frac{1}{k} \sum_{\ell=1}^k w^{(\ell)} \tilde{H}(x, z^{(\ell)}) - \mathbb{E}_{q_\phi} \left[ w \tilde{H}(x, z) \right] \right\| \left\| \frac{1}{k} \sum_{\ell=1}^k w^{(\ell)} - \mathbb{E}_{q_\phi}[w] \right\| \right]$$

$$\leq \frac{2}{\mathbb{E}_{q_\phi}[w]^2} \mathbb{E}_{q_\phi} \left[ \left\| \frac{1}{k} \sum_{\ell=1}^k w^{(\ell)} \tilde{H}(x, z^{(\ell)}) - \mathbb{E}_{q_\phi} \left[ w \tilde{H}(x, z) \right] \right\|^2 \right]^{1/2}$$

$$\times \mathbb{E}_{q_\phi} \left[ \left( \frac{1}{k} \sum_{\ell=1}^k w^{(\ell)} - \mathbb{E}_{q_\phi}[w] \right)^2 \right]^{1/2},$$

where we used Cauchy-Schwarz inequality in the last inequality. On the other hand,

$$\mathbb{E}_{q_\phi} \left[ \left( \frac{1}{k} \sum_{\ell=1}^k w^{(\ell)} - \mathbb{E}_{q_\phi}[w] \right)^2 \right] = \mathbb{V} \left( \frac{1}{k} \sum_{\ell=1}^k w^{(\ell)} \right) \leq \frac{\mathbb{E}_{q_\phi}[w^2]}{k},$$

and

$$\mathbb{E}_{q_\phi} \left[ \left\| \frac{1}{k} \sum_{\ell=1}^k w^{(\ell)} \tilde{H}(x, z^{(\ell)}) - \mathbb{E}_{q_\phi} \left[ w \tilde{H}(x, z) \right] \right\|^2 \right]$$

$$= \text{Tr} \left( \mathbb{V} \left( \frac{1}{k} \sum_{\ell=1}^k w^{(\ell)} \tilde{H}(x, z^{(\ell)}) \right) \right) \leq 4dM^2 \frac{\mathbb{E}_{q_\phi}[w^2]}{k}.$$

Finally, we deduce that

$$A_1 \leq \frac{2}{\mathbb{E}_{q_\phi}[w]^2} \frac{1}{\sqrt{k}} \mathbb{E}_{q_\phi}[w^2]^{1/2} \frac{2\sqrt{d}M}{\sqrt{k}} \mathbb{E}_{q_\phi}[w^2]^{1/2} = \frac{\mathbb{E}_{q_\phi}[w^2]}{\mathbb{E}_{q_\phi}[w]^2} \frac{4\sqrt{d}M}{k}.$$

Using the assumption on the boundedness of $\|\nabla_\theta \log p_\theta(x, z)\|$ and the Markov inequality, we obtain:

$$A_2 \leq 2M\mathbb{P} \left( 2\frac{1}{k} \sum_{\ell=1}^k w^{(\ell)} \leq \mathbb{E}_{q_\phi}[w] \right)$$

$$\leq 2M\mathbb{P} \left( 2 \left( \frac{1}{k} \sum_{\ell=1}^k w^{(\ell)} - \mathbb{E}_{q_\phi}[w] \right) \leq -\mathbb{E}_{q_\phi}[w] \right)$$

$$\leq 2M\mathbb{P} \left( \left| \frac{1}{k} \sum_{\ell=1}^k w^{(\ell)} - \mathbb{E}_{q_\phi}[w] \right| \geq \frac{\mathbb{E}_{q_\phi}[w]}{2} \right) \leq \frac{\mathbb{E}_{q_\phi}[w^2]}{\mathbb{E}_{q_\phi}[w]^2} \frac{8M}{k},$$

which concludes the proof. $\qquad\square$

---

**Algorithm 2 Adaptive Stochastic Approximation for IWAE**

---

**Input:** Initial point $\theta_0$, maximum number of iterations $n$, step sizes $\{\gamma_k\}_{k\geq 1}$ and a hyperparameter $\alpha \geq 0$ to control the bias and MSE.
**for** $k = 0$ to $n-1$ **do**
   Compute the stochastic update $\nabla_{\theta,\phi}\mathcal{L}_{k^\alpha}^{\text{IWAE}}(\theta_k, \phi_k; X_{k+1})$ using $k^\alpha$ samples from the variational posterior distribution and adaptive steps $A_k$.
   Set $\theta_{k+1} = \theta_k - \gamma_{k+1}A_k\nabla_\theta\mathcal{L}_{k^\alpha}^{\text{IWAE}}(\theta_k, \phi_k; X_{k+1})$.
   Set $\phi_{k+1} = \phi_k - \gamma_{k+1}A_k\nabla_\phi\mathcal{L}_{k^\alpha}^{IWAE}(\theta_k, \phi_k; X_{k+1})$.
**end for**
**Output:** $(\theta_k)_{0\leq k\leq n}$

---

## B.2 BR-IWAE

In this section, we provide additional details on the Biased Reduced Importance Weighted Autoencoder (BR-IWAE). In IWAE, instead of estimating the gradient of the ELBO with respect to $\theta$ via the Monte Carlo method, we estimate the gradient of the true objective function $\mathbb{E}_{p_\theta(\cdot|x)}\left[\nabla_\theta \log p_\theta(x, z)\right]$ using the BR-SNIS estimator [14]. This estimator aims to reduce the bias of self-normalized importance sampling estimators without increasing the variance.

---

**Algorithm 3 BR-IWAE Gradient Estimator**

---

**Input:** Maximum number of iterations $t_{\max}$ for MCMC and number of samples $k$ from the variational distribution $q_\phi(\cdot \mid x)$.
**Initialization:** Draw $\tilde{z}_0$ from the variational distribution $q_\phi(\cdot \mid x)$.
**for** $t = 0$ to $t_{\max} - 1$ **do**
   Draw $I_{t+1} \in \{1, \ldots, k\}$ uniformly at random and set $z_{t+1}^{I_{t+1}} = \tilde{z}_t$.
   Draw $z_{t+1}^{1:k\backslash\{I_{t+1}\}}$ independently from the variational distribution $q_\phi(\cdot \mid x)$.
   Compute the unnormalized importance weights:

$$w_{t+1}^{(\ell)} = \frac{p_\theta(x, z_{t+1}^{(\ell)})}{q_\phi(z_{t+1}^{(\ell)} \mid x)} \quad \forall \ell \in \{1, \ldots, k\}.$$

   Normalize importance weights:

$$\omega_{t+1}^{(\ell)} = \frac{w_{t+1}^{(\ell)}}{\sum_{\ell=1}^{N} w_{t+1}^{(\ell)}} \quad \forall \ell \in \{1, \ldots, k\}.$$

   Select $\tilde{z}_{t+1}$ from the set $z_{t+1}^{1:k}$ by choosing $z_{t+1}^\ell$ with probability $\omega_{t+1}^{(\ell)}$.
**end for**
**Output:** $\left(z_t^{1:k}\right)_{1\leq t\leq t_{\max}}$ and $\left(\omega_t^{1:k}\right)_{1\leq t\leq t_{\max}}$.

---

The BR-SNIS estimator of $\mathbb{E}_{p_\theta(\cdot|x)}\left[\nabla_\theta \log p_\theta(x, z)\right]$ is given by:

$$\widehat{\nabla}_\theta \log p_\theta(x, z_{t_0:t_{\max}}^{1:k}) = \frac{1}{t_{\max} - t_0} \sum_{t=t_0+1}^{t_{\max}} \sum_{\ell=1}^{k} \omega_t^{(\ell)} \nabla_\theta \log p_\theta(x, z_t^\ell) \,,$$

where $t_0$ corresponds to a burn-in period. By [14, Theorem 4] the bias of this estimator decreases exponentially with $t_0$. The BR-IWAE algorithm proceeds in two steps, which are repeated during optimization:

- Update the parameter $\phi$ as in the IWAE algorithm, that is, for all $n \geq 1$:

$$\phi_{n+1} = \phi_n - \gamma_{n+1}A_n\nabla_\phi\mathcal{L}_k^{\text{IWAE}}(\theta_n, \phi_n; X_{n+1}) \,.$$

- Update the parameter $\theta$ by estimating (9) using BR-SNIS as detailed in Algorithm 3:

$$\phi_{n+1} = \phi_n - \gamma_{n+1}A_n\widehat{\nabla}_\theta \log p_\theta(X_{n+1}, z_{t_0:t_{\max}}^{1:k}) \,.$$

## B.3 Some Other Techniques for Reducing Bias

In the previous section, we discussed one technique for reducing bias, BR-IWAE. Here, we provide an overview of some other bias reduction techniques within our context. First, the jackknife bias-corrected estimator [62] is defined as:

$$\mathcal{L}^{\text{Jackknife}}(\theta, \phi; x) = k\mathcal{L}_k^{\text{IWAE}}(\theta, \phi; x) - (k-1)\mathcal{L}_{k-1}^{\text{IWAE}}(\theta, \phi; x) \,,$$

which achieves a reduced bias of $\mathcal{O}(k^{-2})$. This can also be generalized to have a bias of order $\mathcal{O}(k^{-m})$ for some $m \geq 1$ by considering:

$$\mathcal{L}_{k,m}^{\text{Jackknife}} = \sum_{j=0}^{m} c(k, m, j)\mathcal{L}_{k-j}^{\text{IWAE}} \,,$$

where the coefficients $c(k, m, j)$ are given as

$$c(k, m, j) = (-1)^j \frac{(k-j)^m}{(m-j)!j!} \,.$$

The Delta method Variational Inference (DVI) [71] is defined by:

$$\mathcal{L}_k^{DVI} = \mathbb{E}_{q_\phi^{\otimes k}(\cdot|x)} \left[ \log \frac{1}{k} \sum_{\ell=1}^{k} w^{(\ell)} + \frac{\bar{s}_k^2}{2k\bar{w}_k} \right] \,,$$

where

$$w^{(\ell)} = \frac{p_\theta(x, z^{(\ell)})}{q_\phi(z^{(\ell)} \mid x)}, \quad \bar{w}_k = \frac{1}{k} \sum_{\ell=1}^{k} w^{(\ell)} \quad \text{and} \quad \bar{s}_k^2 = \frac{1}{k-1} \sum_{\ell=1}^{k} (w^{(\ell)} - \bar{w}_k)^2 \,.$$

The Monte Carlo estimator of the Delta method Variational Inference objective achieves a reduced bias of $\mathcal{O}(k^{-2})$. Some other techniques for reducing bias include the iterated bootstrap for bias correction, the debiasing lemma [57], and Multi-Level Monte Carlo and its variants [39].

# C Application of Our Theorem to Bilevel and Conditional Stochastic Optimization

## C.1 Stochastic Bilevel Optimization

We consider the Stochastic Bilevel Optimization problem given by:

$$\min_{\theta \in \mathbb{R}^d} V(\theta) = \mathbb{E}_\xi \left[ f(\theta, \phi^*(\theta); \xi) \right] \quad \text{(upper-level)} \tag{11}$$

subject to

$$\phi^*(\theta) \in \underset{\phi \in \mathbb{R}^q}{\operatorname{argmin}} \mathbb{E}_\zeta \left[ g(\theta, \phi; \zeta) \right] \quad \text{(lower-level)}$$

where the upper and inner level functions $f$ and $g$ are both jointly continuously differentiable and $\xi$ and $\zeta$ are random variables. The goal of equation (11) is to minimize the objective function V with respect to $\theta$, where $\phi^*(\theta)$ is obtained by solving the lower-level minimization problem. This bilevel problem involves many machine learning problems with a hierarchical structure, which include hyper-parameter optimization [31], metalearning [30], policy optimization [37] and neural network architecture search [55]. The gradient of the objective function $V$ is given by:

$$\nabla V(\theta) = \nabla_\theta f(\theta, \phi^*(\theta)) - \nabla_{\theta\phi} g(\theta, \phi^*(\theta)) v^*,$$

where $v^*$ is the solution of the following linear system:

$$\nabla_\phi^2 g(\theta, \phi^*(\theta)) v = \nabla_\phi f(\theta, \phi^*(\theta)) .$$

Instead of computing $v^*$, the solution of the linear system above, [43, 17] proposes a method to estimate $v^*$. This estimation introduces bias in the gradient of the objective function.

Consider the following assumptions.

**H6** For all $\theta \in \mathbb{R}^d$, $g(\theta, \phi)$ is strongly convex with respect to $\phi$ with parameter $\mu_g > 0$.

**H7** (Regularity Lipschitz condition) Assume that $f, \nabla f, \nabla g, \nabla^2 g$ are respectively Lipschitz continuous with Lipschitz constants $\ell_{f,0}, \ell_{f,1}, \ell_{g,1}$ and $\ell_{g,2}$.

Assumptions H6 and H7 are the same assumptions used in [17] to obtain the convergence results with SGD. Furthermore, these two assumptions ensure that the first- and second-order derivatives of $f$ and $g$, as well as the solution mapping $\phi^*(\theta)$, are well-behaved.

**Proposition C.1.** *([17, Lemma 2.2]) Under Assumption 6, we have:*

$$\nabla V(\theta) = \nabla_\theta f(\theta, \phi^*(\theta)) - \nabla_{\theta\phi}^2 g(\theta, \phi^*(\theta)) \left[ \nabla_\phi^2 g(\theta, \phi^*(\theta)) \right]^{-1} \nabla_\phi f(\theta, \phi^*(\theta)) . \tag{12}$$

Due to the dependence of the minimizer of the lower-level problem $\phi^*(\theta)$, obtaining an unbiased estimate of $\nabla V(\theta)$ is challenging. To address this, we replace $\phi^*(\theta)$ in the gradient with $\phi$ and define

$$\bar{\nabla}_\theta f(\theta, \phi) := \nabla_\theta f(\theta, \phi) - \nabla_{\theta\phi}^2 g(\theta, \phi) \left[ \nabla_\phi^2 g(\theta, \phi) \right]^{-1} \nabla_\phi f(\theta, \phi) .$$

Furthermore, by estimating $\left[ \nabla_\phi^2 g(\theta, \phi) \right]^{-1}$, we define the stochastic update $H_k$ [17] as follows:

$$H_k = \nabla_\theta f(\theta_k, \phi_{k+1}; \xi_k) - \nabla_{\theta\phi}^2 g\left(\theta_k, \phi; \zeta_k^{(0)}\right) \widehat{G} \nabla_\phi f(\theta_k, \phi_{k+1}; \xi_k), \tag{13}$$

where $\widehat{G} = \frac{N}{\ell_{g,1}} \prod_{i=1}^{N'} \left( I - \frac{1}{\ell_{g,1}} \nabla_\phi^2 g\left(\theta_k, \phi_{k+1}; \zeta_k^{(i)}\right) \right)$ with $N'$ is drawn from $\{1, \ldots, N\}$ uniformly at random and $\left\{ \zeta^{(1)}, \ldots, \zeta^{(N')} \right\}$ are i.i.d. samples.

In Algorithm 4, we perform $T$ steps of SGD on the lower-level variable $\phi_k$ before updating the upper-level variable $\theta_k$ using adaptive methods such as Adagrad, RMSProp, or AMSGRAD.

**Lemma C.2.** *([33, Lemma 2.2]) Under Assumptions H6 and H7, for all $(\theta, \theta') \in \mathbb{R}^d \times \mathbb{R}^d$,*

$$\|\nabla V(\theta) - \nabla V(\theta')\| \le L_V \|\theta - \theta'\| ,$$

*with the constant $L_V$ is given by*

$$L_V = \ell_{f,1} + \frac{\ell_{g,1}(\ell_{f,1} + L_f)}{\mu_g} + \frac{\ell_{f,0}}{\mu_g} \left( \ell_{g,2} + \frac{\ell_{g,1}\ell_{g,2}}{\mu_g} \right),$$

*and $L_f$ is defined as $L_f = \ell_{f,1} + \frac{\ell_{g,1}\ell_{f,1}}{\mu_g} + \frac{\ell_{f,0}}{\mu_g} \left( \ell_{g,2} + \frac{\ell_{g,1}\ell_{g,2}}{\mu_g} \right).$*

**Algorithm 4 Stochastic Bilevel Optimization**

**Input:** Initial points $\theta_0, \phi_0$, maximum number of iterations for the upper-level $n$ and for the lower-level $T$, step sizes $\{\gamma_k, \tilde{\gamma}_k\}_{k \geq 1}$, momentum parameters $\rho_1, \rho_2 \in [0, 1)$ and regularization parameter $\delta \geq 0$.
Set $m_0 = 0, V_0 = 0$ and $\hat{V}_0 = 0$
**for** $k = 0$ to $n - 1$ **do**
    Set $\phi_{k,0} = \phi_k$.
    **for** $t = 0$ to $T - 1$ **do**
        $\phi_{k,t+1} = \phi_{k,t} - \tilde{\gamma}_{k+1} \nabla_\phi g (\theta_k, \phi_{k,t}; \zeta_{k,t})$
    **end for**
    Set $\phi_{k+1} = \phi_{k,T}$.
    Compute the stochastic update $H_k$ using $\phi_{k+1}$.
    $m_k = \rho_1 m_{k-1} + (1 - \rho_1) H_k$
    $V_k = \rho_2 V_{k-1} + (1 - \rho_2) H_k H_k^\top$
    $\hat{V}_k = \max \left( \hat{V}_{k-1}, \mathrm{Diag}(V_k) \right)$
    $A_k = \left[ \delta I_d + \hat{V}_k \right]^{-1/2}$
    $\theta_{k+1} = \theta_k - \gamma_{k+1} A_k m_k$
**end for**
**Output:** $(\theta_k, \phi_k)_{0 \leq k \leq n}$

---

**Lemma C.3.** *Under Assumptions H6 and H7, the following inequalities hold:*

$$\|\nabla V (\theta_k) - \mathbb{E}[H_k \mid \mathcal{F}_k]\|^2 \leq 2L_f^2 \|\phi_{k+1} - \phi^* (\theta_k)\|^2 + 2\tilde{b}_k^2 ,$$

$$\left\| \bar{\nabla}_\theta f(\theta, \phi) \right\| \leq \ell_{f,0} + \frac{\ell_{g,1} \ell_{f,0}}{\mu_g},$$

*where $L_f = \ell_{f,1} + \frac{\ell_{g,1} \ell_{f,1}}{\mu_g} + \frac{\ell_{f,0}}{\mu_g} \left( \ell_{g,2} + \frac{\ell_{g,1} \ell_{g,2}}{\mu_g} \right)$ and $\tilde{b}_k = \ell_{g,1} \ell_{f,1} \frac{1}{\mu_g} \left( 1 - \frac{\mu_g}{\ell_{g,1}} \right)^N$.*

*Proof.* For the bias term, since $\nabla V (\theta_k) = \bar{\nabla} f (\theta_k, \phi^* (\theta_k))$, we have:

$$\|\nabla V (\theta_k) - \mathbb{E}[H_k \mid \mathcal{F}_k]\|^2$$
$$= \left\| \bar{\nabla} f (\theta_k, \phi^* (\theta_k)) - \bar{\nabla} f (\theta_k, \phi_{k+1}) + \bar{\nabla} f (\theta_k, \phi_{k+1}) - \mathbb{E}[H_k \mid \mathcal{F}_k] \right\|^2$$
$$\leq 2 \left\| \bar{\nabla} f (\theta_k, \phi^* (\theta_k)) - \bar{\nabla} f (\theta_k, \phi_{k+1}) \right\|^2 + 2 \left\| \bar{\nabla} f (\theta_k, \phi_{k+1}) - \mathbb{E}[H_k \mid \mathcal{F}_k] \right\|^2$$
$$\leq 2L_f^2 \|\phi_{k+1} - \phi^* (\theta_k)\|^2 + 2\tilde{b}_k^2 ,$$

where we used [33, Lemma 2.2] for the first term and [37, Lemma 11] for the second term.

For the second inequality, we have:

$$\left\| \bar{\nabla}_\theta f(\theta, \phi) \right\| = \left\| \nabla_\theta f(\theta, \phi) - \nabla^2_{\theta\phi} g(\theta, \phi) \left[ \nabla^2_\phi g(\theta, \phi) \right]^{-1} \nabla_\phi f(\theta, \phi) \right\|$$
$$\leq \| \nabla_\theta f(\theta, \phi) \| + \left\| \nabla^2_{\theta\phi} g(\theta, \phi) \right\| \left\| \left[ \nabla^2_\phi g(\theta, \phi) \right]^{-1} \right\| \| \nabla_\phi f(\theta, \phi) \|$$
$$\leq \ell_{f,0} + \frac{\ell_{g,1} \ell_{f,0}}{\mu_g} .$$

$\square$

**Theorem C.4.** *Assume that H6 and H7 hold. Let $\theta_n \in \mathbb{R}^d$ be the $n$-th iterate of Algorithm 4, $\gamma_n = c_\gamma n^{-1/2}$ and $\tilde{\gamma}_n = c_{\tilde{\gamma}} n^{-1/2}/T$. For any $n \geq 1$, let $R \in \{0, \ldots, n\}$ be a uniformly distributed random variable. Assume the boundedness of the variance of the estimators of $\nabla f$, $\nabla g$, and $\nabla^2 g$. Then,*

$$\mathbb{E}\left[ \|\nabla V (\theta_R)\|^2 \right] = \mathcal{O} \left( \frac{\log n}{\sqrt{n}} + b_n \right).$$

*Proof.* By using Lemma C.3, $V$ is smooth and Lemma C.3, the bias and the gradient of $V$ are bounded. Using our Corollary 4.5, we obtain:

$$\mathbb{E}\left[\|\nabla V\left(\theta_R\right)\|^2\right] = \mathcal{O}\left(\frac{\log n}{\sqrt{n}} + b_n\right),$$

where

$$b_n = \mathcal{O}\left(\frac{\sum_{k=0}^n \gamma_{k+1}\tilde{b}_k^2 + \sum_{k=0}^n \gamma_{k+1}\|\phi_{k+1} - \phi^*\left(\theta_k\right)\|^2}{\sqrt{n}}\right).$$

Then, with [33, Lemma 2.3] and [17, Lemma 3], we derive:

$$b_n = \mathcal{O}\left(\frac{\sum_{k=0}^n \gamma_{k+1}\tilde{b}_k^2}{\sqrt{n}} + \frac{1}{\sqrt{n}}\right).$$

$\square$

We achieve a classical convergence rate of $\mathcal{O}(\log n/\sqrt{n})$ for Stochastic Bilevel Optimization problems. Two types of bias emerge in this context: firstly, the challenge of directly computing $\phi^*(\theta)$, and secondly, the necessity of estimating $[\nabla_\phi^2 g(\theta, \phi)]^{-1}$.

Our results extend those of [17] to the adaptive case, particularly Adagrad, RMSProp, and AMS-GRAD. This provides convergence guarantees for the Alternating Stochastic Gradient Descent (ALSET) method. We can apply our convergence analysis to Stochastic Min-Max and Compositional Problems, as well as to the Actor-Critic method with linear value function approximation [49], which can be viewed as a special case of the Stochastic Bilevel algorithm.

## C.2 Conditional Stochastic Optimization

We now consider a class of Conditional Stochastic Optimization:

$$\min_{\theta \in \mathbb{R}^d} V(\theta) := \mathbb{E}_\xi\left[f_\xi\left(\mathbb{E}_{\eta|\xi}\left[g_\eta(\theta, \xi)\right]\right)\right], \tag{14}$$

where $f_\xi(\cdot) : \mathbb{R}^q \to \mathbb{R}$ depends on the random vector $\xi$ and $g_\eta(\cdot, \xi) : \mathbb{R}^d \to \mathbb{R}^q$ is a vector-valued function dependent on both random vectors $\xi$ and $\eta$. The inner expectation is taken with respect to the conditional distribution of $\eta$ given $\xi$. Given certain conditions on the regularity of these functions, the gradient of $V$ as defined in (14) can be expressed as:

$$\nabla V(\theta) = \mathbb{E}_\xi\left[\left(\mathbb{E}_{\eta|\xi}\left[\nabla g_\eta(\theta, \xi)\right]\right)^\top \nabla f_\xi\left(\mathbb{E}_{\eta|\xi}\left[g_\eta(\theta, \xi)\right]\right)\right]. \tag{15}$$

Constructing an unbiased stochastic estimator of this gradient can be both costly and, in some cases, impractical. Instead, we opt for a biased estimator of $\nabla V(\theta)$, using just one sample $\xi$ and $m$ i.i.d. samples $\{\eta_j\}_{j=1}^m$ from the conditional distribution of $\eta$ given $\xi$:

$$\widehat{\nabla} V(\theta; \xi, \{\eta_j\}_{j=1}^m) := \left(\frac{1}{m}\sum_{j=1}^m \nabla g_{\eta_j}(\theta, \xi)\right)^\top \nabla f_\xi\left(\frac{1}{m}\sum_{j=1}^m g_{\eta_j}(\theta, \xi)\right). \tag{16}$$

**H8** For all $\xi$ and $\eta$, assume that $f_\xi(\cdot)$, $\nabla f_\xi(\cdot)$, $g_\eta(\cdot, \xi)$, and $\nabla g_\eta(\cdot, \xi)$ are respectively Lipschitz continuous with Lipschitz constants $\ell_{f,0}$, $\ell_{f,1}$, $\ell_{g,0}$ and $\ell_{g,1}$.

**H9** For all $\theta$ and $\xi$, we assume that $\mathbb{E}_{\eta|\xi}\left[\|g_\eta(\theta, \xi) - \mathbb{E}_{\eta|\xi}\left[g_\eta(\theta, \xi)\right]\|^2\right] \le \sigma_g^2$.

**Lemma C.5.** *([40, Lemma 2.2]) Under Assumptions H8 and H9, the following holds:*

$$\left\|\mathbb{E}\left[\widehat{\nabla} V(\theta; \xi, \{\eta_j\}_{j=1}^m)\right] - \nabla V(\theta)\right\|^2 \le \frac{\ell_{g,0}^2 \ell_{f,1}^2 \sigma_g^2}{m}.$$

**Lemma C.6.** *Under Assumption H8, we have:*

$$\|\nabla V(\theta) - \nabla V(\theta')\| \le \left(\ell_{g,1}\ell_{f,0} + \ell_{g,0}^2\ell_{f,1}\right)\|\theta - \theta'\|,$$

$$\|\nabla V(\theta)\| \le \ell_{g,0}\ell_{f,0}.$$

*Proof.* Denoting $G_\theta = \mathbb{E}_{\eta|\xi} [g_\eta(\theta, \xi)]$ and $\nabla G_\theta = \mathbb{E}_{\eta|\xi} [\nabla g_\eta(\theta, \xi)]$, we establish the smoothness of $V$ and Boundness of $\nabla V$.

Smoothness of $V$:

$$
\begin{aligned}
\|\nabla V(\theta) - \nabla V(\theta')\| &= \left\|\mathbb{E}_\xi \left[\nabla G_\theta^T \nabla f_\xi (G_\theta)\right] - \mathbb{E}_\xi \left[\nabla G_{\theta'}^T \nabla f_\xi (G_{\theta'})\right]\right\| \\
&\leq \left\|\mathbb{E}_\xi \left[\nabla G_\theta^T \nabla f_\xi (G_\theta)\right] - \mathbb{E}_\xi \left[\nabla G_{\theta'}^T \nabla f_\xi (G_\theta)\right]\right\| \\
&\quad + \left\|\mathbb{E}_\xi \left[\nabla G_{\theta'}^T \nabla f_\xi (G_\theta)\right] - \mathbb{E}_\xi \left[\nabla G_{\theta'} \nabla f_\xi (G_{\theta'})\right]\right\| \\
&\leq \left\|\mathbb{E}_\xi \left[(\nabla G_\theta - \nabla G_{\theta'})^T \nabla f_\xi (G_\theta)\right]\right\| \\
&\quad + \left\|\mathbb{E}_\xi \left[\nabla G_{\theta'}^T (\nabla f_\xi (G_\theta) - \nabla f_\xi (G_{\theta'}))\right]\right\| \\
&\leq \mathbb{E}_\xi \left[\|\nabla G_\theta - \nabla G_{\theta'}\| \|\nabla f_\xi (G_\theta)\|\right] \\
&\quad + \mathbb{E}_\xi \left[\|\nabla G_{\theta'}\| \|\nabla f_\xi (G_\theta) - \nabla f_\xi (G_{\theta'})\|\right] \\
&\leq \ell_{g,1} \ell_{f,0} \|\theta - \theta'\| + \ell_{g,0} \ell_{f,1} \mathbb{E}_\xi \left[\|G_\theta - G_{\theta'}\|\right] \\
&\leq \ell_{g,1} \ell_{f,0} \|\theta - \theta'\| + \ell_{g,0}^2 \ell_{f,1} \|\theta - \theta'\| .
\end{aligned}
$$

Boundness of $\nabla V$:

$$
\begin{aligned}
\|\nabla V(\theta)\| &= \left\|\mathbb{E}_\xi \left[\nabla G_\theta^T \nabla f_\xi (G_\theta)\right]\right\| \\
&\leq \mathbb{E}_\xi \left[\|\nabla G_\theta\| \|\nabla f_\xi (G_\theta)\|\right] \leq \ell_{g,0} \ell_{f,0} .
\end{aligned}
$$

$\square$

**Theorem C.7.** *Assume that H8 and H9 hold. Let $\gamma_n = c_\gamma n^{-1/2}$, $A_n$ denote the adaptive matrix in AMSGRAD and $\rho_1, \rho_2 \in [0, 1)$. For any $n \geq 1$, let $R \in \{0, \dots, n\}$ be a uniformly distributed random variable. Then,*

$$
\mathbb{E}\left[\|\nabla V(\theta_R)\|^2\right] = \mathcal{O}\left(\frac{\log n}{\sqrt{n}} + b_n\right),
$$

*where $b_n$ is defined by writing $m_k$ as the number of conditional samples at iteration $k$:*

$$
b_n = \mathcal{O}\left(\frac{\sum_{k=0}^n \frac{m_k}{\sqrt{k}}}{\sqrt{n}}\right).
$$

*Proof.* This is an immediate implication of Theorem 4.6 using Lemmas C.5 and C.6. $\square$

These results can also be extended to the Federated Conditional Stochastic Optimization problem [75], which is defined by:

$$
\min_{\theta \in \mathbb{R}^d} V(\theta) = \frac{1}{L} \sum_{\ell=1}^L \mathbb{E}_{\xi_\ell} \left[f_{\xi_\ell}^\ell \left(\mathbb{E}_{\eta_l|\xi_\ell} \left[g_{\eta_\ell}^\ell (\theta, \xi_\ell)\right]\right)\right],
$$

where $\mathbb{E}_{\xi_\ell} f_{\xi_\ell}^\ell (\cdot) : \mathbb{R}^q \to \mathbb{R}$ is the outer-layer function on the $\ell$-th device with the randomness $\xi_\ell$, and $\mathbb{E}_{\eta_\ell|\xi_\ell} g_{\eta_\ell}^\ell (\cdot, \xi_\ell) : \mathbb{R}^d \to \mathbb{R}^q$ is the inner-layer function on the $\ell$-th device with respect to the conditional distribution of $\eta_\ell$ given $\xi_\ell$. If the functions $f_{\xi_\ell}^\ell (\cdot)$ and $g_{\eta_\ell}^\ell (\cdot, \xi_\ell)$ for all $L$ devices verify Assumptions H8 and H9, we obtain the same convergence rate.

The following Table 2 provides a comprehensive summary of the key points, including the verification of our assumptions and the convergence results obtained in both Stochastic Bilevel Optimization and Conditional Stochastic Optimization.

Table 2: Bilevel and Conditional Stochastic Optimization with our biased adaptive SA framework.

| Applications | Stochastic Bilevel Optimization | Conditional Stochastic Optimization |
|---|---|---|
| Problem | $\min\limits_{\theta\in\mathbb{R}^d} \mathbb{E}_\xi\left[f(\theta,\phi^*(\theta);\xi)\right]$ s.t. $\phi^*(\theta)\in\operatorname*{argmin}\limits_{\phi\in\mathbb{R}^q}\mathbb{E}_\zeta\left[g(\theta,\phi;\zeta)\right]$ | $\min\limits_{\theta\in\mathbb{R}^d}\mathbb{E}_\xi\left[f_\xi\left(\mathbb{E}_{\eta\mid\xi}\left[g_\eta(\theta,\xi)\right]\right)\right]$ |
| Gradient | $\nabla_\theta f(\theta,\phi^*(\theta))-\nabla_{\theta\phi}g(\theta,\phi^*(\theta))v^*$ | $\mathbb{E}_\xi\left[\left(\mathbb{E}_{\eta\mid\xi}\left[\nabla g_\eta(\theta,\xi)\right]\right)^T\nabla f_\xi\left(\mathbb{E}_{\eta\mid\xi}\left[g_\eta(\theta,\xi)\right]\right)\right]$ |
| Lipchitz Constant H2 | $\ell_{f,1}+\frac{\ell_{g,1}\left(\ell_{f,1}+L_f\right)}{\mu_g}+\frac{\ell_{f,0}}{\mu_g}\left(\ell_{g,2}+\frac{\ell_{g,1}\ell_{g,2}}{\mu_g}\right)$ | $\ell_{g,1}\ell_{f,0}+\ell_{g,0}^2\ell_{f,1}$ |
| Bias Control H3 | $\ell_{g,1}\ell_{f,1}\frac{1}{\mu_g}\left(1-\frac{\mu_g}{\ell_{g,1}}\right)^N$ | $\frac{\ell_{g,0}^2\ell_{f,1}^2\sigma_g^2}{m}$ |
| Gradient Bound H5 | $\ell_{f,0}+\frac{\ell_{g,1}\ell_{f,0}}{\mu_g}$ | $\ell_{g,0}\ell_{f,0}$ |
| Convergence | $\mathcal{O}\left(\frac{\log n}{\sqrt{n}}+b_n\right)$ | $\mathcal{O}\left(\frac{\log n}{\sqrt{n}}+b_n\right)$ |

# D  Some Other Examples of Biased Gradients with Control on Bias

In this section, we explore examples of applications using biased gradient estimators while having control over the bias.

## D.1  Self-Normalized Importance Sampling

Let $\pi$ be a probability measure on a measurable space $(\mathsf{X},\mathcal{X})$. The objective is to estimate $\pi(f)=\mathbb{E}_\pi[f(X)]$ for a measurable function $f:\mathcal{X}\to\mathbb{R}^d$ such that $\pi(|f|)<\infty$. Assume that $\pi(\mathrm{d}x)\propto w(x)\lambda(\mathrm{d}x)$, where $w$ is a positive weight function and $\lambda$ is a proposal probability distribution, and that $\lambda(w)=\int w(x)\lambda(\mathrm{d}x)<\infty$. For a function $f:\mathcal{X}\to\mathbb{R}^d$ such that $\pi(|f|)<\infty$, the identity

$$\pi(f)=\frac{\lambda(\omega f)}{\lambda(\omega)}\,,\tag{17}$$

leads to the Self-Normalized Importance Sampling (SNIS) estimator:

$$\Pi_N f\left(X^{1:N}\right)=\sum_{i=1}^N\omega_N^i f\left(X^i\right),\quad \omega_N^i=\frac{w\left(X^i\right)}{\sum_{\ell=1}^N w\left(X^\ell\right)}\,,$$

where $X^{1:N}=\left(X^1,\ldots,X^N\right)$ are independent draws from $\lambda$ and the $\omega_N^i$ are called the normalized weights. [1] shows that the bias of the SNIS estimator can be expressed as:

$$\left\|\mathbb{E}\left[\Pi_N f\left(X^{1:N}\right)-\pi(f)\right]\right\|\le\frac{12}{N}\frac{\lambda\left(\omega^2\right)}{\lambda(\omega)^2}\,.$$

This particular type of estimator can be found in the domain of Monte Carlo methods, particularly in the context of Bayesian inference and Sequential Monte Carlo methods.

## D.2  Sequential Monte Carlo Methods

We focus here in the task of estimating the parameters, denoted as $\theta$, in Hidden Markov Models. In this context, the hidden Markov chain is denoted by $(X_t)_{t\ge0}$. The distribution of $X_0$ has density $\chi$ with respect to the Lebesgue measure $\mu$ and for all $t\ge0$, the conditional distribution of $X_{t+1}$ given $X_{0:t}$ has density $m_\theta(X_t,\cdot)$. It is assumed that this state is partially observed through an observation process $(Y_t)_{0\le t\le T}$. The observations $Y_{0:t}$ are assumed to be independent conditionally on $X_{0:t}$ and, for all $0\le t\le T$, the distribution of $Y_t$ given $X_{0:t}$ depends on $X_t$ only and has density $g_\theta(X_t,\cdot)$ with respect to the Lebesgue measure. The joint distribution of hidden states and observations is given by

$$p_\theta(x_{0:T},y_{0:T})=\chi(x_0)g_\theta(x_0,y_0)\prod_{t=0}^{T-1}m_\theta(x_t,x_{t+1})g_\theta(x_{t+1},y_{t+1})\,.$$

Our objective is to maximize the likelihood of the model:

$$p_\theta(y_{0:T})=\int p_\theta(x_{0:T},y_{0:T})\,\mathrm{d}x_{0:T}\,.$$

To use a gradient-based method for this maximization problem, we need to compute the gradient of the objective function. Under simple technical assumptions, by Fisher's identity,

$$\nabla_\theta \log p_\theta(y_{0:T}) = \int \nabla_\theta \log p_\theta(x_{0:T}, y_{0:T}) p_\theta(x_{0:T}|y_{0:T}) \mathrm{d}x_{0:T}$$

$$= \mathbb{E}_{x_{0:T} \sim p_\theta(.|y_{0:T})} \left[\nabla_\theta \log p_\theta(x_{0:T}, y_{0:T})\right]$$

$$= \mathbb{E}_{x_{0:T} \sim p_\theta(.|y_{0:T})} \left[\sum_{t=0}^{T-1} s_{t,\theta}(x_t, x_{t+1})\right],$$

where $s_{t,\theta}(x, x') = \nabla_\theta \log\{m_\theta(x, x') g_\theta(x, y_{t+1})\}$ for $t > 0$ and by convention $s_{0,\theta}(x, x') = \nabla_\theta \log g_\theta(x, y_0)$. Given that the gradient of the log-likelihood represents the smoothed expectation of an additive functional, one may opt for Online Smoothing algorithms to mitigate computational costs. The estimation of the gradient $\nabla_\theta \log p_\theta(y_{0:T})$ is given by:

$$H_\theta(y_{0:T}) = \sum_{i=1}^N \frac{\omega_T^i}{\Omega_T} \tau_{T,\theta}^i,$$

where $\{\tau_{T,\theta}^i\}_{i=1}^N$ are particle approximations obtained using particles $\{(\xi_T^i, \omega_T^i)\}_{i=1}^N$ targeting the filtering distribution $\phi_T$, i.e. the conditional distribution of $x_T$ given $y_{0:T}$. In the Forward-only implementation of FFBSm [20], the particle approximations $\{\tau_{T,\theta}^i\}_{i=1}^N$ are computed using the following formula, with an initialization of $\tau_0^i = 0$ for all $i \in [\![1, N]\!]$:

$$\tau_{t+1,\theta}^i = \sum_{j=1}^N \frac{\omega_t^j m_\theta(\xi_t^j, \xi_{t+1}^i)}{\sum_{\ell=1} \omega_t^\ell m_\theta(\xi_t^\ell, \xi_{t+1}^i)} \left\{\tau_{t,\theta}^j + s_{t,\theta}(\xi_t^j, \xi_{t+1}^i)\right\}, \quad t \in \mathbb{N}.$$

The estimator of the gradient $H_\theta(y_{0:T})$ computed by the Forward-only implementation of FFBSm is biased. The bias and MSE of this estimator are of order $\mathcal{O}(1/N)$ [20], where $N$ corresponds to the number of particles used to estimate it. Using alternative recursion methods to compute $\{\tau_{T,\theta}^i\}_{i=1}^N$ results in different algorithms, such as the particle-based rapid incremental smoother (PARIS) [64] and its pseudo-marginal extension [34] and Parisian particle Gibbs (PPG) [13]. In such cases, one can also control the bias and MSE of the estimator.

### D.3  Policy Gradient for Average Reward over Infinite Horizon

Consider a finite Markov Decision Process (MDP) denoted as $(\mathcal{S}, \mathcal{A}, R, P)$, where $\mathcal{S}$ represents the state space, $\mathcal{A}$ denotes the action space, $R : \mathcal{S} \times \mathcal{A} \to [0, R_{\max}]$ is a reward function, and $P$ is the transition model. The agent's decision-making process is characterized by a parametric family of policies $\{\pi_\theta\}_{\theta \in \mathbb{R}^d}$, employing the soft-max parameterization. The reward function is given by:

$$V(\theta) := \mathbb{E}_{(S,A) \sim v_\theta}[\mathrm{R}(S, A)] = \sum_{(s,a) \in \mathcal{S} \times \mathcal{A}} v_\theta(s, a) \mathrm{R}(s, a),$$

where $v_\theta$ represents the unique stationary distribution of the state-action Markov Chain sequence $\{(S_t, A_t)\}_{t \geq 1}$ generated by the policy $\pi_\theta$. Let $\lambda \in (0, 1)$ be a discount factor and $T$ be sufficiently large, the estimator of the gradient of the objective function $V$ is given by:

$$H_\theta(S_{1:T}, A_{1:T}) = \mathrm{R}(S_T, A_T) \sum_{i=0}^{T-1} \lambda^i \nabla \log \pi_\theta(A_{T-i}; S_{T-i}),$$

where $(S_{1:T}, A_{1:T}) := (S_1, A_1, \ldots, S_T, A_T)$ is a realization of state-action sequence generated by the policy $\pi_\theta$. It's important to note that this gradient estimator is biased, and the bias is of order $\mathcal{O}(1 - \lambda)$ [44].

### D.4  Zeroth-Order Gradient

Consider the problem of minimizing the objective function $V$. The zeroth-order gradient method is particularly valuable in scenarios where direct access to the gradient of the objective function is

challenging or computationally expensive. The zeroth-order gradient oracle obtained by Gaussian smoothing [61] is given by:

$$H_\theta(X) = \frac{V(\theta + \tau X) - V(\theta)}{\tau} X \; , \tag{18}$$

where $\tau > 0$ is a smoothing parameter and $X \sim \mathcal{N}(0, I_d)$ a random Gaussian vector. [61, Lemma 3] provide the bias of this estimator:

$$\|\mathbb{E}[H_\theta(X)] - \nabla V(\theta)\| \le \frac{\tau}{2} L(d+3)^{3/2} \; . \tag{19}$$

The application of these zeroth-order gradient methods can be found in generative adversarial networks [58, 16].

### D.5 Compressed Stochastic Approximation: Coordinate Sampling

The coordinate descent method is based on the iteration:

$$\theta_{n+1} = \theta_n - \gamma_{n+1} H_{\theta_n}(X_{n+1})_{j_n} e_{j_n} \; ,$$

where $\{e_1, \ldots, e_d\}$ is the canonical basis of $\mathbb{R}^d$ and $H_{\theta_n}(X_{n+1})_j$ is the $j$-th coordinate of the gradient. The randomized coordinate selection rule chooses $j_n$ uniformly from the set $\{1, 2, \ldots, d\}$. Alternatively, the Gauss-Southwell selection rule [63] uses:

$$j_{n+1} := \underset{j \in \{1, \ldots, d\}}{\operatorname{argmax}} |H_{\theta_n}(X_{n+1})_j| \; .$$

This corresponds to a greedy selection procedure since at each iteration we choose the coordinate with the largest directional derivative. Another approach to choosing $j_n$ is Coordinate Sampling [53], a variant of the stochastic gradient descent algorithm that incorporates a selection step by sampling to perform random coordinate descent. The distribution of $\zeta_{n+1}$, which selects the coordinate, is characterized by the probability weights vector $(w_n^{(1)}, \ldots, w_n^{(d)})$ defined as:

$$w_n^{(j)} = \mathbb{P}(\zeta_{n+1} = j | \mathcal{F}_n), \quad j \in \{1, \ldots, d\} \; .$$

This distribution of $\zeta_{n+1}$ is referred to as the coordinate sampling policy. The Stochastic Coordinate Gradient Descent algorithm is defined by:

$$\theta_{n+1} = \theta_n - \gamma_{n+1} D(\zeta_{n+1}) H_{\theta_n}(X_{n+1}) \; ,$$

where $D(k) = e_k e_k^\top \in \mathbb{R}^{d \times d}$ has its entries equal to 0 except for the $(k, k)$ entry, which is 1. Observe that the distribution of the random matrix $D(\zeta_{n+1})$ is fully characterized by the matrix $D_n = \mathbb{E}[D(\zeta_{n+1}) | \mathcal{F}_n] = \operatorname{Diag}(w_n^{(1)}, \ldots, w_n^{(d)})$. In this context, $A_n$ represents a diagonal matrix $D_n$ where the diagonal terms characterize the probability weights for sampling each coordinate. These weights typically depend on preceding iterations and even on current gradients. In this case, we always have $\beta_{n+1} \le 1$ and to control the minimum eigenvalue, we only require a lower bound on the probability weights. This method can be easily extended to incorporate biased gradients and adaptive steps by introducing $\bar{A}_n = D_n A_n$, where $A_n$ represents the adaptive matrix as before, and $D_n$ is the matrix of probability weights.

# E Experiment details and supplementary results

## E.1 Experiment with a Synthetic Time-Dependent Bias

To illustrate Theorem 4.1 and the impact of bias, we consider in Figure 3 a simple least squares objective function $V(\theta) = \|A\theta\|^2/2$ in dimension $d = 10$. We artificially add to every gradient a zero-mean Gaussian noise with variance $\sigma^2 = 0.01$ and a bias term $r_n = C_r n^{-r}$ at each iteration $n$. We use Adagrad with a learning rate $\gamma = 1/2$, $\beta = 0$ and $\lambda = 0$. Then, the bound of Theorem 4.1 is of the form $\mathcal{O}(n^{-1/2} + n^{-r})$.

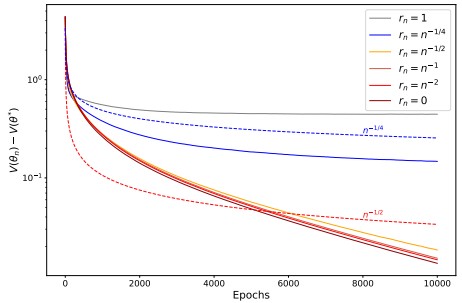 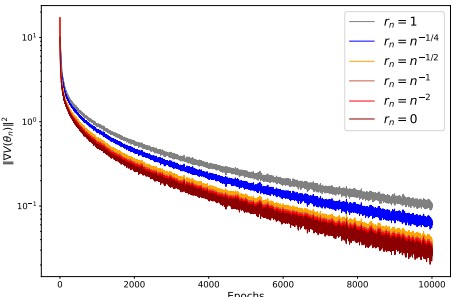

Figure 3: Value of $V(\theta_n) - V(\theta^*)$ (on the left) and $\|\nabla V(\theta_n)\|^2$ (on the right) with Adagrad for different values of $r_n = n^{-r}$ and a learning rate $\gamma_n = n^{-1/2}$. The dashed curve corresponds to the expected convergence rate $\mathcal{O}(n^{-1/4})$ for $r = 1/4$ and $\mathcal{O}(n^{-1/2})$ for $r \geq 1/2$.

We explore different values of $r_n \in \{1, n^{-1/4}, n^{-1/2}, n^{-1}, n^{-2}, 0\}$, where $r_n = 1$ corresponds to constant bias, $r_n = 0$ for an unbiased gradient, and the others exhibit decreasing bias. First, note that the impact of a constant bias term ($r_n = 1$) on the risk and the norm of gradients never vanishes. From $r_n = 1$ to $r_n = n^{-1/2}$, the effect of the bias decreases until a threshold is reached where there is no significant improvement. The convergence rate in the case $r_n = n^{-1/2}$ is then the same as in the case without bias, illustrating the fact that in this case the dominating term comes from the learning rate.

## E.2 Additional Experiments of IWAE

In this section, we provide detailed information about the experiments on CIFAR-10. We also conduct additional experiments on the FashionMNIST dataset. For all experiments, we use Adagrad, RMSProp, and Adam with a learning rate decay given by $\gamma_n = C_\gamma/\sqrt{n}$, where $C_\gamma = 0.01$ for Adagrad and $C_\gamma = 0.001$ for RMSProp and Adam. The momentum parameters are set to $\rho_1 = 0.9$ and $\rho_2 = 0.999$, and the regularization parameter $\delta$ is fixed at $5 \times 10^{-2}$. The impact of this regularization parameter will be illustrated later.

**Datasets.** We conduct our experiments on two datasets: FashionMNIST [76] and CIFAR-10. The FashionMNIST dataset is a variant of MNIST and consists of 28x28 pixel images of various fashion items, with 60,000 images in the training set and 10,000 images in the test set. CIFAR-10 consists of 32x32 pixel images categorized into 10 different classes. The dataset is divided into 60,000 images in the training set and 10,000 images in the test set.

**Models.** For FashionMNIST, we use a fully connected neural network with a single hidden layer consisting of 400 hidden units and ReLU activation functions for both the encoder and the decoder. The latent space dimension is set to 20. We use 256 images per iteration (235 iterations per epoch). For CIFAR-10 and CIFAR-100, we use a Convolutional Neural Network (CNN) architecture with 3 Convolutional layers and 2 fully connected layers with ReLU activation functions. The latent space dimension is set to 100. For both datasets, we use 256 images per iteration (196 iterations per epoch).

We estimate the log-likelihood using the VAE, IWAE, and BR-IWAE models, all of which are trained for 100 epochs. Training is conducted using the SGD, SGD with momentum, Adagrad, RMSProp,

and Adam algorithms with a decaying learning rate, as mentioned before. For SGD, we employ the clipping method to clip the gradients to prevent excessively large steps.

For this experiment, we set $k = 5$ samples in both IWAE and BR-IWAE, while restricting the maximum iteration of the MCMC algorithm to 5 and the burn-in period to 2 for BR-IWAE. For comparison, we estimate the Negative Log-Likelihood using these three models with SGD, SGD with momentum, Adagrad, RMSProp, and Adam, and the results are presented in Table 3. Similar to the case of CIFAR-10, we observe that IWAE outperforms VAE, while BR-IWAE outperforms IWAE by reducing bias in all cases. The adaptive methods surpass SGD, and momentum further improves their performances. Consequently, Adam excels among all algorithms due to its adaptive steps and momentum.

Table 3: Comparison of Negative Log-Likelihood on the FashionMNIST Test Set (Lower is Better).

| Algorithm | VAE | IWAE | BR-IWAE |
|---|---|---|---|
| SGD | 247.2 | 244.9 | 244.0 |
| SGD with momentum | 244.6 | 240.2 | 238.4 |
| Adagrad | 245.8 | 241.4 | 240.5 |
| RMSProp | 242.6 | 239.3 | 237.8 |
| Adam | 240.3 | 237.8 | 236.1 |

Similarly, as we did in the case of CIFAR-10, we incorporate a time-dependent bias that decreases by choosing a bias of order $\mathcal{O}(n^{-\alpha})$ at iteration $n$. We vary the value of $\alpha$ for both FashionMNIST and CIFAR-100.

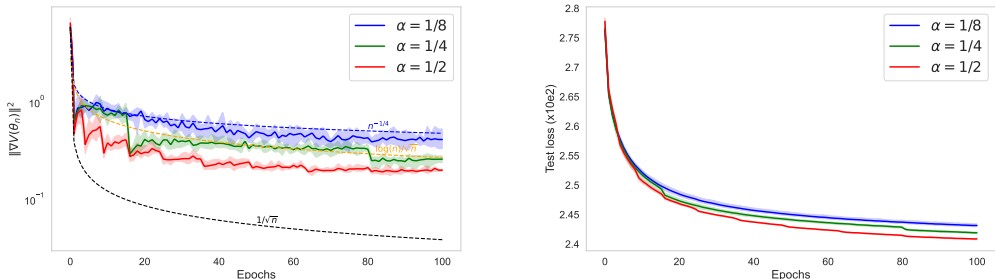

Figure 4: IWAE on the FashionMNIST Dataset with Adagrad for different values of $\alpha$. Bold lines represent the mean over 5 independent runs.

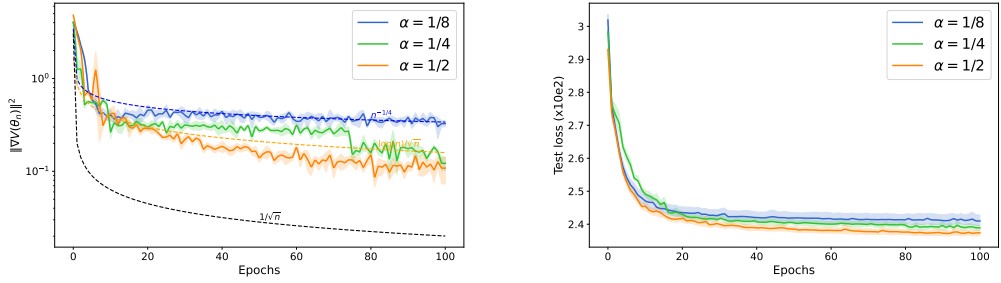

Figure 5: IWAE on the FashionMNIST Dataset with RMSProp for different values of $\alpha$. Bold lines represent the mean over 5 independent runs.

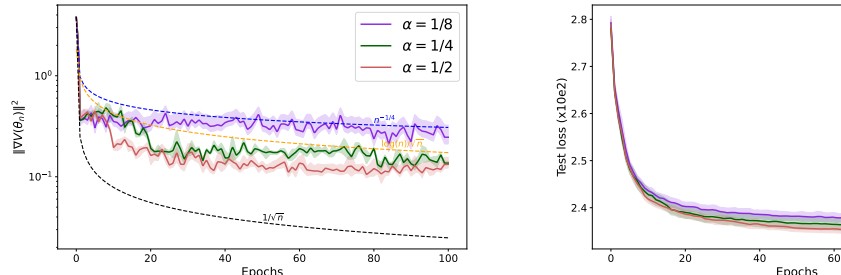

Figure 6: IWAE on the FashionMNIST Dataset with Adam for different values of $\alpha$. Bold lines represent the mean over 5 independent runs.

All figures are plotted on a logarithmic scale for better visualization and with respect to the number of epochs. The dashed curve corresponds to the expected convergence rate $\mathcal{O}(n^{-1/4})$ for $\alpha = 1/8$, and $\mathcal{O}(\log n/\sqrt{n})$ for $\alpha = 1/4$, as well as for $\alpha = 1/2$, just as in the case of CIFAR-10. We can clearly observe that for all cases, convergence is achieved when $n$ is sufficiently large. In the case of the FashionMNIST dataset, the bound seems tight, and the convergence rate of $\mathcal{O}(n^{-1/2})$ does not seem to be possible to reach, in contrast to the case of CIFAR-10 where the curves corresponding to $\alpha = 1/4$ and $\alpha = 1/2$ approach the $\mathcal{O}(n^{-1/2})$ convergence rate. For all figures, with a larger $\alpha$, the convergence in both the squared gradient norm and negative log-likelihood occurs more rapidly.

**Additional Experiments on CIFAR-10 Dataset.**

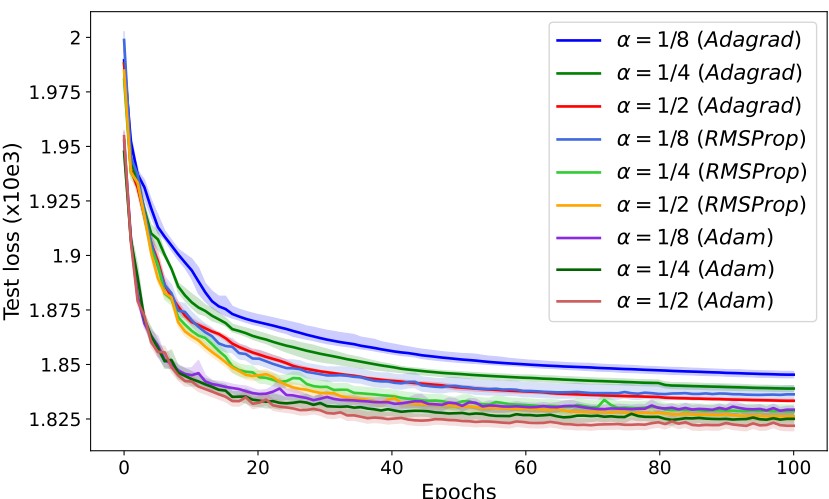

Figure 7: Negative Log-Likelihood on the test set on the CIFAR-10 Dataset for IWAE with Adagrad, RMSProp, and Adam. Bold lines represent the mean over 5 independent runs.

**The effect of $C_\gamma$.**

Figure 8 illustrates the convergence in both the squared gradient norm and the negative log-likelihood for $C_\gamma = 0.001$ and $C_\gamma = 0.01$ in Adagrad. In the case of the squared gradient norm, we have only plotted the results for $C_\gamma = 0.001$ for better visualization, and the plot for $C_\gamma = 0.01$ was already presented in Figure 2. It is clear that when $C_\gamma$ is set to 0.001, the convergence of the negative log-likelihood is slower. Similarly, the convergence in the squared gradient norm for $C_\gamma = 0.001$ achieves convergence, but it is slower compared to the case of $C_\gamma = 0.01$.

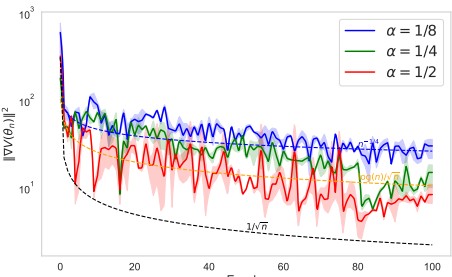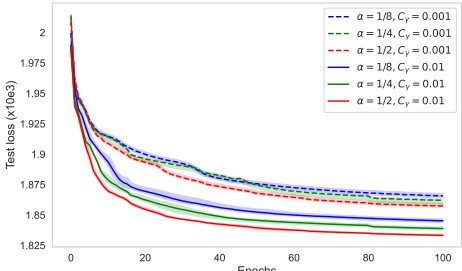

Figure 8: IWAE on the CIFAR-10 Dataset with Adagrad for different values of $\alpha$ and $C_\gamma$. Bold lines represent the mean over 5 independent runs.

**The Impact of regularization parameter $\delta$.**

In Section A.7, we discussed the impact of the regularization parameter $\delta$ in Adam. It has been empirically observed that the performance of adaptive methods can be sensitive to the choice of this parameter. Here, we illustrate the impact of this regularization parameter in IWAE. To achieve this, we plot the test loss for different sets of values for $\delta \in \{10^{-8}, 10^{-5}, 10^{-3}, 10^{-2}, 5 \times 10^{-2}, 10^{-1}\}$ in Figure 9.

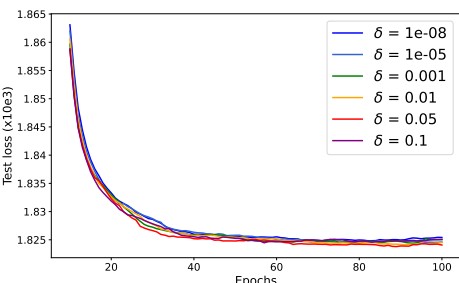

Figure 9: IWAE on the CIFAR-10 Dataset with Adam for different values of $\delta$. Lines represent the mean over 5 independent runs.

Our experimental results align with prior work [77, 67, 73], affirming the consistent impact of $\delta$. Notably, we find that employing $\delta = 5 \times 10^{-2}$ yields improved performance in IWAE.

**The Impact of Bias over Time.**

Our experiments illustrate the negative log-likelihood with respect to epochs, and we observed that a higher value of $\alpha$ leads to faster convergence. The key point to consider when tuning $\alpha$ is that while convergence may be faster in terms of iterations, it may lead to higher computational costs. To illustrate this, we set a fixed time limit of 1000 seconds and tested different values of $\alpha$, plotting the test loss as a function of time in Figure 10. It is clear that with $\alpha = 1/8$, the convergence is always slower, whereas choosing $\alpha = 1/4$ achieves faster convergence than $\alpha = 1/2$. While the difference may seem small here, with more complex models, the disparity becomes significant. Therefore, it is essential to tune the value of $\alpha$ to attain fast convergence and reduce computational time.

In this paper, all simulations were conducted using the Nvidia Tesla T4 GPU. The total computing hours required for the results presented in this paper are estimated to be around 100 to 200 hours of GPU usage.

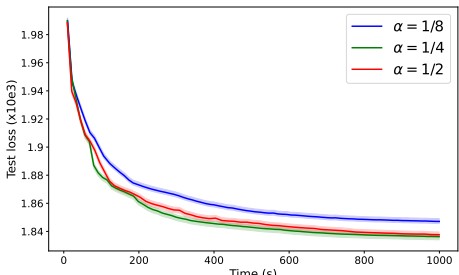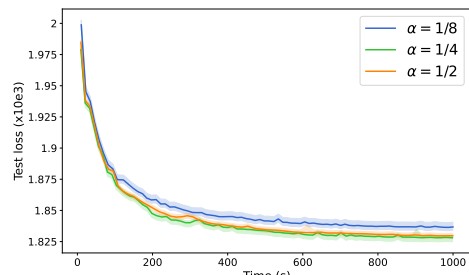

Figure 10: Negative Log-Likelihood on the test set of the CIFAR-10 Dataset for IWAE with Adagrad (on the left) RMSProp (on the right) for Different Values of $\alpha$ over time (in seconds). Bold lines represent the mean over 5 independent runs.

