# OpenReview forum: "Non-asymptotic Analysis of Biased Adaptive Stochastic Approximation"
_NeurIPS.cc/2024/Conference — NeurIPS 2024 poster_

### Official Review · Reviewer_nF8X · 2024-07-08

**Soundness:** 2
**Presentation:** 3
**Contribution:** 2
**Rating:** 4
**Confidence:** 3

**Summary:**

This paper studied convergence rate of general adaptive stochastic approximation with biased gradient. This is an important field since only biased gradient is accessible in many practical machine learning problem. The authors establied non-asymptotic bound based on various assumption, some of which are quite strong.

**Strengths:**

The authors provided convergence analysis of general biasd adaptive optimization method, and applied their results to popular adaptive algorithms including Adagrad, RMSProp and AMSGrad. They also mentioned various practical machine learning problems suffering from biased gradient. Extensive numerical experiments were conducted to verify the theoretical rate.

**Weaknesses:**

1. The techinque is not novel compared with existing works on biased SGD, e.g. [1]. The authors directly impose an upper and lower bound of preconditioning matrix in H3, in which case the analysis is not very different from that of SGD. A better way is to establish the boundedness of preconditioning matrix along the tracjectory as in e.g., [2], instead of taking it as prior assumption.

2. The authors claimed that they analyzed Adam throught out the paper, but in fact they only did AMSGRAD. This is misleading since they are two totally different algorithms with distinct practical performance. I suggest the authors correct this point.

### Typos:

line 230: $\mathcal{O}(n^{-1/4})$->$\mathcal{O}(\log n/n^{1/2})$

[1] Liu, Yin, and Sam Davanloo Tajbakhsh. "Adaptive stochastic optimization algorithms for problems with biased oracles." arXiv preprint arXiv:2306.07810 (2023).

[2] Li, Haochuan, Alexander Rakhlin, and Ali Jadbabaie. "Convergence of adam under relaxed assumptions." Advances in Neural Information Processing Systems 36 (2024).

**Questions:**

In Theorem 4.2, what does $\delta_j$ stand for? Does it correspond to $\delta$ in eq (4)? What is the dependence of $\delta$ in the final convergence rate of Adagrad and other algorithms?

**Limitations:**

The limitations are pointed out in the weakness part.

---

> ### Author Rebuttal · Authors · 2024-08-06
>
> Thank you for your comments.
>
> It is true that the stochastic optimization literature is very rich and is the focus of a great deal of research activity.  For instance, the work proposed in [81] focuses on the theoretical analysis of stochastic optimization problems with biased gradients.
> First of all, the proof for adaptive biased SA differs from that of biased SGD in how we control the preconditioning matrix. The adaptivity mentioned in [81] refers to how the authors update the gradient, not to the adaptive steps used in Adagrad, RMSProp, and Adam. The title was even changed since the first submission by removing "Adaptive".
> Furthermore, our results for Adagrad and RMSProp in Section 4.3, as well as the extension to Adam in Section 4.4, provide new theoretical bounds for adaptive optimization algorithms widely used in machine leaning. Their proofs are significantly different from the sketch proof of biased SGD.
>
> In Theorems 4.1 and 4.2, since the exact form of $A_n$ is unknown (as we aim to encompass all adaptive algorithms), we must assume certain properties about the preconditioning matrix, specifically the bound of its eigenvalues (H4). However, when applied to algorithms such as Adagrad, RMSProp, and Adam (AMSGRAD), we have demonstrated that this assumption holds.
> In [82], they only consider the convergence results of Adam in an unbiased setting, which means they do not need to make assumptions about the preconditioning matrix. The assumption on the preconditioning matrix in our case is only for the general case. The convergence results for AMSGRAD with biased gradients do not take the boundedness of the preconditioning matrix into account.
>
> We believe that the originality of the contribution of the paper is that we cover adaptive step sizes applied to the biased gradient setting with relaxed assumptions (for instance expected smoothness) as supported for instance by Reviewer aKkE and Reviewer 3iVV.
>
> $\textbf{The authors claimed that they analyzed Adam throught out the paper, ... I suggest the authors correct this point.}$
>
> - Since AMSGRAD is a minor modification of Adam and is not as well known as Adam, we believed it makes more sense to refer to Adam. Even in packages like PyTorch and TensorFlow, Adam has an option "AMSGRAD" to use this algorithm. The difference between these two algorithms is already mentioned in lines 221-225 of Section 4.4 of the paper. Furthermore, [64] highlights the convergence difficulties of Adam while proposing AMSGRAD.
> However, we have revised the paper to address this point and clarify the distinction between the two algorithms.
>
> $\textbf{In Theorem 4.2, what does $\delta_{j}$ stand for? ...}$
>
> - In Theorem 4.2, $\delta_{j}$ is defined as $\delta_{j} = L\gamma_{j}^{2}\beta_{j}^{2} /2$. This notation is introduced to simplify the terms in the convergence rate analysis, and $\delta$ represents the regularization parameter.
> Regarding the final convergence rate, there is a term $\log \left( 1 + \frac{nM^2}{\delta}\right) / \sqrt{n}$, which is asymptotically similar to $\log n / \sqrt{n}$.
>
> $\textbf{The authors establied non-asymptotic bound based on various assumption, some of which are quite strong.}$
>
> - As the other assumptions are classical in the literature, we set the focus more on H3 and H4.
> As discussed after Assumption H3, it is shown to be a minimal requirement for achieving the convergence rate in adaptive SA (see also [20]).
> Assumption H4 [10, 32] relates to the adaptive algorithm. We have shown that it can be verified in Adagrad, RMSprop, and Adam. For other algorithms where verification is difficult, we can use the truncation method described after this assumption.
> The truncation method involves replacing the random matrices $\tilde{A}\_n = \min${$||A\_n||,\beta\_{n+1}$}$A\_n/||A\_n||$). It is also stated that truncation is only used with very low probability, which means that the estimators $A_{n}$ are minimally affected by these truncations [32]. Since we choose $\beta_{n+1}$, we have control over the convergence rate.
> We provided additional comments on the assumptions for instance to Reviewer 1qy9.
>
> $\textbf{line 230: $\mathcal{O}\left(n^{-1/4}\right) \rightarrow \mathcal{O}\left(\log n / \sqrt{n} \right)$}$
>
> - If the additive bias $b_n$ in the convergence rate is of order $\mathcal{O}\left(\log n / \sqrt{n} \right)$, we achieve a convergence rate of $\mathcal{O}\left(\log n / \sqrt{n} \right)$. However, in line 230, we want to state that if the bias of the gradient estimator is of order $\mathcal{O}\left(n^{-1/4}\right)$ (since $r_{n+1}$ represents an additive bias term, generally of the order of the square of the bias of the gradient estimator, as mentioned after H3), then we achieve a convergence rate of $\mathcal{O}\left(\log n / \sqrt{n} \right)$.

---

> > ### Comment · Reviewer_nF8X · 2024-08-08
> >
> > Thanks for the reply. I would like to clarify that regarding w1, I notice in the analysis of Adagrad, RMSProp and AMSGrad, the Assumption H5 is used. This assumption, along with the existence of $\delta$, directly imposes an upper and lower bound on the preconditioning matrix globally. Hence I don't think the analysis will be too much different from biasd SGD. This is different from [2] which establishes the boundedness along the tracjectory.

---

> > > ### Author Response · Authors · 2024-08-10
> > >
> > > Thank you for your comments. Regarding Adagrad and RMSProp, although the analysis may appear similar to that of biased SGD under Assumption H3(i), verifying H3(i) requires managing the form $A_n$, which differs from the approach used in biased SGD, which is detailed in line 573. Additionally, the analysis of AMSGrad differs significantly from both the general case and the specific applications to Adagrad and RMSProp.
> > >
> > > The proof for AMSGrad is derived from scratch and does not rely on assumptions about the boundedness of eigenvalues. However, we do assume that gradients are bounded (as per Assumption H5). Even with this assumption, the proof for AMSGrad is different from the analysis of biased SGD, as detailed in Appendix A.6.
> > >
> > > While Assumption H5 is common in adaptive algorithm literature [8, 64, 70, 71], recent works have provided theoretical guarantees for unbiased gradients without relying on the boundedness of gradients (e.g., [82]).
> > > Our work focuses on adaptive algorithms with biased gradients, including scenarios with decreasing bias. We have shown that in various applications, such as Stochastic Bilevel Optimization and Conditional Stochastic Optimization, all our assumptions, including H3 and H5, can be verified.
> > > Most results that do not rely on Assumption H5 are provided with high probability. Furthermore, the obtained bounds generally have a rate of $\mathcal{O}\left(\text{poly}(\log n) / \sqrt{n} \right)$, with a loss of a logarithmic factor in the polynomial.
> > > However, even some recent works on SGD and adaptive methods continue to rely on the assumption of bounded gradients (e.g., see [1, 2]).
> > >
> > > One approach to avoid assumption H5 is to use the truncation method. To control the smallest eigenvalue in Adagrad, we can consider a truncation that leads to
> > >
> > > $$
> > > A_{n} = \left( \text{diag} \left( \beta_{n+1}^{-2}I_{d} + \frac{1}{n+1}\sum_{k=0}^{n}H_{\theta_{k}} \left( X_{k+1} \right) H_{\theta_{k}}\left( X_{k+1} \right)^{T} \mathbf{1}\_{|| H_{\theta_{k}} \left( X\_{k+1} \right) ||^{2} \leq v\_{k+1}} \right)   \right)^{-1/2} ,
> > > $$
> > > so that
> > > $$
> > > \lambda_{min} \left( A_{n} \right) \geq  \left(\beta_{1}^{-1} + \frac{1}{\sqrt{n+1}}\sqrt{\sum_{k=0}^{n}v_{k+1}} \right)^{-1} .
> > > $$
> > > Then, choosing $v_{k} = C_{v}k^{-v}$ with $v \geq 0$ leads to
> > > $$
> > > \lambda_{\min} \left( A_{n} \right) \geq \frac{1}{\beta_{1}^{-1} + \sqrt{C_{v}} (n+1)^{v/2}} =: \lambda_{n+1}.
> > > $$
> > > The value of $v$ can be chosen arbitrarily to achieve the desired convergence rate. However, the larger the chosen $v$, the less truncation is applied, which may result in a slower convergence rate. It is also possible to adjust other parameters, such as $\gamma_{n+1}$ and $\beta_{n+1}$.
> > >
> > > While there are a few other works on the scalar version of Adagrad with biased gradients, their assumptions are stronger than ours.
> > > Our work is the first step to providing the general convergence results for the biased gradients and adaptive algorithms for any form of $A_n$, which applies to Adagrad and RMSProp and extends to AMSGRAD under the assumption of bounded gradients.
> > >
> > > Additionally, at the suggestion of Reviewer ioBT and Reviewer BcqM, we have provided convergence results for the Martingale and Markov Chain cases. Our work focuses on deriving convergence results for various adaptive algorithms based on commonly used assumptions in the literature, applicable to many different scenarios. We believe that our paper is a valuable contribution to the community, and that exploring convergence results for biased gradients without relying on Assumption H5 may require a separate paper.
> > > We plan to discuss the boundedness of gradients along the trajectory and will address this in future work.
> > >
> > >
> > > [1] Xidong Wu, Jianhui Sun, Zhengmian Hu, Junyi Li, Aidong Zhang, and Heng Huang. Federated conditional stochastic optimization. In Advances in Neural Information Processing Systems, volume 36, 2024.
> > >
> > > [2] Xidong Wu, Feihu Huang, Zhengmian Hu, and Heng Huang. Faster adaptive federated learning. In Proceedings of the AAAI conference on artificial intelligence, volume 37, pages 10379–10387, 2023.

---

### Official Review · Reviewer_3iVV · 2024-07-11

**Soundness:** 3
**Presentation:** 4
**Contribution:** 3
**Rating:** 7
**Confidence:** 3

**Summary:**

This paper aims to analyze SGD with biased gradients in a non-asymptotic manner, where the steps  are also adaptive. In particular, under certain assumptions and conditions, it provides convergence guarantees and establishes convergence rates for a critical point of non-convex smooth functions for various adaptive methods.

Overall, I think the theoretical results of this paper are interesting and insightful.

**Strengths:**

- This paper is well-organized and clearly written. In particular, it well demonstrates the motivation of the study in the introduction and accurately compares its contributions with previous related works. All the theorems and the corresponding set of assumptions are concretely presented. In addition, the structures of proofs in the Appendix are also clear and easy to follow.

- The proposed methods are general in the sense that it not only covers the adaptive step sizes (e.g., Adam) but also applies to the biased gradient, making it a comprehensive and interesting work in the direction of analyzing non-asymptotic convergence guarantees for variants of SGD.

- The numerical experiments for IWAE and BR-IWAE are comprehensive and well support the theoretical claims.

**Weaknesses:**

One thing unclear to me is the role of parameters that are related to sampling noise (e.g. batch size) in the theorems. These parameters are supposed to be crucial since they are actually closely related to the training process and generalization. How do theorems in this paper characterize the effects of such parameters (other than some simple bounds from the assumptions)?

**Questions:**

- Please see the weaknesses part.

- In addition, although I understand that the settings considered in this paper are novel, there are already a lot of works in closely related topics. Does there exist any technical difficulty that cannot be solved by techniques from previous works? If it does, what are the technical difficulties compared to previous works?

**Limitations:**

The authors have adequately addressed the limitations in the main part of the paper.

---

> ### Author Rebuttal · Authors · 2024-08-06
>
> Thank you for your positive and thoughtful review and the opportunity to clarify our contributions.
>
> - In our framework, the gradient is computed using a single sample, as is common in many theoretical results. However, this can easily be extended to account for batch size. In such a case, instead of having $\sigma^2$ (variance noise), $\sigma^2$ would be scaled by the batch size.
> For instance, denoting as $B_k$ the batch size at iteration $k$, the convergence rate in Theorem 4.2 transforms into the following:
> $$
> \mathbb{E}\left[\left\| \nabla V\left(\theta_{R}\right)\right\|^{2}\right] \leq 2\frac{\mathbb{E}[V(\theta_{0}) - V(\theta^{*})] + \sum_{k=0}^{n} w_{k+1} \delta_{k+1} \sigma_{k}^2 / B_{k}  + \sum_{k=0}^{n} w_{k+1} \gamma_{k+1} \lambda_{k+1} r_{k+1} }{\sum_{j=0}^{n} w_{j+1} \gamma_{j+1} \lambda_{j+1}}.
> $$
> Here, we observe that taking a large $B_k$ leads to a smaller second term in the convergence rate (the reduction of the variance).
> It is well-known that increasing the batch size leads to faster convergence but also increases computation time. Therefore, a balance must be struck between batch size and computational efficiency. These comments will be added in the revised version of the paper.
>
>
> - Most of the existing literature on biased SA focuses on SGD with biased gradients, assuming a constant bias throughout the iterations. Only a few studies address biased gradients with adaptive steps, and these typically focus on the scalar version of Adagrad with strong assumptions.
> For instance, [3] studied convergence results for biased gradients with Adagrad in the Markov chain case, focusing on the norm of the gradient of the Moreau envelope while assuming the boundedness of the objective function.
> In contrast, our work provides convergence rates in a broader and more general setting that encompasses various applications and adaptive algorithms. One of the key challenges addressed in our work is the need to manage the preconditioning matrix effectively.
> A significant technical difficulty in this context arises from the term $\mathbb{E} \left[ \left\langle \nabla V \left( \theta_{n} \right), A_{n}H_{\theta_{n}} \left( X_{n+1} \right) \right\rangle \mid \mathcal{F}\_{n} \right]$, where the stochasticity depends on $X_{n+1}$, which is part of the biased gradient estimator $H_{\theta_{n}} \left( X_{n+1} \right)$, as well as the preconditioning matrix $A_n$. To address this challenge, Assumption H3$(i)$ was introduced.
> Furthermore, we have shown that for Adagrad, Adam, and RMSProp (Corollary 4.5), a sufficient condition to verify H3$(i)$ is to bound the bias of the gradient, that is, to show that there exists $\tilde b_{n+1}$ such that
> $$||\mathbb{E}[H_{\theta_n}(X_{n+1})\mid\mathcal{F}\_n]-\nabla V(\theta_n)||\leq\tilde b\_{n+1}.$$
> Compared to existing convergence rates for Adagrad with biased gradients, our approach is notable for its weaker assumptions and its coverage of both the scalar and diagonal versions of Adagrad. Additionally, it introduces novel results for other algorithms, such as RMSProp and Adam.

---

> > ### Comment · Reviewer_3iVV · 2024-08-09
> > **Thanks for the clarification**
> >
> > I thank the authors for the detailed response which addressed my concerns and questions. I keep my positive score unchanged.

---

### Official Review · Reviewer_1qy9 · 2024-07-20

**Soundness:** 4
**Presentation:** 4
**Contribution:** 3
**Rating:** 7
**Confidence:** 3

**Summary:**

This paper proposes non-asymptotic convergence guarantuees on several gradient-based optimization methods, ranging from the SGD to AdaGrad-like methods, when the estimation of the gradients is biased. More specifically, the convergence results include a theorem with the Polyak-Lojasiewicz condition and another one without the PL condition, both in a non-convex smooth setting.

**Strengths:**

# Originality

This paper studies a problem that is not yet broadly dealt with.

# Clarity

This paper is easy to read and the statements are clear.

# Significance

The setup of the provided convergence results include non-convex smooth settings, which are very common in ML.

# Quality

Overall, the theoretical results are sound and the proofs are easy to read (and seem to involve only classical techniques).

The experimental section provides examples where the estimation is naturally biased, which helps to evaluate the significance of the results.

**Weaknesses:**

# Signifiance

How limitating are hypotheses H1-5? It is difficult to check when the proposed models fulfill them.

**Questions:**

How limitating are hypotheses H1-5? It is difficult to check when the proposed models fulfill them.

**Limitations:**

-

---

> ### Author Rebuttal · Authors · 2024-08-06
>
> Thank you for your positive and thoughtful comments.
>
> There are three types of hypotheses discussed: those concerning only the adaptive algorithm, those concerning only the application (objective function), and those concerning both. These assumptions are discussed in the paper, but we will provide more detailed comments in the revised version.
>
> -  Assumption H4 [10, 32] relates only to the adaptive algorithm. We have shown that it can be verified in Adagrad, RMSprop, and Adam. For other algorithms where verification is difficult, we can use the truncation method described after this assumption.
>     The truncation method involves replacing the random matrices $\tilde{A}\_n = \min${$||A\_n||,\beta\_{n+1}$}$A\_n/||A\_n||$). It is also stated that truncation is only used with very low probability, which means that the estimators $A_{n}$ are minimally affected by these truncations [32]. Since we choose $\beta_{n+1}$, we have control over the convergence rate.
>
> - Assumption H1 relates to the Polyak-Łojasiewicz (PL) condition. We provide convergence rates with and without this condition.  This assumption is weaker than strong convexity and remains satisfied even when the function is non-convex. Note that it has been studied theoretically [42] and has been verified empirically in many applications, such as in deep networks [24] and for Linear Quadratic Regulator [26].
>
> - Assumption H2 is crucial for obtaining the convergence rate [9, 56]. This assumption can be restricted to the generalized smoothness condition [80]. Since we assume the boundedness of gradients (H4), the smoothness and generalized smoothness are equivalent. Assumption H4 is standard in adaptive algorithms [18, 64, 70, 71] and, in practice, can be satisfied by clipping the gradient.
>
> - Assumption H3 is the only assumption that relates to both the adaptive algorithm and the objective function. H3$(ii)$ is a relaxed assumption on the gradient variance, referred to as “expected smoothness” [20], as mentioned in our paper and noted by reviewer aKkE.
>     Since H3$(i)$ involves the preconditioning matrix and the objective function, providing a necessary and sufficient condition in a general setting is challenging.
>     However, we have shown that for Adagrad, Adam, and RMSProp (Corollary 4.5), a sufficient condition to verify H3$(i)$ is to bound the bias of the gradient, that is, to show that there exists $\tilde b_{n+1}$ such that
> $$||\mathbb{E}[H_{\theta_n}(X_{n+1})\mid\mathcal{F}\_n]-\nabla V(\theta_n)||\leq\tilde b\_{n+1}.$$
>     Applications, where this sufficient condition can be verified, are listed in Appendices C and D. We also refer reviewer ioBT to see how to verify H3$(i)$.
>
> Furthermore, we have demonstrated that our framework is applicable to numerous scenarios, including Bilevel Optimization and Conditional Stochastic Optimization, where all these assumptions are verified.

---

### Official Review · Reviewer_aKkE · 2024-07-21

**Soundness:** 3
**Presentation:** 3
**Contribution:** 3
**Rating:** 7
**Confidence:** 4

**Summary:**

This paper studies the non-asymptotic convergence guarantees of SGD with adaptive step sizes and (time-dependent) biased gradient estimators for nonconvex smooth functions. Applications to AdaGrad, RMSProp and Adam are developed. Numerical experiments on bilevel and conditional stochastic optimization, as well as deep VAE are used to illustrate the established theoretical results.

**Strengths:**

This work considers the biased gradient setting, which is away from the standard unbiased gradient setting like most standard results in the literature. This work also considers a more recently relaxed assumption on the gradient variance called “expected smoothness”, which makes the analysis even more general.

**Weaknesses:**

The bounded spectrum assumption of $A_n$ might not hold for every optimizer considered, and it remains to see how to relax it.

**Questions:**

Sometimes “adaptive step sizes” is referred to as “adaptive steps” in the paper. I guess using “adaptive step sizes” for all instances will avoid possible confusion. Can you also address the above point in weaknesses?

**Limitations:**

Yes.

---

> ### Author Rebuttal · Authors · 2024-08-06
>
> Thank you for your positive and thoughtful comments.
>
> We have changed "adaptive steps" to "adaptive step sizes" in the revised paper.
>
> In Theorems 4.1 and 4.2, since the exact form of $A_n$ is unknown (as we aim to cover all adaptive algorithms), we must assume certain properties about the preconditioning matrix, specifically the bound on its eigenvalues (H4). However, when applied to algorithms such as Adagrad, RMSProp, and Adam, we have shown that this assumption holds.
> It can be challenging to verify this assumption for some other algorithms, such as Stochastic Newton.
> As discussed after Assumption H4, the truncation method can be used for $A_{n}$ as done in [32].
> The truncation method involves replacing the random matrices $\tilde{A}\_n = \min${$||A\_n||,\beta\_{n+1}$}$A\_n/||A\_n||$). It is also stated that truncation is only used with very low probability, which means that the estimators $A_{n}$ are minimally affected by these truncations [32]. Since we choose $\beta_{n+1}$, we have control over the convergence rate.
>
> Although controlling the minimal eigenvalue is not strictly necessary, ensuring the boundedness of the gradient is generally sufficient to control this quantity in algorithms such as Adagrad, RMSProp, and Adam. Further details on controlling the minimum and maximum eigenvalues are provided in line 566 for Adagrad and line 572 for RMSProp. For Adam, the control is the same as for RMSProp since both use the same preconditioning matrix $A_n$.

---

> > ### Comment · Reviewer_aKkE · 2024-08-12
> > **Response to Rebuttal**
> >
> > Thanks so much to the authors for your rebuttal. Your rebuttal has addressed my concern.

---

### Official Review · Reviewer_BcqM · 2024-07-22

**Soundness:** 3
**Presentation:** 3
**Contribution:** 3
**Rating:** 7
**Confidence:** 3

**Summary:**

Stochastic and adaptive optimization algorithms are commonly used in advanced machine learning techniques. However, the analysis of non-convex optimization with biased gradients is lacking in the literature. This work considers the general scenario of optimizing machine learning objectives with practical optimizers given biased gradient estimators. Convergence bounds for different optimizers are proved with or without the Polyak-Łojasiewicz condition, and developed under an adaptive (preconditioning) matrix, which indicates the possibility of achieving the same convergence speed as with unbiased estimators by proper hyperparameter tuning.

The results are theoretically applied to bi-level optimization and others, empirically applied to importance weighted variational-autoencoder (IWAE) optimization. It is shown that the proved bounds apply to several applications. The experiments on Cifar-10 and FasionMNIST conclude that controlling the bias is beneficial to the convergence speed when the bias term dominates the optimizer term and the benefits become marginal after a certain threshold.

**Strengths:**

My overall judgement of the work is quite positive from its logic flow. However, I am not able to check all the details of the proofs.

The paper performs theoretical analysis of a common issue that covers a wide range of settings. Early in the paper, the background and application of adaptive stochastic approximation are established, which includes commonly used optimizers like SGD, Adagrad, RMSProp and Adam. The theoretical part is thorough with hyperparameters and different assumption groups, which is quite general. The most important assumption in the paper bounds the bias of the gradient, which also seems reasonable.

The settings of the IWAE experiment are not realistic, since an increasing number of samples are assumed during optimization. However, it can be viewed as a verification instead of an application of the bounds. As commented by the authors, the experiments with FasionMNIST indicate that the proved bound may be tight. The combination of theoretical and empirical evidences makes the paper strong.

**Weaknesses:**

Despite having wide applications, the experiments only include one case that verifies the proved bounds. The paper spent some efforts in bi-level optimization, but did not provide empirical analysis. IWAE has a remote connection to bi-level optimization when q is flexible enough, which could be utilized with a different optimization scheduling.

minor: line 135, there seems to be a redundant factor of 1/2 for $r_{n+1}$

**Questions:**

The variational distribution q may also affect the convergence. In the experiments, is q fixed or trained? I think the analysis applies to fixed q, but would like to learn about assumptions and implementation details here.

The introduction also talked about Markov chain based optimization schemes, but the analysis later in the paper seems to be far from it. How is assumption H3 achieved in such case? How are the theorems connected to the Martingale and Markov chain cases?

The main analysis in the paper is limited to optimization with vanishing bias, which does not directly apply to some practical methods (e.g. relaxation with Gumbel-softmax/concrete distribution, variational sequential Monte Carlo that drops part of the gradient). Can you provide a useful bound when r_n is a (or only bounded by a) constant from Theorem 4.2?

**Limitations:**

I do not see a limitation section and it would be beneficial to sketch future works by discussing limitations.

---

> ### Author Rebuttal · Authors · 2024-08-06
>
> Thank you for your positive and thoughtful review and the opportunity to clarify our contributions.
>
> $\textcolor{blue}{\textbf{Link Between IWAE and Bilevel Optimization}}$
>
> We agree that assuming an increasing number of samples during optimization in IWAE is unrealistic; it is used merely to illustrate our convergence results. In our experiments, we have illustrated our results in the PL case using an artificial example (see Appendix E.1). To avoid oversimplification, we also train the variational distribution $q$ rather than keeping it fixed. In this context, we are also in a biased gradient case, which is more aligned with Stochastic Bilevel Optimization.
> As you noted, there is indeed a connection between IWAE and bilevel optimization.
> The problem of VAE can be considered a Stochastic Bilevel Optimization problem, given by:
> $$\min_\theta V(\theta)=\mathbb{E}\_\pi[\log p_\theta(x)]=\mathcal{L}(\theta,\phi^*(\theta)) + \mathbb{E}\_\pi[\text{D}\_{KL}(q\_{\phi^*(\theta)}(z|x)||p\_\theta(z|x))]$$
> subject to
> $$\phi^*(\theta)\in\arg\min_\phi\mathcal{L}(\theta,\phi).$$
> Additionally, the IWAE problem can also be interpreted as Conditional Stochastic Optimization by setting $f$ to correspond to the identity function and $g_z((\theta,\phi),x)=\log p_\theta(x,z)/q_\phi(z|x)$ in Eq (14).
> All discussions on the link between IWAE and Stochastic Bilevel Optimization, as well as Conditional Stochastic Optimization IWAE, will be added in the experiments section. Additional details on the model configuration will be included in Appendix E in the revised paper.
>
> $\textcolor{blue}{\textbf{Discussion about the Martingale and Markov Chain Cases}}$
>
> As you mentioned in your comments, the purpose of Assumption H3 is to provide a very general framework that covers all possible applications and adaptive algorithms. In many cases, $X_n$ is i.i.d. (noise as a Martingale difference) or forms a Markov Chain.
> For IWAE, this applies to the Martingale case, but we also provide examples of applications involving the Markov Chain case, such as Sequential Monte Carlo Methods discussed in Appendix D.2.
>
> Since H3$(i)$ involves both the preconditioning matrix and the objective function, providing a necessary and sufficient condition in a general setting is challenging. However, we have shown that for Adagrad, Adam, and RMSProp (Corollary 4.5), a sufficient condition to verify H3$(i)$ is to bound the bias of the gradient, that is, to show that there exists $\tilde b_{n+1}$ such that
> $$||\mathbb{E}[H_{\theta_n}(X_{n+1})\mid\mathcal{F}\_n]-\nabla V(\theta_n)||\leq\tilde b\_{n+1}.$$
> Whether $X_n$ is i.i.d. or a Markov Chain, the goal is simply to find $\tilde b_n$. We have used this approach to verify H3$(i)$ for several applications, such as Stochastic Bilevel Optimization and Conditional Stochastic Optimization, which are discussed in Section 5.1 and Appendix C.
> Following also the comments of Reviewer ioBT, we provide the form of $\tilde b_n$ for both the i.i.d. and Markov cases.
>
> ### $\textbf{I.I.D case}$
> Let us assume that {$X_{n}, n \in \mathbb{N}$}  is an i.i.d. sequence. If the mean field function (the conditional expectation of the gradient $h(\theta_n)=\mathbb{E}[H_{\theta_n}(X_{n+1})\mid\mathcal{F}\_n]$) matches the gradient of the objective function, then there is no bias. Any bias arises from the difference between the mean field function and the true gradient of the objective function.
> In this case, with $\tilde b_{n+1}=||h(\theta_n)-\nabla V(\theta_n)||$, Assumption H3$(i)$ is verified.
>
> ### $\textbf{Markov Chain case}$
> We now assume that {$X_{n}, n \in \mathbb{N}$} is a Markov Chain. In this case, even if $h(\theta) = \nabla V(\theta)$, there is an additional bias introduced by the Markov Chain's properties. Specifically, the total bias consists of two components: one due to the difference between the mean field function and the true gradient of the objective function, and the other due to the Markov Chain’s characteristics.
> We define the stochastic update as:
> $$H_{\theta_{k}}\left(X_{k+1}\right)=\frac{1}{T}\sum_{i=1}^{T}H_{\theta_k}\left(X_{k+1}^{(i)}\right).$$
> When using $T$ samples per step to compute the gradient, if {$X_{n}, n \in \mathbb{N}$} is an ergodic Markov chain with stationary distribution $\pi$ and $||h(\theta_n)-\nabla V(\theta_n)||$ is bounded, Assumption H3$(i)$ is verified with $\tilde b_{n+1}=||h(\theta_n)-\nabla V(\theta_n)||+M\sqrt{\tau_\text{mix}/T}$, where $h(\theta)=\int H_{\theta}(x)\pi(dx)$ and $\tau_\text{mix}$ is the mixing time.
>
> If the general optimization problem reduces to the following stochastic optimization problem with Markov noise, as considered in most of the literature [77, 78, 79]:
> $$\min_{\theta\in\mathbb{R}^d}V(\theta):=\mathbb{E}\_{x\sim\pi}[f(\theta;x)], $$
> where $\theta\mapsto f(\theta;x)$ is a loss function, and $\pi$ is some stationary data distribution of the Markov Chain and $H_{\theta_k}(X_{k+1}^{(i)})=\nabla f(\theta_k;X_{k+1}^{(i)})$, then $\tilde b_{n+1}=M\sqrt{\tau_\text{mix}/T}$, similar to SGD with Markov Noise [78].
>
> ### $\textbf{Convergence Results in Both Cases}$
> Once we have the form of the bias of the gradient $\tilde b_n$, we can determine the convergence rate for our theorems. For instance, the convergence rate in Theorem 4.1 becomes $\mathcal{O}(n^{-\gamma+2\beta+\lambda}+||h(\theta_n)-\nabla V(\theta_n)||^{2})$ for i.i.d case and $\mathcal{O}(n^{-\gamma+2\beta+\lambda}+||h(\theta_n)-\nabla V(\theta_n)||^{2}+M^2\tau_{\text{mix}}/T)$ for Markov Chain case.
>
> $\textcolor{blue}{\textbf{The Case with Constant Bias}}$
>
> In fact, our analysis also applies to scenarios with a constant bias. For example, if the additive bias term $r_n$ is bounded by a constant $a$, the convergence rate becomes $\mathcal{O}(\log n/\sqrt{n}+a)$. To ensure that $\mathbb{E}\left[||\nabla V(\theta_{n})||^{2}\right]\leq\varepsilon$ for some $\varepsilon > 0$, the constant $a$ must be less than $\varepsilon$. Otherwise, this constant bias will adversely affect the convergence.

---

> ### Comment · Reviewer_BcqM · 2024-08-07
>
> Thank you for the rebuttal which addresses my questions. After reading the other reviews and responses, I do not have any additional concerns.

---

### Official Review · Reviewer_ioBT · 2024-07-25

**Soundness:** 3
**Presentation:** 3
**Contribution:** 2
**Rating:** 6
**Confidence:** 4

**Summary:**

The authors provide convergence guarantees for the biased adaptive stochastic approximation framework and establishes convergence to a critical point for non-convex setting of Adagrad type methods. The authors illustrate their results in the setting of IWAE.

**Strengths:**

Stated results for the rates of convergence of biased SA under Polyak-Łojasiewic condition are novel, yet the obtained rates are classical in the literature.

**Weaknesses:**

Checking assumption H3 in its current form is not obvious for different types of noise, since it involves taking expectation w.r.t. dynamics of the process $\{\theta_n\}$ itself. With this assumption the proof of the main results is rather classical, following standard techniques in non-convex optimization. At the same time, it is not clear how H3 can be checked even for independent sequence $\{X_n\}$. Checking the same assumption for $\{X_n\}$ being a Markov chain is even more tricky due to correlations between $X_n$ and $\theta_n$. It is typical that previous papers on the subject (see e.g. [Karimi et al, 2019]) state the assumptions, that do not involve expectations over $\theta_n$. I suggest the authors to clarify H3 for independent/martingale/Markov noise separately and to clarify respective assumption (when $X_n$ is a Markov chain - probably in terms of mixing time of this sequence). Provided that my concerns regarding H3 are resolved, I will be happy to increase my score.

Also the bibliography can be complemented by the paper [Dorfman and Levy, 2022], where the authors consider adaptivity of Adagrad-type algorithms to mixing time.

References:
[Dorfman and Levy, 2022] Dorfman, R. and Levy, K.Y. Adapting to mixing time in stochastic optimization with markovian data. In International Conference on Machine Learning 2022, pp. 5429-5446. PMLR.
[Karimi et al, 2019] Karimi, B., Miasojedow, B., Moulines, E. and Wai, H.T. Non-asymptotic analysis of biased stochastic approximation scheme. In Conference on Learning Theory, 2019, pp. 1944-1974.

**Questions:**

See the weakness section.

**Limitations:**

The authors should discuss in more details the types of considered data stream $(X_n)$ and its relations with H3.

---

> ### Author Rebuttal · Authors · 2024-08-06
>
> Thank you for your feedback, we give below some clarifications concerning H3. The purpose of Assumption H3 is to provide a very general framework that covers all possible applications and adaptive algorithms. Since H3 $(ii)$ is a well-known assumption [20], we understand that your question concerns Assumption H3 $(i)$.
>
> We would like to first stress that, with this assumption, the proofs of the main results (Theorems 4.1 and 4.2) still differ from that of biased SGD in the way in which we control the preconditioning matrix $A_n$. Furthermore, the application of these results to Adagrad and RMSProp, as well as the extension to Adam, are significantly different from the proof of biased SGD.
>
> Moreover, it is actually possible to verify Assumption H3 $(i)$ in many settings. It is important to keep in mind that this assumption is very general and depends on three quantities.
>
> - The type of adaptive algorithm (and therefore the form of the matrix $A_n$),
> - The application, that is, the gradient of the objective function $\nabla V(\cdot)$ and its empirical counterpart $H_{\theta_{n}} \left( \cdot \right)$,
> - The stochasticity of the sequence {$X_{n}, n \in \mathbb{N}$}.
>
> Let us first discuss the last point, which concerns the stochasticity of $X_{n}$. It is actually possible to avoid computing ``an expectation with respect to the dynamics of the process itself'', as mentioned in the review. Indeed, using the tower property, we have:
> $$
> \mathbb{E}\left[\left\langle\nabla V\left(\theta_{n}\right),A_{n}H_{\theta_{n}}\left(X_{n+1}\right)\right\rangle\right]=\mathbb{E}\left[ \mathbb{E}\left[\left\langle\nabla V\left(\theta\_{n}\right),A\_{n}H_{\theta\_{n}}\left(X\_{n+1}\right)\right\rangle |\mathcal{F}\_{n}\right]\right],
> $$
> where $(\mathcal{F}\_{n})\_{n\geq 0}$ represents the filtration generated by the random variables $(\theta\_{0}, X_1, ..., X_n)$.
> Then, we need to verify that
> $$
> \mathbb{E}\left[\left\langle\nabla V\left(\theta_{n}\right),A_{n}H_{\theta_{n}}\left(X_{n+1}\right)\right\rangle |\mathcal{F}\_{n}\right]\geq \lambda_{n+1}\left(||\nabla V(\theta_{n})||^{2}-r_{n+1}\right).$$
> Here, the stochasticity depends only on $X_{n+1}$, and not on the dynamics of the process {$\theta_{k}$}$_{k \leq n}$.
>
> Moreover, we have shown that for Adagrad, Adam, and RMSProp (Corollary 4.5), a sufficient condition to verify H3$(i)$ is to bound the bias of the gradient, that is, to show that there exists $\tilde b_{n+1}$ such that
> $$
> ||\mathbb{E}[H_{\theta_{n}}(X_{n+1})\mid\mathcal{F}\_{n}]-\nabla V(\theta_{n})||\leq\tilde b\_{n+1}.
> $$
>
> Then, whatever the assumptions on stochasticity, that is, whether $X_n$ is i.i.d. (the noise as a Martingale difference) or a Markov Chain, the goal is simply to find $\tilde b_n$. We have used this approach to verify H3$(i)$ for several applications such as Stochastic Bilevel Optimization and Conditional Stochastic Optimization (in which case, the sequence  {$X_{n}, n \in \mathbb{N}$} is arbitrary) or  Sequential Monte Carlo Methods, which corresponds to a Markov Chain case. This is discussed in Section 5.1 of the main paper and in Appendix C and D.
>
> However, we agree that this was not clear enough in the first version. We will use the extra page to clarify this point. To help the reader, we will give more details on the form of $\tilde b_n$ in the i.i.d. and Markov cases, as detailed below.
>
> #### $\textbf{I.I.D case}$
> Let us assume that {$X_{n}, n \in \mathbb{N}$}  is an i.i.d. sequence. If the mean field function (the conditional expectation of the gradient $h(\theta_{n}) = \mathbb{E}\left[H_{\theta_{n}}\left(X_{n+1}\right) \mid \mathcal{F}\_{n}\right]$) matches the gradient of the objective function, then there is no bias. Any bias arises from the difference between the mean field function and the true gradient of the objective function.
> In this case, with $ \tilde b_{n+1} = ||h(\theta_n) - \nabla V(\theta_n)|| $, Assumption H3$(i)$ is verified.
>
> #### $\textbf{Markov Chain case}$
> We now assume that {$X_{n}, n \in \mathbb{N}$} is a Markov Chain. In this case, even if $h(\theta) = \nabla V(\theta)$, there is an additional bias introduced by the Markov Chain's properties. Specifically, the total bias consists of two components: one due to the difference between the mean field function and the true gradient of the objective function, and the other due to the Markov Chain’s characteristics.
>
> We define the stochastic update as:
> $$H_{\theta_{k}}\left(X_{k+1}\right) = \frac{1}{T} \sum_{i=1}^{T} H_{\theta_{k}}\left(X_{k+1}^{(i)}\right). $$
>
> When using $T$ samples per step to compute the gradient, if {$X_{n}, n \in \mathbb{N}$} is an ergodic Markov chain with stationary distribution $\pi$ and $||h(\theta_n) - \nabla V(\theta_n)||$ is bounded, Assumption H3$(i)$ is verified with $\tilde b_{n+1} = ||h(\theta_n) - \nabla V(\theta_n)|| + M \sqrt{\tau_{\text{mix}}/T}$, where $h(\theta) = \int H_{\theta}(x) \pi(dx)$ and $\tau_{\text{mix}}$ is the mixing time defined in [78].
>
> If the general optimization problem reduces to the following stochastic optimization problem with Markov noise, as considered in most of the literature [77, 78, 79]:
> $$
> \min_{\theta \in \mathbb{R}^d} V(\theta) := \mathbb{E}\_{x \sim \pi} [f(\theta; x)],
> $$
> where $\theta \mapsto f(\theta; x)$ is a loss function, and $\pi$ is some stationary data distribution of the Markov Chain and $H_{\theta_{k}}\left(X_{k+1}^{(i)}\right)=\nabla f(\theta_{k}; X_{k+1}^{(i)})$, then  $\tilde b_{n+1}=M\sqrt{\tau_{\text{mix}}/T}$, similar to SGD with Markov Noise [78].
>
> ### $\textbf{Convergence Results in Both Cases}$
> Once we have the form of the bias of the gradient $\tilde b_n$, we can determine the convergence rate for our theorems. For instance, the convergence rate in Theorem 4.1 becomes $\mathcal{O}(n^{-\gamma+2\beta+\lambda} + ||h(\theta_n) - \nabla V(\theta_n)||^{2})$ for i.i.d case and $\mathcal{O}(n^{-\gamma+2\beta+\lambda} + ||h(\theta_n) - \nabla V(\theta_n)||^{2} + M^2 \tau_{\text{mix}}/T)$ for Markov Chain case.

---

> > ### Comment · Reviewer_ioBT · 2024-08-09
> >
> > Thank you for detailed answer. I will increase my score to 6.

---

### Author Rebuttal · Authors · 2024-08-06

We would like to express our gratitude to the reviewers for their helpful reviews and constructive feedback, which have helped us further improve our paper. Below, we provide a common response to comments made by several reviewers, followed by our point-by-point responses addressing all the specific concerns. Please note that all reference numbers correspond to those in the original paper, with any new references listed below.

Here are some comments on our results in the Martingale and Markov Chain cases, as requested by some reviewers. We provide below the convergence results for both the Martingale and Markov Chain cases. First, we want to highlight that our framework is more general and encompasses all possible applications and adaptive algorithms. In many cases, the data $X_n$ is i.i.d. (with noise as a Martingale difference) or forms a Markov Chain.

### $\textbf{Obtaining the form of the bias}$

We provide the bound $\tilde b_n$ of the gradient estimator $|| \mathbb{E}[H_{\theta_{n}}(X_{n+1}) \mid \mathcal{F}\_{n} ] - \nabla V(\theta_{n}) ||$ for both the i.i.d. and Markov cases.

#### $\textbf{I.I.D case}$
Let us assume that {$X_{n}, n \in \mathbb{N}$}  is an i.i.d. sequence. If the mean field function (the conditional expectation of the gradient $h(\theta_{n}) = \mathbb{E}\left[H_{\theta_{n}}\left(X_{n+1}\right) \mid \mathcal{F}\_{n}\right]$) matches the gradient of the objective function, then there is no bias. Any bias arises from the difference between the mean field function and the true gradient of the objective function.
In this case, $ \tilde b_{n+1} = ||h(\theta_n) - \nabla V(\theta_n)|| $.

#### $\textbf{Markov Chain case}$
We now assume that {$X_{n}, n \in \mathbb{N}$} is a Markov Chain. In this case, even if $h(\theta) = \nabla V(\theta)$, there is an additional bias introduced by the properties of the Markov Chain. Specifically, the total bias consists of two components: one due to the difference between the mean field function and the true gradient of the objective function, and the other due to the characteristics of the Markov Chain.

We define the stochastic update as:
$$H_{\theta_{k}}\left(X_{k+1}\right) = \frac{1}{T} \sum_{i=1}^{T} H_{\theta_{k}}\left(X_{k+1}^{(i)}\right). $$

When using $T$ samples per step to compute the gradient, if {$X_{n}, n \in \mathbb{N}$} is an ergodic Markov chain with stationary distribution $\pi$ and $||h(\theta_n) - \nabla V(\theta_n)||$ is bounded, then $\tilde b_{n+1} = ||h(\theta_n) - \nabla V(\theta_n)|| + M \sqrt{\tau_{\text{mix}}/T}$, where $h(\theta) = \int H_{\theta}(x) \pi(dx)$ and $\tau_{\text{mix}}$ is the mixing time defined below..

If the general optimization problem reduces to the following stochastic optimization problem with Markov noise, as considered in most of the literature [77, 78, 79]:
$$
\min_{\theta \in \mathbb{R}^d} V(\theta) := \mathbb{E}\_{x \sim \pi} [f(\theta; x)],
$$
where $\theta \mapsto f(\theta; x)$ is a loss function, and $\pi$ is some stationary data distribution of the Markov Chain and $H_{\theta_{k}}\left(X_{k+1}^{(i)}\right)=\nabla f(\theta_{k}; X_{k+1}^{(i)})$, then  $\tilde b_{n+1}=M\sqrt{\tau_{\text{mix}}/T}$, similar to SGD with Markov Noise [78].

### $\textbf{Convergence Results in Both Cases}$
Once we have the form of the bias of the gradient $\tilde b_n$, we can determine the convergence rate for our theorems. For instance, the convergence rate in Theorem 4.1 becomes $\mathcal{O}(n^{-\gamma+2\beta+\lambda} + ||h(\theta_n) - \nabla V(\theta_n)||^{2})$ for i.i.d case and $\mathcal{O}(n^{-\gamma+2\beta+\lambda} + ||h(\theta_n) - \nabla V(\theta_n)||^{2} + M^2 \tau_{\text{mix}}/T)$ for Markov Chain case.

- The mixing time $\tau_{\text{mix}}$ of a Markov chain with stationary distribution $\pi$ and transition kernel $P$ is defined as
$
\tau_{\text{mix}} := \inf $ { $t : \sup_{x} D_{\text{TV}}(P^t(x, \cdot), \pi) \leq \frac{1}{4}$},
where $D\_{\text{TV}}$ denotes the total variation distance.

[77] John C Duchi, Alekh Agarwal, Mikael Johansson, and Michael I Jordan. Ergodic mirror descent. SIAM Journal on Optimization, 22(4):1549–1578, 2012.

[78] Ron Dorfman and Kfir Yehuda Levy. Adapting to mixing time in stochastic optimization with markovian data.
In International Conference on Machine Learning, pages 5429–5446. PMLR, 2022.

[79] Aleksandr Beznosikov, Sergey Samsonov, Marina Sheshukova, Alexander Gasnikov, Alexey Naumov, and Eric
Moulines. First order methods with markovian noise: from acceleration to variational inequalities. In Advances
in Neural Information Processing Systems, volume 36, 2024.

[80] Jingzhao Zhang, Tianxing He, Suvrit Sra, and Ali Jadbabaie. Why gradient clipping accelerates training: a theoretical justification for adaptivity. In International Conference on Learning Representations, 2020.

[81] Yin Liu and Sam Davanloo Tajbakhsh. Adaptive stochastic optimization algorithms for problems with biased
oracles. arXiv preprint arXiv:2306.07810, 2023.

[82] Haochuan Li, Alexander Rakhlin, and Ali Jadbabaie. Convergence of adam under relaxed assumptions. Advances
in Neural Information Processing Systems, 36, 2024.

---

### Decision · Program_Chairs · 2024-09-25

**Decision:**

Accept (poster)

**Comment:**

This paper studies the convergence rate for a set of biased adaptive stochastic estimations (e.g., Adagrad, RmsProp), which is different from the prior works on the unbiased estimations. Most reviewers acknowledge the theoretical contributions of this paper and suggest acceptance. The authors should take the reviewers' comments (especially Reviewer nF8X) into consideration in the revision.